# Perovskite solar cells with enhanced thermal fatigue resistance under extreme temperature cycling

Cem Yilmaz[1,10], Ali Buyruk[1,10], Yating Shi[2,10], Sergej Levashov [3], Xiaole Li[4], Rik Hooijer [1], Jian Huang [1], Hao Zhu[1], Oliver Fischer [5,6], Martin C. Schubert [5,6], Caner Deger [7], Ilhan Yavuz [7], Esma Ugur[1], Gilles Lubineau [4], Johanna Eichhorn [3], Fei Zhang [2,8,9] ✉ & Erkan Aydin [1] ✉

Metal halide perovskite solar cells combine high power density with low-cost manufacturing, but durability under repeated extreme temperature cycling remains insufficiently understood. We investigate thermal fatigue under cycling between −80 °C and +80 °C as an accelerated stress protocol. Mismatched thermal expansion between the perovskite absorber and glass substrate induces biaxial tensile strain, leading to degradation at the substrate–perovskite interface and within grain boundaries. To mitigate these failure modes, we introduce a co-additive molecular strategy based on lipoic acid, dihydrolipoic acid, and a sulfonium-based derivative to enhance interfacial adhesion, while in situ polymerization during annealing reinforces grain-boundary cohesion. This dual reinforcement improves robustness and performance, achieving stabilized efficiencies of 26% under standard solar illumination. Devices retain 84% of initial efficiency after 16 extreme temperature cycles. Our experiments reveal that thermal exposure duration is more critical than cycle number, with most degradation occurring during initial cycles.

Metal halide perovskite solar cells (PSCs) have attracted significant attention due to their high power conversion efficiencies, low fabrication costs, and compatibility with lightweight device architectures[1]. These attributes make PSCs attractive for applications in which high specific power and mechanical compliance are required. They present a potential alternative to conventional III–V-based photovoltaic technologies, which are well known for their robustness under extreme operating conditions but are limited by high material and fabrication costs and relatively short operational lifetimes[2]. However, the long-term mechanical and structural stability of layered perovskite device stacks under repeated thermomechanical stress remains a key challenge limiting their broader deployment.

Repeated temperature cycling induces volumetric expansion and contraction within the solar cell stack, leading to mechanical fatigue and, ultimately, delamination, crack formation, or failure at weakly bonded interfaces[2,3]. Such effects are particularly pronounced in multilayer thin-film devices due to mismatches in the coefficients of thermal expansion (CTE) between adjacent layers. In PSCs, large CTE

[1]Department of Chemistry, Ludwig-Maximilians-Universität München (LMU), Butenandtstraße 11 (E), Munich, Germany. [2]School of Chemical Engineering and Technology, Tianjin University, Tianjin, China. [3]Physics Department, School of Natural Sciences, Technical University of Munich, Am Coulombwall 4, Garching, Germany. [4]Mechanics of Composites for Energy and Mobility Lab, King Abdullah University of Science and Technology (KAUST), Thuwal, Saudi Arabia. [5]Fraunhofer Institute for Solar Energy Systems ISE, Heidenhofstr, Freiburg, Germany. [6]Chair of Photovoltaic Energy Conversion, Department of Sustainable Systems Engineering INATECH, University of Freiburg, Emmy-Noether-Str, Freiburg, Germany. [7]Department of Physics, Marmara University, Ziverbey, Türkiye. [8]Collaborative Innovation Center of Chemical Science and Engineering (Tianjin), Tianjin, China. [9]Haihe Laboratory of Sustainable Chemical Transformations, Tianjin, China. [10]These authors contributed equally: Cem Yilmaz, Ali Buyruk, Yating Shi. ✉e-mail: fei_zhang@tju.edu.cn; erkan.aydin@cup.uni-muenchen.de

differences between the glass substrate ($3.7 \times 10^{-6} K^{-1}$)[4], the transparent conductive oxide (TCO) (e.g., ITO, $8.5 \times 10^{-6} K^{-1}$)[4], and the perovskite absorber (e.g., FAPbI$_3$, $-203 \times 10^{-6} K^{-1}$)[5] concentrate strain at grain boundaries and heterointerfaces, accelerating mechanical degradation during temperature cycling.

Extreme temperature cycling represents a particularly severe stress condition for thin-film photovoltaic devices and is encountered in several scenarios, including high-altitude platforms, aerospace systems, and accelerated laboratory stress testing. In comparison to standard terrestrial qualification protocols—such as IEC thermal cycling tests typically limited to −40 °C to +85 °C—these conditions involve wider temperature excursions and often faster thermal ramp rates, resulting in rapid stress evolution within the device stack. Environments, such as low-Earth orbit (LEO) are frequently cited as representative examples of extreme thermal cycling conditions, where repeated transitions between sunlight and shadow lead to pronounced temperature fluctuations, with cycling frequencies approaching ~6000 cycles per year and thermal ramp rates on the order of 4 °C–5 °C min$^{-1}$, resulting in rapid stress evolution within the solar cell stack[6–9]. Recent systematic studies have also examined perovskite solar cell performance under extreme temperature cycling and in-orbit conditions, further highlighting the importance of understanding thermal fatigue for space deployment[10–12]. Understanding thermally induced mechanical fatigue under these conditions is therefore essential for improving the durability of thin-film photovoltaic technologies for extreme temperature conditions.

Previous studies have demonstrated that incorporating functional additives into perovskite precursor solutions can improve film quality, defect tolerance, and mechanical resilience. In particular, polymerizable or cross-linkable additives are introduced into the perovskite precursor, polymerizing during annealing and localizing at grain boundaries to heal defects and improve film quality[13]. For example, polymers containing isocyanate groups include poly(oxime-urethanes)[14] and polyurea with PDMS blocks[15], polymers with disulfide groups include polyurethane elastomers (PUDS)[16], polyurethanes with pendant fullerene units (C$_{60}$-PU)[17], and polyurethanes without fullerene moieties[18]. Finally, polymers with carboxylic acid groups include poly(LA)[19] and its salt form, poly(TA-NI)[20], which is based on a hydrazide-derived ammonium ion. Across all these systems, thermal-triggered healing has resulted in more than 80% recovery of solar cell performance after stress testing[11–18], while systems based on poly(LA) and poly(TA-NI) demonstrated recovery rates exceeding 90%[17,18]. Notably, the behavior of such materials under thermal cycling tests, particularly their protective effects on PSCs, remains unexplored and represents a significant knowledge gap in this field.

To improve adhesion at the interface between the TCO and the perovskite layer, several surface-engineering strategies have been developed. Reported approaches include increasing surface hydroxylation by replacing crystalline TCOs with amorphous ones[21,22], removing terminal hydroxyls and hydrolysis byproducts via combined HF and UV–ozone treatments[23], introducing hetero-chiral linker molecules[24], and employing self-assembled molecules (SAMs), such as iodine-terminated carbazole derivatives or bifunctional thiol–carboxylic acid systems with varying alkyl chain lengths[25]. However, most reported studies focus primarily on initial device performance or environmental stability under moisture or light exposure, while systematic investigations of interfacial robustness under repeated extreme temperature cycling remain limited.

In this work, we use 5-(1,2-dithiolan-3-yl)pentanoic acid (α-lipoic acid, LA) and functionalize it further as part of our two-step reinforcement strategy, and the synthetic procedures are provided in Supplementary Fig. 1, while the NMR analyses are shown in Supplementary Fig. 2–4. To promote inter-grain connectivity, LA is incorporated into the perovskite precursor solution with the expectation that it can undergo in situ polymerization during crystallization (as shown in Fig. 1b). Specifically, we aim to investigate whether, upon heating above 70 °C, the disulfide bonds in LA would initiate thermally driven ring-opening polymerization, and whether, during cooling, the carboxylic acid groups would form stable hydrogen-bonded dimers. Together, we aim for these processes to enhance cross-linking and mechanical cohesion at the grain boundaries. For interfacial reinforcement between the TCO and perovskite, we aim to enhance chemical interactions beyond the disulfide bonding provided by LA and synthesize and explore its derivatives. Furthermore, to approximate the mechanical stresses associated with these rapid and extreme temperature shifts, we adopt a custom thermal cycling protocol spanning −80 °C to +80 °C, consistent with practical satellite temperature measurements and recent thermal stress studies on PSCs[26–28], which necessitates the development of a dedicated test setup. We systematically evaluate the impact of this dual reinforcement approach on the mechanical durability and performance stability of PSCs.

## Results
### Crosslinking of LA in the Grain Boundaries

LA molecules polymerize at the grain boundaries (as sketched in Fig. 1a, b) within the perovskite bulk during the thermal annealing process, as they are unable to fit the perovskite crystal lattice, due to the large molecular size of both the monomeric and polymeric forms[19,29]. X-ray photoelectron spectroscopy (XPS) results indicate that the poly(LA) primarily interacts with the perovskite lattice through the −COOH side group, which does not participate in polymerization, as shown in Supplementary Fig. 5a, specifically via hydrogen bonding between the −OH group of poly(LA) and perovskite octahedra, −OH⋯IPbI$_5$. Additionally, the =O group (a Lewis base) within the same carboxylic acid functional group is shown to coordinate with under-coordinated Pb$^{2+}$ ions (acting as Lewis acids), as evidenced by the energy shifts (Supplementary Fig. 5b, c) observed in the orbitals of the involved lead atoms[19]. Indirect evidence of LA polymerization is provided by the FTIR spectra (Supplementary Fig. 6a), where the −COOH stretching vibration shifts from 1690 to 1703 cm$^{-1}$ under similar thermal conditions, suggesting the occurrence of polymerization[30]. To further investigate the possible localization of poly(LA) within the perovskite bulk structure, particularly at the grain boundaries, high-angle annular dark-field scanning transmission electron microscopy (HAADF-STEM) and the corresponding energy-dispersive X-ray spectroscopy (EDX) elemental mapping were performed, as shown in Supplementary Fig. 6b, c. Furthermore, XRD analysis revealed that the intensity of the excess PbI$_2$ signal observed in the control group (Supplementary Fig. 7a) was reduced in the LA-treated films, suggesting a potential chemical interaction between LA and PbI$_2$. This interaction is further supported by $^1$H-NMR spectra of LA and its mixture with PbI$_2$ (Supplementary Fig. 7b), which show a slight upfield shift of the proton signal corresponding to the carbonyl group of LA (acting as a Lewis base), from 12.00 to 11.99 ppm[31]. Finally, GIWAXS measurements performed on the same films, as shown in Supplementary Fig. 8, revealed no change in the orientation of the perovskite planes and no additional reflections, indicating the absence of new phases.

Nanoscale atomic force microscopy (AFM) analysis with the quantitative nanomechanical characterization technique revealed that compared to the control group, the perovskite films, which have LA and its derivatives in combination with SAM (4-(7H-dibenzo[c,g]carbazol-7-yl)butyl)phosphonic acid, 4PADCB) at the substrate interface − with a fixed amount of LA incorporated into the bulk − exhibited enhanced adhesion at grain boundaries. Notably, DHLA resulted in an increase of over 50% in average adhesion at grain boundaries, while DMSLA led to an increase of approximately 40%; in contrast, the LA form did not cause a significant change. Note that any overestimation of adhesion forces due to increased surface contact at grain boundaries is minimal, since the depth and width of the grain boundaries at the perovskite/SAM interface are larger than the tip apex curvature.

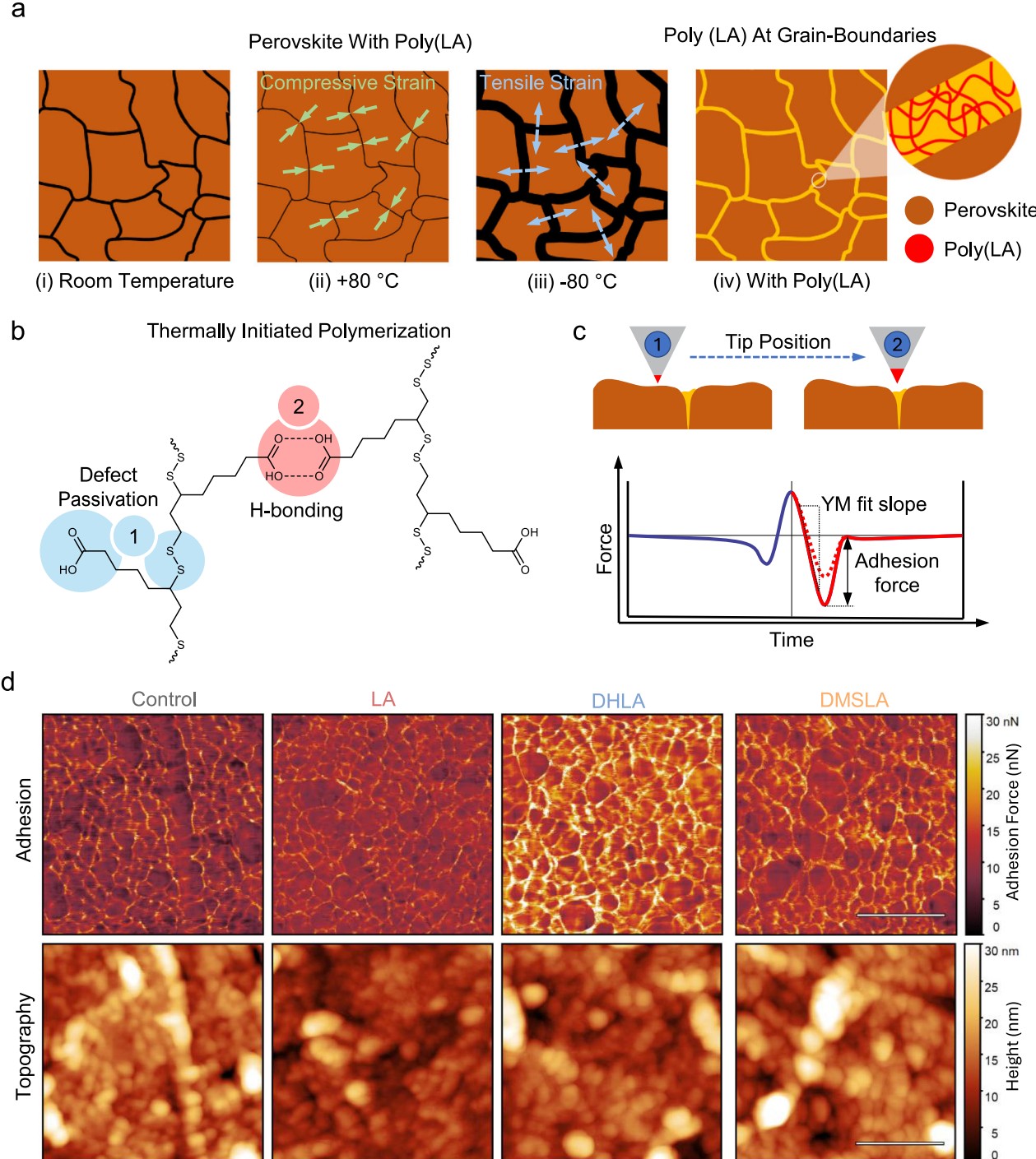

**Fig. 1 | Nanoscale mechanical behavior of perovskite film surface and grain boundaries. a** A sketch demonstrating the thermal behavior of perovskite poly-crystalline structures with (iv) and without poly(LA) (i-iii) at the grain boundaries. **b** Thermal polymerization of LA and its multifunctional role at grain boundaries, facilitating defect passivation (1) and hydrogen bonding interactions (2) between polymer chains. **c** A sketch for the working mechanism of the surface-mapping tip-based nano-mechanical analysis technique. Numbers 1 and 2 indicate the

perovskite grains and the poly(LA)-rich grain boundaries, respectively. **d** Adhesion forces were obtained from the perovskite surface at the HTL contact region using a tip-based nanomechanical surface-mapping technique, alongside topographical imaging. The values shown in the yellow boxes are the average adhesion forces extracted from each corresponding image. The white scale bar corresponds to 400 nm. Identical color scales are used for all samples within each row.

(Fig. 1c). Hence, the enhanced adhesion can be primarily attributed to the polymer, which preferentially localizes at the grain boundaries and plays a significant role in improving adhesion in these critical regions.

Young's modulus values derived from the same imaging dataset (Supplementary Fig. 9), along with adhesion and modulus measurements from the top surfaces of control and LA-containing perovskite films, are presented in the Supporting Information (Supplementary Fig. 10). Additionally, an increase in inter-grain adhesion was observed, which may arise from two contributing factors: (1) enhanced adhesion at the grain boundaries due to polymer cross-linking, and (2) a potential overestimation of mechanical properties resulting from an increased contact area at the grain boundaries (Fig. 1c, bottom line).

Given that the control sample exhibits lower adhesion at the grain boundaries despite having a similar boundary depth, the observed adhesion enhancement can be attributed to the cross-linked polymer network. In contrast, we observe no change in the Young's modulus of the grains (excluding regions with exposed polymer), while a decrease in modulus at the interface is evident, likely due to the inherently lower stiffness of the polymer.

## Linker Molecules at the HTL Perovskite Interface

We engineered the ITO/perovskite interface by co-functionalizing the surface with lipoic acid derivatives together with the standard SAM molecule, 4PADCB, to strengthen interfacial adhesion. All molecules featured carboxylic acid (–COOH) anchoring groups for binding to the TCO electrode. To diversify binding interactions, we introduced sulfur-containing head groups: the disulfide ring in LA, the free thiol (–SH) groups in dihydrolipoic acid (DHLA, 6,8-dimercaptooctanoic acid), and the sulfonium cation in 7-carboxy-3-(methylthio)heptyl dimethyl-sulfonium (DMSLA). The molecular structures of the linker molecules are shown in Fig. 2a.

We examined the adhesion strength of SAM-linker contacts at the ITO/perovskite interface through pull-off testing. Indeed, as such systems are known to have intrinsically a very low toughness, it is important to inhibit any initiation of delamination between the layers. Consequently, adhesion strength is the appropriate metric for quantifying the mechanical integrity of such systems. For this, a stack consisting of glass/ITO/4PADCB- linkers/perovskite/PMMA was prepared, and a dolly was attached to the top surface using an adhesive. PMMA served as an interlayer to protect the underlying layers from the damage of epoxy glue and improve the adhesion strength between the glue and the specimen surface, as shown in Fig. 2b. From the resulting load–displacement graphs, we extracted the interfacial adhesion strength, which increased from 3.61 MPa for the control sample to 4.89 MPa for the DMSLA-treated samples (Fig. 2c). Pull-off tests showed a wide distribution of adhesion strength across all samples, likely arising from small preparation imperfections, such as slight misalignment angles or adhesive layer thickness differences, yet DMSLA-based samples consistently showed the highest average strength. The resulting stress distributions follow the same trend, indicating that DMSLA-treated interfaces require greater force to induce interfacial failure (Fig. 2d).

Beyond macroscopic mechanical analysis, we performed XPS at the substrate/perovskite interface, analyzing each side independently after mechanical peeling. Example samples included LA in the SAM precursor, noting that perovskite also incorporates LA. On the glass–ITO side, the –COO signal was 2.01%, corresponding to the –COOH anchoring groups of the linker molecules. On the perovskite side, the –COO signal was 6.45% for samples without LA in the SAM precursor (Supplementary Fig. 11). In contrast, with LA in the SAM precursor, the –COO signal decreased to 0.96% on the glass–ITO side but increased to 7.09% on the perovskite side. These results indicate a strong interaction between LA in the perovskite layer and LA in the SAM.

Further, we performed Density Functional Theory (DFT) analyses of the bare SAM (4PADCB), LA, and its derivatives–both individually and in mixtures–on the ITO surface and in contact with the adjacent perovskite layer. Among the investigated molecules, DMSLA exhibits the strongest interaction with the perovskite surface, as evidenced by its high interfacial interaction energy, as shown in Fig. 2e-f. In the planar-averaged charge density difference (CDD) plot (Supplementary Fig. 12), DMSLA displays the broadest and most pronounced peak at the interface region, suggesting a significantly enhanced interfacial charge transfer. Furthermore, the isosurface representation of the CDD indicates extensive charge accumulation and depletion regions spanning both the molecular layer and the perovskite substrate, further confirming the superior electronic coupling. In addition, the co-attachment of LA and its derivatives with 4PADCB on the ITO surface, as well as the interactions between the SAM and LA-derived molecules anchored to the ITO surface and the perovskite at the top contact, were investigated using DFT calculations. All related data are provided in the Supporting Information, Supplementary Figs. 13–14. This trend is consistent with our pull-off test results (Fig. 2c, d), which show that the sulfonium cation head group establishes the strongest and most favorable chemical bonding with the perovskite layer. Accordingly, the observed macroscopic improvement in adhesion strength can be attributed to the cumulative effect of these strong and chemically effective interfacial interactions.

We validated the binding of SAM-linker mixtures to the ITO surface using XPS, through the detection of characteristic –COOH-related signals around 289 eV, as shown in Supplementary Fig. 15. To quantify the surface coverage factor of SAMs, we employed cyclic voltammetry (CV) measurements using glass/TCO/SAM-linker stacks[32]. The coating density was calculated through the integration of the redox peak corresponding to the electroactive moiety anchored via the SAM. This method allows an indirect estimation of the surface coverage by correlating the charge passed during the redox process to the number of active molecules present on the surface. The coating density of the SAM molecules–both in their pure forms (controls) and in mixtures with linkers–was calculated using Eq. (1).

$$i_p = \frac{n^2 F^2}{4RTN_A} A\Gamma^* \nu \qquad (1)$$

Here, $i_p$ is the oxidative peak current (A), $\nu$ is the scan rate (V s⁻¹), $n$ is the number of electrons transferred, $F$ is the Faraday constant (96,485 C mol⁻¹), $R$ is the universal gas constant (8.3144 J K⁻¹ mol⁻¹), $T$ is the temperature (K), $N_A$ is Avogadro's number ($6.022 \times 10^{23}$ mol⁻¹), $A$ is the electrode surface area (1.5 cm²), and $\Gamma^*$ (molecules cm⁻²) is the surface coverage, which can be determined from the slope of the $i_p$ versus $\nu$ plot. SAM forms coated on the ITO surface in mixtures with linkers exhibited higher surface coverage compared to the pure SAM, with coating densities reaching the highest value with DMSLA, with $3.75 \times 10^{13}$ molecules cm⁻², while only 4PADCB is $2.84 \times 10^{13}$[33]. Details for the analysis are provided in Supplementary Fig. 16.

Ultraviolet photoelectron spectroscopy (UPS) was used to analyze energy-level alignment between different contact stacks and the crystallized perovskite bulk (with LA) (Supplementary Fig. 17). The smallest energy gap between the ITO work function–modified by the SAM mixture–and the perovskite valence band, critical for efficient charge transfer, was observed for the DMSLA-blended 4PADCB (Supplementary Table 1). This notable downshift in the ITO work function can be attributed to the large dipole moment of DMSLA (31.9 D, see results in Supplementary Fig. 18), which induces an interfacial dipole layer and modifies the surface vacuum level. The reduced energy difference between the levels involved in hole transfer is known to facilitate hole extraction at the modified interface.

Scanning electron microscopy (SEM) was performed to investigate the effects of LA and its derivatives, applied with the SAM layer on ITO substrates, on the perovskite/ITO interface and bulk perovskite morphology. SEM micrographs revealed that the incorporation of LA and its derivatives, both within the bulk and at the SAM–perovskite interface, led to increased grain sizes in the perovskite films. As shown in Supplementary Fig. 19 and supported by the XRD data in Supplementary Fig. 7a, this structural change is accompanied by an improvement in the crystallinity of the perovskite layer. In contrast to the control films, which show visible pinholes, particularly in cross-sectional images, the blended films display a more compact and pinhole-free morphology (Supplementary Figs. 19–21). These findings suggest that the additives influence the number and size of perovskite crystal grains formed during crystallization, thereby affecting grain

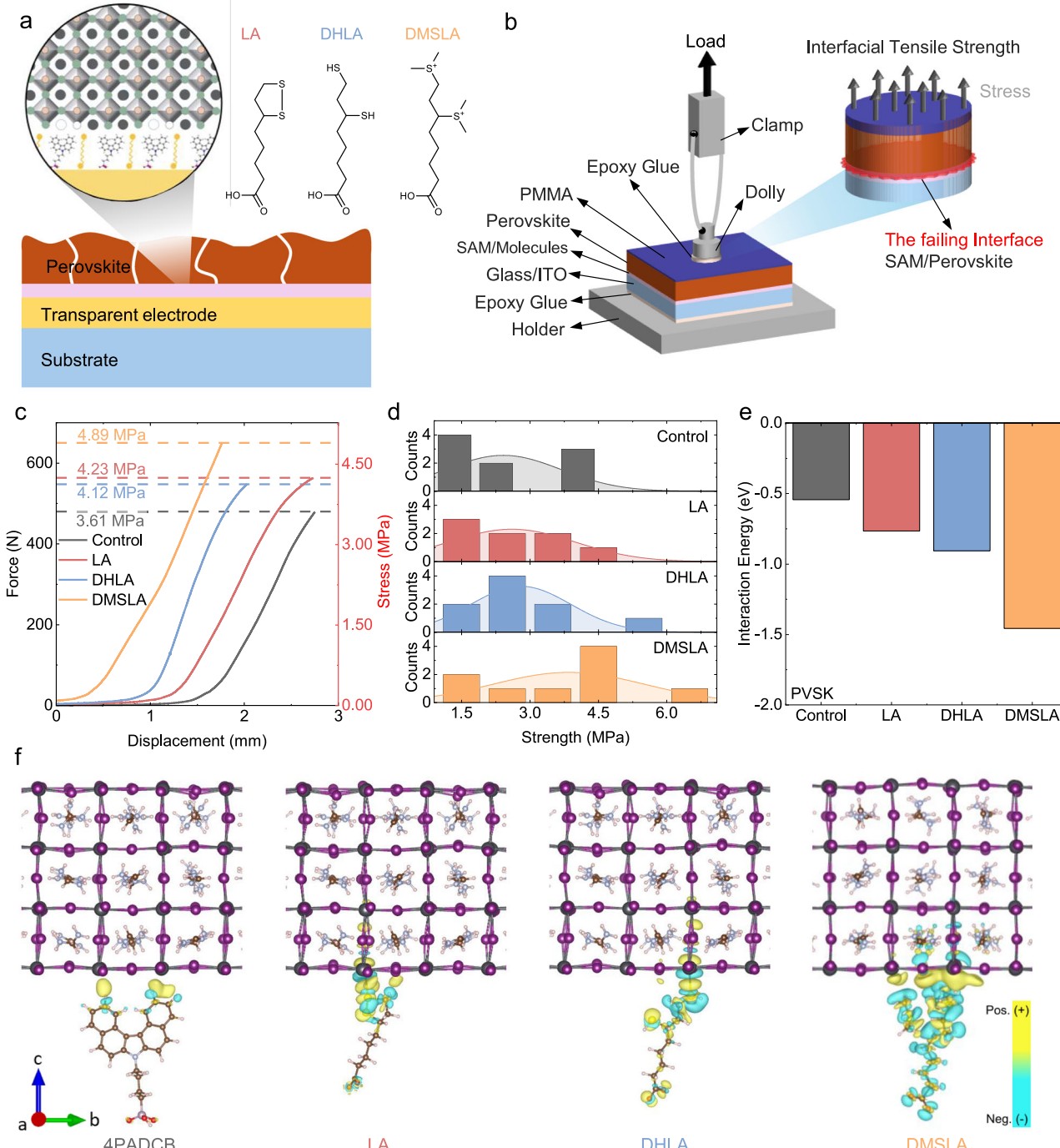

**Fig. 2 | Mechanical analysis of substrate-perovskite interfaces. a** Schematic representation of the HTL contact formed by 4PADCB together with LA, DHLA, and DMSLA at the ITO/perovskite interface. **b** Sketch for the test setup used for the pull-off test. **c** Maximum adhesion strength, and **d** stress values, which are determined from the highest value in each of the four groups. **e** The interaction energies between perovskite and the linker molecules, calculated using a computer-based DFT approach. **f** Interactions between perovskite and the molecules.

size distribution and the prevalence of multi-domain structures[29]. This indicates improved film uniformity and domain connectivity at the interface [25].

## Solar Cell Performances

We fabricated solar cells with the structure of ITO/4PADCB-linkers/$Cs_{0.05}MA_{0.10}FA_{0.85}PbI_3$ (w/o LA)/$C_{60}$/BCP/Ag, as shown in Fig. 3a. Detailed device fabrication for all devices is provided in the Methods section. From $J$-$V$ analysis, we found that the target devices —featuring DMSLA at the HTL interface and LA in the perovskite bulk—exhibited

the highest performance, as shown in Fig. 3b. This device achieved a PCE of 25.21%, with a $V_{OC}$ of 1.16 V, a $J_{SC}$ of 26.06 mA cm$^{-2}$, and a FF of 0.83. We further verified the proposal molecules by transferring our linker approach to colleagues in Tianjin University, and their independent molecule synthesis and device fabrication results showed a close match to those measured in our laboratory, LMU Munich: PCE of 25.98%, with $V_{OC}$ of 1.16 V, $J_{SC}$ of 26.35 mA cm$^{-2}$, and FF of 0.85 (steady-state PCE: 25.51%, in the Fig. 3c). The slight variation in device performances is due to the different perovskite processing recipes and conditions as given in the Methods section.

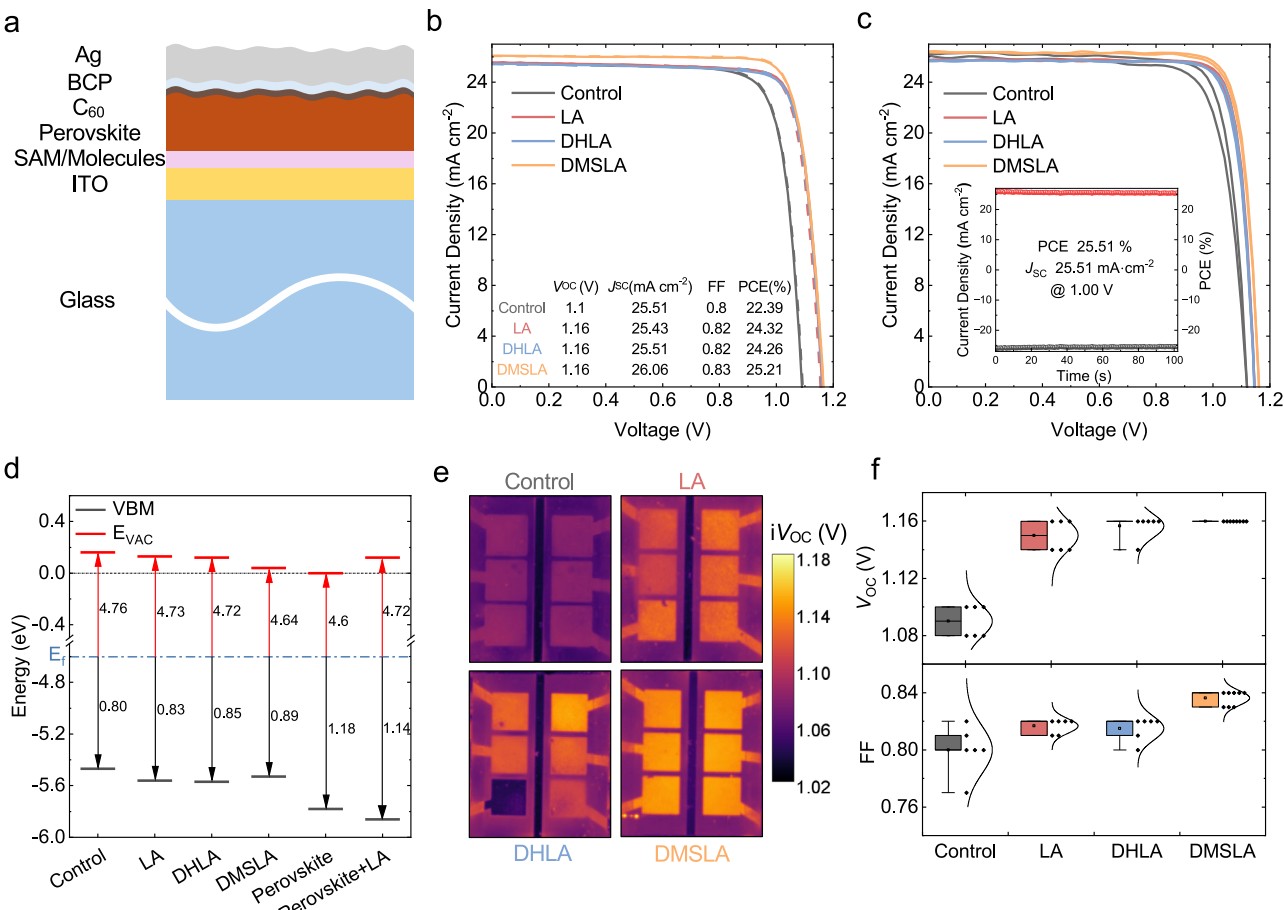

**Fig. 3 | Performance of perovskite solar cells. a** Schematic illustration of the fabricated perovskite solar cells. **b** J–V curves and the corresponding device parameters of the best-performing solar cell devices fabricated at the LMU Munich, and **c** the ones fabricated at Tianjin University. The inset figure shows the steady-state power conversion efficiencies of the champion device. **d** Relative energy levels of the perovskite and ITO/SAM stacks derived from UPS measurements. **e** Photoluminescence (PL) based implied open-circuit Voltage (iV_OC) images

acquired over the full device area (six pixels per device; pixel area: 4 × 4 mm²). **f** Statistical distribution of the V_OC and FF values for the solar cell devices. In the combined box–violin plots, the center line indicates the median; the box limits represent the 25 and 75th percentiles; whiskers extend from the minimum to the maximum values; squares denote the mean; individual points represent the measured data values; and the overlaid curves represent the data distribution density. Error bars indicate standard deviation.

As can be seen from the energy level diagram derived from the UPS data presented in Fig. 3d—which is critical for charge transfer at the perovskite/SAM interface—the combination of 4PADCB with DMSLA results in the smallest energy offset (0.25 eV) between the valence band maximum (VBM) of the modified ITO and that of the LA-doped perovskite. Furthermore, considering the intrinsic dipole moments calculated for the LA derivatives (Supplementary Fig. 18) used in combination with SAM, the significantly large dipole moment of DMSLA and its induced shift in the work function of ITO further support this favorable alignment. Taken together, these findings rationalize the superior $J_{SC}$ and FF observed for the 4PADCB + DMSLA SAM formulation, shown in Fig. 3b–f, compared to the control and other LA derivatives. We assign the enhanced $V_{OC}$ of the devices to the effective passivation of crystal defects at the substrate/perovskite interface by LA and its derivatives, and within the perovskite bulk by LA alone. This dual-passivation approach at both the SAM/perovskite interface and the grain boundaries suppresses trap-assisted recombination throughout the structure, thereby leading to an increase in open-circuit voltage. We investigated the charge carrier recombination dynamics of the samples via steady-state photoluminescence (PL) and time-resolved photoluminescence (TRPL) measurements. The perovskite layers on 4PADCB-linker stacks exhibited slightly higher PL intensity and extended carrier lifetimes according to the TRPL results, compared to the only 4PADCB-based samples (Supplementary

Fig. 22c, d). PL-based implied open-circuit voltage (iV_OC) images[34] in Fig. 3e, acquired over the full device area (six pixels per device), show spatially uniform voltage distributions that are consistent with the electrical performance trends observed in the corresponding solar cells, indicating that the introduced interlayer does not induce macroscopic inhomogeneity. When evaluated alongside the $V_{OC}$–FF comparison in Fig. 3f, the iV_OC values derived from PL are slightly lower than the corresponding $V_{OC}$ values, as expected; nevertheless, both measurements—performed at different institutes and at different times—exhibit consistent trends in V_OC enhancement across the device series. The accuracy of the $J_{SC}$ values was validated through integration of the external quantum efficiency (EQE) spectra (Supplementary Fig. 23), yielding consistent results with the measured $J_{SC}$, specifically 25.10 mA cm⁻². Additional statistical distributions of the device performance parameters from LMU Munich and Tianjin University are provided in the Supporting Information (Supplementary Figs. 24–25).

**Thermal Fatigue Stability Analysis**

We performed the thermal fatigue test using a custom-built closed-lid setup, as standard climate chambers could not reach the required –80 °C cryogenic range, which induces the highest stress at the perovskite–substrate interface. The samples—one device per condition, each comprising six pixels—were cycled between –80 °C and +80 °C in a stainless-steel container, with controlled heating and

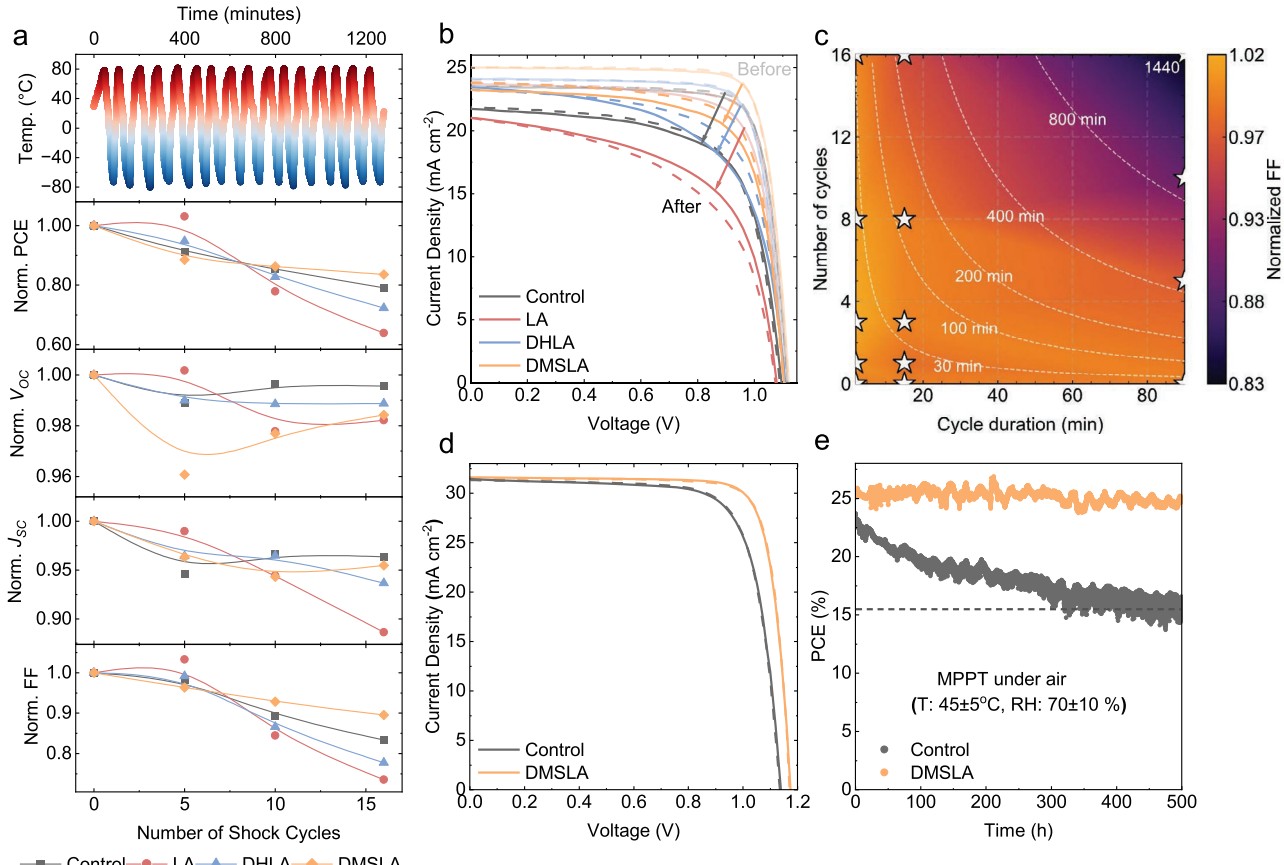

**Fig. 4 | Thermal fatigue test of perovskite solar cell. a** Temperature profile of the thermal cycling protocol and the corresponding evolution of device parameters over sixteen cycles. **b** *J–V* characteristics of representative devices measured under ~1000 W m⁻² equivalent 1-sun intensity illumination before and after thermal cycling. **c** Contour maps showing the dependence of normalized fill factor (FF) on thermal cycle duration and number of cycles for control perovskite solar cells; white dashed lines indicate contours of constant total thermal-exposure time **d** *J–V* characteristics of representative devices measured under ~1360 W m⁻² illumination (AM0 approximation) **e** MPPT results of encapsulated perovskite solar cells under ~1000 W m⁻² equivalent 1-sun intensity, measured at 45 °C ± 5 °C and 70 ± 10% relative humidity. In panels b and d, solid lines denote forward scans and dashed lines denote reverse scans.

cooling rates of +3.40 and −3.80 °C min⁻¹, respectively, and a total cycle duration of ~90 min under dry ambient conditions (Fig. 4a, b). The 90 min cycle duration was chosen to approximate a symmetric heating–cooling sequence (~45 min per half-cycle), reflecting characteristic thermal timescales reported for low-Earth-orbit environments[35] and ensuring sufficient thermal equilibration of the full device stack. A comparative summary of reported thermal-cycling studies on PSCs under space-relevant or terrestrial conditions is provided in Supplementary Table 2. Thermal cycling was continued until the performance of the target devices dropped below ~90% of their initial power conversion efficiency. A schematic and further explanation of the experimental setup are provided in Supplementary Fig. 26. After 16 thermal cycles under the applied temperature protocol, encapsulated solar cells with DMSLA retained 84% of their initial performance, compared to 79% for the control group. In contrast, the solar cells treated with LA and DHLA showed inferior stability, retaining only 64% and 72% of their initial performance, respectively. Performance degradation was observed across all photovoltaic parameters, with the most pronounced losses occurring in the FF. As shown in Fig. 4a, control devices exhibited an FF reduction of ~15–20%, whereas DMSLA-modified devices showed a smaller loss of ~10%. This enhanced stability is likely due to the ability of DMSLA to reinforce mechanical adhesion at the perovskite/SAM interface and suppress structural degradation caused by repeated thermal expansion and contraction. The transient decrease and subsequent recovery in $V_{OC}$ observed for DMSLA-treated devices may arise from interfacial relaxation during repeated thermal cycling, reflecting temporary perturbation of interfacial energetics followed by stabilization upon continued cycling. Further studies would be required to determine the exact mechanism. We note that the adhesion ranking (DMSLA > DHLA > LA > control) does not directly mirror the thermal-cycling results because adhesion represents only one component of thermomechanical stability. Thermal cycling also depends on the chemical robustness of the linker molecules and the ability of the grain boundaries to accommodate strain. DMSLA combines strong and stable interfacial bonding—supported by our DFT, XPS, CV, and UPS data—with higher chemical stability under thermal stress, whereas DHLA's thiol groups are more reactive and may undergo changes during cycling. This interplay of interfacial adhesion, chemical stability, and grain-boundary reinforcement explains the superior cycling performance of DMSLA. Furthermore, the *J-V* characteristics of representative devices from each group, measured before and after the thermal fatigue test, are shown in Fig. 4b. To further elucidate the origin of performance degradation under thermal cycling, the evolution of all photovoltaic parameters (PCE, FF, $V_{OC}$, and $J_{SC}$) was analyzed across different cycling conditions in detail and is summarized in Supplementary Figs. 27–29. A consolidated comparison of the parameter changes is provided in Supplementary Fig. 30, which shows that the dominant degradation pathway is associated with a loss in fill factor, while variations in $V_{OC}$ and $J_{SC}$ remain comparatively minor. The contour plot shown in Fig. 4c reveal that degradation trends correlate more strongly with the accumulated thermal-exposure time than with

the number of cycles alone. Together, these results indicate that thermomechanical fatigue primarily manifests through resistive and interfacial losses rather than bulk recombination or optical absorption losses. Additional *J-V* measurements under ~1,360 W m$^{-2}$ (AM0-approximated) illumination further confirm the robustness of the DMSLA-modified devices under elevated operational stress (Fig. 4d). Statistical distribution and additional information are provided in Supplementary Figs. 31 and 32. Maximum power point tracking (MPPT) analysis of the encapsulated solar cells—conducted under conditions of 45 °C ± 5 °C, 70 ± 10% relative humidity, and ~1,000 W m$^{-2}$ equivalent 1-sun illumination- revealed that DMSLA-based target devices remained almost unchanged, while the control samples lost 33% of their initial efficiency after 500 h (Fig. 4e). The high PCE retention observed during MPPT measurements, compared to the lower retention seen in thermal fatigue tests, indicates that extreme temperature cycling may present a more significant challenge than continuous MPPT operation. Nonetheless, maintaining operational stability at elevated temperatures remains equally important and should not be underestimated. The initial *J-V* curves and corresponding photovoltaic parameters prior to MPPT testing are provided in Supplementary Fig. 33.

## Discussion

In this work, we introduced a dual molecular reinforcement strategy to mitigate thermomechanical degradation in PSCs subjected to repeated extreme temperature cycling. This extreme cycling induces severe stress at the contact interfaces, particularly due to the much higher CTE of perovskites relative to substrate materials. In this two-step strategy, grain-to-grain cohesion was enhanced by incorporating α-lipoic acid (LA) into the perovskite precursor, enabling in situ polymerization during thermal processing, while interfacial adhesion between the perovskite layer and the underlying substrate was strengthened through chemical modification of the disulfide ring to a sulfonium group ($-S^+(CH_3)_2$), a methylated cationic moiety. Ultimately, the functional groups responsible for these effects—closed-ring sulfur, thiol, and sulfonium salt—were shown to form strong, direct interactions with the perovskite crystal lattice. Their combined contribution acts like a molecular suspension system, critically supporting structural integrity during thermal cycling.

While molecular tailoring strategies have been reported in the context of PSCs and flexible devices, our work uniquely combines grain-boundary reinforcement and interface stabilization to address thermal fatigue over a wide temperature range. The observed improvements in performance retention under repeated cycling highlight the importance of targeting mechanically vulnerable regions within multilayer perovskite device stacks, rather than focusing solely on optoelectronic optimization.

Looking forward, further improvements in thermal fatigue resistance may be achieved through the rational design of multifunctional molecular additives capable of controlled cross-linking across both grain boundaries and interfaces. Extending thermal cycling protocols to higher cycle numbers using automated testing platforms will be essential for establishing long-term degradation trends and for correlating accelerated stress tests with operational lifetime. More broadly, the concepts demonstrated here provide a general framework for improving the durability of perovskite photovoltaics operating under severe temperature cycling conditions.

## Methods

### Materials

Cesium iodide (CsI, 99.5%), Lead (II) iodide (PbI2, 99.99%), Lead(II) Chloride (PbCl2, 99.0%), α-lipoic acid (LA), and dihydrolipoic acid (DHLA) were sourced from TCI. N, N-dimethylformamide (DMF, 99.8%), dimethyl sulfoxide (DMSO, 99.8%), Isopropanol (IPA, 99.8%), Chlorobenzene (CB, 99.9%), and ethanol (EtOH, 99.8%) were purchased from Sigma Aldrich. Methylammonium iodide (MAI, 99.5%), Methylammonium bromide (MABr, 99.5%), and formamidinium iodide (FAI, 99.5%) were purchased from GreatCell Solar Materials. Fullerene (C60, 99.5%) and (4-(7H-dibenzo[c,g]carbazol-7-yl)butyl)phosphonic acid (4PADCB) were purchased from Lumtec. Indium tin oxide (ITO, YXKJGI-0006,15 Ω sq$^{-1}$) and Bathocuproine (BCP) were purchased from Yingkou Advanced Election Technology Co., Ltd. Lithium fluoride (LiF) and Ethanediamine dihydroiodide (EDAI2) were purchased from Xi'an Polymer Light Technology, China. All chemicals were used as it is without further purification.

### Synthesis of DMSLA

DMSLA was synthesized via a two-step methylation procedure (Supplementary Fig. 1). In the first step, the dithiol carboxylic acid precursor was reacted in a biphasic toluene/water system in the presence of aqueous NaOH and tetrabutylammonium iodide as a phase-transfer catalyst. Iodomethane was added dropwise, and the reaction mixture was stirred at room temperature for 12 h. The reaction was then acidified to pH 1 using 2 M HCl, followed by the addition of brine. The product was extracted with dichloromethane, and the combined organic layers were dried over anhydrous MgSO4. Solvent removal under reduced pressure afforded the methylthio-functionalized intermediate as a yellow solid without further purification.

In the second step, the methylthio-functionalized intermediate was dissolved in ethanol, and iodomethane was added dropwise at room temperature. The reaction mixture was stirred for 24 h, after which the solvent was removed under reduced pressure. Addition of diethyl ether induced precipitation of a yellow solid, which was collected by filtration and washed three times with diethyl ether to yield the final product, 1,3-bis(dimethylsulfonio)heptane-7-carboxylic acid diiodide (DMSLA).

Full synthetic schemes and spectroscopic characterization (^1H and ^13 C NMR) are provided in the Supplementary Information (Supplementary Fig. 1–4).

### Single junction device fabrication LMU Munich

ITO-coated glass substrates were sequentially cleaned with acetone and isopropyl alcohol (IPA) in an ultrasonic bath for 15 min each, then dried under a stream of nitrogen gas. They were subsequently treated with oxygen plasma for 15 min before the coating process. Following this, the substrates were transferred into a nitrogen-filled glove box. A solution of 4PADCB (0.5 mg mL$^{-1}$), prepared in methanol and blended solutions containing LA, DHLA, or DMSLA (in a 4:1 weight ratio), was spin-coated onto the ITO surface at 3,000 rpm for 30 seconds. The films were then annealed on a hot plate at 100 °C for 10 min. The perovskite precursor solution was prepared in 1 mL of a DMF/DMSO (4:1, v/v) mixture, containing 1.7 M Cs$_{0.05}$MA$_{0.10}$FA$_{0.85}$PbI$_3$ and doped with LA (0.2 mg mL$^{-1}$). This solution was deposited onto the modified ITO substrates by spin coating at 5000 rpm for 45 s. Anisole (250 μL) was used as an antisolvent, dropped onto the spinning substrate at the 15th second. The resulting films were immediately annealed at 100 °C for 30 min. Notably, no surface passivation has been applied to perovskite layers. Subsequently, C$_{60}$ (28 nm), BCP (7 nm) (or 20 nm SnO$_2$ by ALD followed by 70 nm IZO by sputtering for thermal-shock test), and finally a metallic silver electrode (120 nm) was thermally evaporated in an MBraun thermal evaporator with 1 Å s$^{-1}$ deposition rate through a shadow mask.

### Single junction device fabrication Tianjin University

The 1.6 M perovskite precursor solution with the composition of FA$_{0.8}$MA$_{0.15}$Cs$_{0.05}$PbI$_3$ (bandgap: 1.55 eV) was prepared by fully dissolving CsI: 20.8 mg, MABr: 7.6 mg, MAI: 38.1 mg, FAI: 234 mg, PbI$_2$: 738 mg, PbCl$_2$:18 mg in a mixed solvent of DMF: 800 μL, DMSO:200 μL. The 4PADCB solution with a concentration of 0.5 mg mL$^{-1}$ was prepared by dissolving it in EtOH. The EDAI$_2$ solution with a concentration

of 0.5 mg mL$^{-1}$ was prepared by dissolving it in IPA. Inverted device architecture was ITO/SAMs/PVK/LiF/C$_{60}$/BCP/Ag. The ITO was washed with detergent, deionized water, acetone, and ethanol in sequence for 30 minutes. After that, the cleaned ITO substrate was dried by N$_2$ gas and then treated with plasma for 15 minutes. The 100 μL 4PADCB (0.5 mg mL$^{-1}$) dissolved in ethanol, as a hole transport material, was spin-coated on ITO for 30 s at 3000 rpm and then annealed at 100 °C for 10 min. Subsequently, 60 μL perovskite precursor was deposited on the 4PADCB layer. The perovskite solution was spin-coated at 1000 rpm for 10 s, and 5000 rpm for 30 s. At 16 s before the end of the procedure, 200 μL of CB as the antisolvent was dripped into the pre-cast film surface. After that, the substrates were quickly transferred to a hot plate at 100 °C for 50 min annealing. For post-treatment, the 55 μl EDAI$_2$ dissolved in IPA was spin-coated on the perovskite surface at 5000 rpm for 30 s and then annealed at 100 °C for 10 min. The spin-coating processes were all conducted at room temperature (about 25 °C) in a N$_2$-filled glovebox. Finally, 1 nm LiF at a rate of 0.1 Å s$^{-1}$, 25 nm C$_{60}$ at a rate of 0.1 Å s$^{-1}$, 6 nm BCP at a rate of 0.1 Å s$^{-1}$, and 100 nm silver electrode at a rate of 1.0 Å s$^{-1}$ were thermally evaporated, respectively, under high vacuum ( < 4 × 10$^{-4}$ Torr).

## Solar cell device encapsulation

The device is covered with two barrier films (polyethylene terephthalate), using Polyolefin Elastomer (POE) as the top and bottom encapsulation material and butyl rubber as the edge sealant. Under a temperature of 110 °C and a pressure of 1000 mbar, the assembly is pressed for 20 min to soften the edge sealant and cure the encapsulation material.

## Solar cell performance analysis

Current density–voltage ($J$–$V$) measurements were performed using a Newport Oriel Sol 2 A solar simulator, coupled with a Keithley 2401 source meter. Devices were illuminated through a shadow mask, defining an active area of 0.1 cm$^2$. $J$–$V$ curves were recorded under standard AM 1.5 G illumination from a Xenon arc lamp, with the light intensity calibrated to 100 mW cm$^{-2}$ using a Fraunhofer ISE-certified silicon reference cell, with a stabilization time of 5 s and a voltage step size of 0.02 V, and the measurements were performed in ambient atmosphere at room temperature. Additionally, at LMU, a spectro-photometer was used to measure the spectrum of the solar simulator, and the illumination intensity was directly calibrated to minimize spectral mismatch by using the provided EQE data together with the NREL AM 1.5 G reference spectrum. EQE curves were acquired using a solar cell spectral response measurement system (QE-R, Enli Technology Co. Ltd).

## NMR

$^1$H NMR and $^{13}$C NMR spectra were recorded using a Bruker AVANCE III HD 400 MHz spectrometer. Chemical shifts were calibrated using the solvent signal DMSO-d$_6$ as an internal reference.

## CV measurements

CV measurements were performed using a CHI660 electrochemical workstation (CH Instruments). An Ag/AgCl (3 M NaCl) reference electrode and a Pt-wire counter electrode were used. The CV experiments were performed in a solution containing 1,2-dichlorobenzene (o-DCB) with 0.1 M tetrabutylammonium hexafluorophosphate (TBA$^+$PF$_6^-$). The geometric area of the ITO electrode exposed to the solution for electrochemical measurement was 1.245 cm$^2$, and the CV data were measured at scan rates of 100 mV s$^{-1}$, 200 mV s$^{-1}$, 300 mV s$^{-1}$, and 400 mV s$^{-1}$. Samples for the C-V measurements were prepared by depositing 4PADCB or a mixture of 4PADCB:LA/DHLA/DMSLA (in a molar ratio of 2:1) in ethanol solution was uniformly spread on ITO and allowed to rest for 10 s, followed by spinning of the films at 3000 rpm for 30 s. The films were then annealed at 100 °C for 10 min.

## UPS and XPS analysis

X-ray photoelectron spectroscopy (XPS) was performed on a Thermo Scientific K-Alpha with an Al-Kα X-ray source. Ultraviolet photoemission spectroscopy (UPS) was performed using a Kratos Analytical ESCALAB-250Xi photoelectron spectrometer, with He(I) excitation at 21.22 eV.

## UV Vis absorption spectroscopy

UV–Vis optical measurements were carried out using a PerkinElmer Lambda 1050 spectrometer equipped with an integrating sphere. Both transmittance ($T$) and reflectance ($R$) spectra were recorded on thin-film samples deposited on glass substrates. The absorptance ($A$) of the films was then calculated according to the relation $A = 1 - T - R$, which inherently corrects for baseline offsets by accounting for substrate transmission and surface reflection losses. The resulting spectra were used to construct Tauc plots for bandgap estimation, as shown in Supplementary Fig. 22a, b. All films exhibited similar absorption characteristics with no major spectral shifts, although a slight red shift was observed for the DMSLA-treated sample. The estimated optical bandgaps for all conditions were approximately 1.55 eV, confirming negligible variation among the samples (Supplementary Fig. 22b).

## Time-resolved PL spectroscopy

Time-resolved PL spectra of the perovskite films fabricated on different SAM-modified ITO substrates were studied using a Picoquant FluoTime 300 spectrofluorometer with an excitation wavelength of 375 nm.

## Scanning electron microscopy

The morphology of perovskite films on textured Si with different self-assembled monolayers (SAMs) was analyzed using an in-house FEI Helios Nanolab G3 UC DualBeam scanning electron microscope (SEM). Cross-sectional and top-view images were acquired at 2 kV and 3 kV, respectively, to minimize beam-induced damage to the perovskite film while ensuring adequate resolution and contrast, using the TLD detector. Samples, prepared from the active area, were mounted on silver paste, and no additional conductive coating was applied.

## GIWAXS and XRD

GIWAXS measurements were carried out on an Anton-Paar SAXSpoint 2.0 with a Primux 100 microfocus source with Cu-K$_{α1}$ radiation (λ = 1.5406 Å) and a Dectris Eiger R 1 M 2D Detector.XRD.

XRD measurements were carried out on a Bruker D8 Discover diffractometer in Bragg–Brentano geometry, using Ni-filtered Cu-K$_{α1}$ radiation (λ = 1.5406 Å) and an apposition-sensitive LynxEye detector.

## AFM

AFM measurements were performed using a Bruker Dimension Icon in PeakForce Tapping mode, which enables the simultaneous capture of mechanical properties and topography. The used cantilevers (RTESPA-300) were initially characterized on calibration samples (PFQNM-SMPKIT-12M) and by a thermal tune to determine the deflection sensitivity, spring constant, and tip radius. All scans were performed in a nitrogen atmosphere to reduce the degradation of the perovskite. For the scans, we used a scan rate of 0.8 Hz, 512 samples/line, and a peak force of 20 nN for the top side and 5 nN for the peeled bottom side of the perovskite. Data analysis was performed with the open-source software Gwyddion.

## Pull off tests

In this procedure, the sample was prepared following the same steps as in the device fabrication process up to the deposition of the perovskite layer. The perovskite surface was first coated with a thin layer of PMMA to protect the underlying perovskite layer from moisture and to prevent direct contact with the epoxy adhesive used in subsequent preparation steps. A 10 wt% PMMA solution in chlorobenzene was spin-

coated onto the prepared perovskite film surfaces at 2000 rpm for 30 s, followed by drying at room temperature for 24 h. A 10 mm-diameter circular steel dolly was then bonded onto the PMMA-coated surface using epoxy adhesive (Huntsman Araldite 420). The assembly was cured at room temperature for 1 hour, followed by an additional curing step at 60 °C for 4 h to ensure full polymerization of the adhesive. To ensure consistency across all tests, the volume of epoxy applied was carefully controlled so that the bondline area remained consistent. After curing, a steel wire was used to connect the dolly to the upper grip of the universal testing machine. All tests were performed using an Instron 5944 Single Column Electro-Mechanical Testing Machine equipped with a 2 kN load cell, under a constant displacement rate of 0.1 mm/min(quasi-static). The interfacial tensile strength was calculated by dividing the maximum measured pull-off force by the bonded area.

### Thermal cycling tests

Thermal cycling experiments were performed using a custom-built closed-lid setup designed to achieve reproducible and well-controlled temperature cycling between −80 °C and +80 °C. To prevent frost formation and maintain dry ambient conditions, dry air was circulated within the sealed chamber, and cycling was initiated only after the integrated DHT22 temperature–humidity sensor indicated 0% relative humidity. The sample temperature was monitored directly from the device surface using a K-type thermocouple placed in contact with the substrate to accurately capture the thermal load experienced by the cells. A schematic representation of the thermal fatigue experimental setup is provided in Supplementary Fig. 26.

### Computational methodology

We carried out first-principles calculations based on DFT using a plane-wave basis set in conjunction with the projector augmented-wave (PAW) method. To describe exchange–correlation interactions, we employed the PBEsol functional, a revised version of the generalized gradient approximation developed by Perdew, Burke, and Ernzerhof[36]. Long-range dispersion interactions were accounted for using the DFT-D3 correction proposed by Grimme[37]. Initial structures were constructed using experimentally reported lattice parameters of ITO and FAPbI$_3$. All self-consistent field and structural optimization calculations were performed using a plane-wave energy cutoff of 400 eV and a k-point mesh with a spacing of $0.03 \times 2\pi/\text{Å}$. Both atomic positions and lattice parameters were relaxed using the conjugate gradient algorithm until the maximum residual force on each atom was less than 0.02 eV/Å. Unless otherwise specified, all simulations were conducted using the VASP software package[38,39] following this protocol.

### Reporting summary

Further information on research design is available in the Nature Portfolio Reporting Summary linked to this article.

## Data availability

The data generated in this study are provided in the Supplementary Information/Source Data file. All other data are available from the corresponding authors on request. Source data are provided with this paper.

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

## Acknowledgements

This work is funded by the European Research Council (ERC) under the European Union's Horizon Europe Research and Innovation Program (INPERSPACE, Grant Agreement No. 101077006 to E.A.). The authors thank Dr. Steffen Schmidt for his assistance with the SEM measurements and for preparing the samples for TEM analysis. We also thank Dr. Markus Döblinger for conducting the TEM measurements. We acknowledge Hongqiang Guo for the synthesis and characterization of DMSLA in Tianjin. We further thank Tianjin Meitong Intelligent Technology Co., Ltd. for providing the LED Solar Simulator (MT-LED-110) used for AM0 measurements.

## Author contributions

A.B. conceived the idea, A.B., C.Y., and Y.S. planned and carried out the experiments. C.Y. built the thermal cycling test setup and assisted with the tests. Y.S. conducted device fabrication, analysis, and carried out UPS, XPS, and coverage factor analysis. S.L. and J.E. performed and analyzed nanoscale mechanical measurements. X.L. and G.L. conducted and analyzed pull-off tests. J.H. analyzed the UPS data. H.Z. contributed to the optimization of the solar cells. R.H. carried out GIWAXS measurements and assisted with the synthesis. O.F. and M.S. executed PL and EL measurements on the devices and evaluated the results. C.D. and I.Y. conducted computational simulations and analyzed the outcomes. E.U. contributed to the analysis of optoelectronic measurements. F.Z. contributed to the analysis of device performance and related measurements and supervised the experiments in Tianjin. E.A. supervised the work and secured funding. All authors contributed to writing.

## Funding

## Competing interests

The authors declare no competing interests.
