## [Transparent Peer Review file · Nature Communications]

Perovskite Solar Cells with Enhanced Thermal Fatigue Resistance under Extreme Temperature Cycling

Corresponding Author: Dr Erkan Aydin

Version 0:

Reviewer comments:

Reviewer #1

(Remarks to the Author)

The authors present a thorough investigation on the impact of a self-assembled monolayer using a lipoic acid and derivatives. There is rich investigation in the SAM and its functions at the grain boundaries of the perovskite - the methodology is sound. There is suitable information for the work to be reproduced. It is a concern of this reviewer that this manuscript makes broad claims of suitability for space with only 1-day low Earth orbit equivalent thermal cycling and no analysis of the devices in any other space relevant environments.

1. Line 85 sentence beginning "Across all these systems..." should cite where these claims come from, I believe they are from ref 17 and 18 but would be good to re-cite.
2. The claims of this SAM being suitable for implementation in space are over inflated. These devices show moderate enhancement to only one stressor at 24 hrs. Authors should temper these claims by mentioning environmental caveats that could prove to be detrimental (vacuum, outgassing, UV degradation, ionizing radiation, etc.) or demonstrating experimentally or computationally that the SAM will exhibit longevity beyond 24 hours and that it can endure other stressors present in the space environment. The authors would likely agree that a material enduring 24 hours does not mean it will exhibit more durability than the control material that undergoes marginally more degradation (~5% absolute enhancement on retained performance. How can the authors demonstrate or suggest that the SAM would continue to outperform the control beyond 24 hours of thermal cycling and considering other relevant space environmental stressors?
3. JV measurements and MPPT were conducted at AM 1.5, but longevity claims are made for operation in space. Authors should comment on expected observations or variations at AM0 given they're framing this work for space applications. The solar illumination should be added in the body of the text because readers will assume AM0.
4. The paragraph beginning at Line 72 with "During thermal cycling..." should site other perovskite investigations into impact of thermal cycling including Krause et al <https://doi.org/10.1002/solr.202300468>
5. The claim of the first systematic study of extreme low temperature cycling experiments is invalid and should be edited or removed (conclusion line 377) to acknowledge the similar systematic work published by Chen et al <https://doi.org/10.1021/acseenergylett.4c00988> and Li et al <https://doi.org/10.1002/aenm.202202887> The first extreme low temperature cycling experiments were probably conducted by Delmas et al who did thermal cycling analysis on the International Space Station <https://doi.org/10.1002/aenm.202203920>. Authors should read these works and consider citing all the above.
6. Supplemental information line 227 in "pull-off tests" section second sentence – which surface was first coated with a thin layer of PMMA? Please confirm in text.

Reviewer #2

(Remarks to the Author)

While the manuscript presents a detailed technical investigation into the adhesion properties between the perovskite and

HTL (SAM) in the presence of LA and its derivatives, as well as the full device performance variation under thermal cycling test, I do not find the novelty sufficiently strong for acceptance in Nature Communications. Poly-LA has already been reported (ref. 17) to serve as a passivating agent that simultaneously optimises interfacial contact, enhances perovskite crystallinity, and maintains robust mechanical properties. In this manuscript, the authors report that LA and its derivatives also form polymers after annealing around the grain boundaries, which is conceptually very similar to the previously published work.

The thermal cycling investigation was conducted between $-80\text{ }^{\circ}\text{C}$ and $+80\text{ }^{\circ}\text{C}$, and the results show a positive effect after incorporating DMSLA into the HTL. However, the title suggests that the work was carried out under extreme LEO (Low Earth Orbit) conditions, which is somewhat misleading. Thermal vacuum cycling tests should be included if the authors wish to retain the current title. In the conclusion, the authors explicitly mention that thermal vacuum testing will be the next step. Additionally, the manuscript contains numerous vague descriptions, making it difficult to follow. To help the authors improve the clarity and quality of the paper, the following points should be addressed before it can be considered for publication in another journal.

1. One of the key advantages of perovskite materials for space applications is their high specific power. This should be emphasized in the introduction (lines 48–49).
2. Line 51: If I understand correctly, the authors state that III–V solar cells exhibit a short operational lifetime. However, III–V solar cells are widely used in current space applications. It would be helpful to clarify which type of commercial solar cells the authors are comparing with—are they referring to Si-based solar cells?
3. Lines 91–92: There is some confusion regarding the substrate and perovskite description. Does the term “substrate” here refer to TCE without any charge transport layer deposited? In the following sentences (lines 92–94), were all the mentioned methods performed on ITO substrates or on non-TCE substrates? This section should be rewritten for better clarity.
4. Lines 103–108: The ring-opening of LA is mentioned in the text, but the molecular structure of LA is not shown in this part of the section, making it difficult for readers to follow until reading Fig. 2a. It is suggested to either include the LA structure in the Supporting Information (around Figs. S1–S3) or refer to Fig. 2a directly in the text.
5. Please add “(DHLA)” after “ ^1H NMR of 6,8-bis(methylthio)octanoic acid” in Fig. S2 caption, consistent with Fig. S3.
6. It would be helpful to label the atom numbers in the chemical structures and indicate the corresponding numbers in the ^1H NMR spectra shown in Figs. S1 and S2.
7. It is suggested to introduce the full chemical name of LA earlier in the text, before its first appearance in Section 2.2 (around line 175).
8. Please verify the schematic structure of DMSLA in Fig. 2a; it appears incorrect. There seems to be an extra carbon atom attached to one of the sulphur atoms.
9. Line 192: The meaning of this sentence is unclear. Does the weak adhesion observed for LA and DHLA samples result from imperfections in sample preparation?
10. In the caption of Fig. S10, please clarify how the authors ensured that LA molecules remained confined within the grain boundaries and were not exposed on the surface.
11. In Table S1, the data for the perovskite with LA are missing.
12. Line 297: Please verify whether the energy offset is 0.02 eV rather than 0.04 eV.
13. How was the bandgap calculated? If it was derived from UV–vis spectra, please provide the corresponding absorption spectra with a proper baseline.
14. Page 12, line 328: Please explain how the device temperature was maintained at $-80\text{ }^{\circ}\text{C}$ when immersed in liquid N_2 .
15. The control devices retained 79% of their initial efficiency after 16 cycles, DMSLA showed 84%, and DHLA showed 72%. These differences appear minor (within $\pm 5\%$ deviation). How many devices were measured to draw this conclusion?
16. Although the variation is small, why does the VOC curve in Fig. 4a for DMSLA show a decrease after five cycles, followed by a gradual increase after 16 cycles?
17. In Materials section in SI, BCP and LiF were mentioned twice and also sourced from different suppliers. Please indicate the reason.
18. Dyesol company no longer exists. Please confirm FA and MA were sourced from GreatCell Solar.
19. Please provide more details in the caption of Fig. S24. Do the dark blue and light blue boxes correspond to the reverse and forward scans, respectively? The same clarification is suggested for the colours used in Figs. S24 and S25.

Reviewer #3

(Remarks to the Author)
Report on NCOMMS-24-15009

Title: Perovskite Solar Cells with Enhanced Thermal Fatigue Resistance in Low-Earth Orbit Space Extremes

by Ali Buyruk et al.

The manuscript addresses a crucial aspect for the employment of perovskite solar cells in Low Earth Orbit (LEO): their resilience to thermal fatigue. The authors propose an innovative two-step approach, based on the use of α -lipoic acid and its derivatives, to improve perovskite film cohesion and mitigate delamination. To validate this strategy, they employed several experimental techniques such as XRD, FTIR, AFM, SEM, GIWAXS, which confirmed a significant improvement in both internal grain cohesion and interfacial adhesion. Thermal fatigue tests, performed with a custom-built setup and a thermal cycling protocol ranging from $-80\text{ }^{\circ}\text{C}$ to $+80\text{ }^{\circ}\text{C}$ for 16 cycles (equivalent to one day of operation), indicate encouraging stability. The best performing device achieved an efficiency of 26% and retained 84% of its initial performance after the thermal fatigue test.

The manuscript is well written, logically structured, and contextualized within the literature. The experimental work is carefully executed, and the short-term data are convincing within the tested scope. The analytical framework is solid, and the interpretation of the available data is careful.

However, while the proposed strategy is novel and the results are encouraging, in our opinion, the current experimental validation does not provide sufficient evidence to substantiate claims of aerospace suitability for the following reasons:

- 1) The thermal cycling test (16 cycles) is far too limited to represent orbital conditions, which typically involve hundreds or thousands of cycles.
- 2) The working conditions of solar cells in space are extremely harsh, involving not only thermal fluctuations but also high vacuum, atomic oxygen, and high-energy particle radiation (such as protons, electrons, neutrons, γ -rays). Although the authors briefly acknowledge the need for vacuum tests to better simulate the space environment, they do not consider the importance of radiation, leaving a fundamental gap in assessing material's long-term stability. Without such tests, the applicability of the solar cells for aerospace missions remains speculative.
- 3) The proposed additive engineering approach, based on organic additives (lipoic acid and its derivatives), is innovative and could have relevance for improving the durability of perovskite solar cells more broadly. However, the radiation stability of these organic molecules is a fundamental issue that should be addressed, either experimentally or via literature-supported discussion.

The lack of comprehensive environmental validation severely limits the impact of the findings for aerospace applications. In its present form, the study is best considered as a proof-of-concept for improved perovskite stability, not as evidence of readiness for space deployment.

Suggested improvements

- Increase the number of thermal cycles to a more representative level (hundreds or thousands).
- Include tests under vacuum condition and radiation exposure (protons, electrons, γ -rays).
- Provide a discussion of the radiation stability of α -lipoic acid derivatives, supported by prior studies.

Overall assessment

This study introduces a promising approach to improving perovskite solar cell stability and provides convincing short-term laboratory results. However, in its current form, the manuscript does not provide sufficient experimental evidence to demonstrate suitability for aerospace applications. The lack of extended thermal cycling, vacuum testing, and radiation stability assessment constitutes a major gap. Therefore, we do not find the work suitable for publication in Nature Communications at this stage. Substantial additional testing and contextualization would be required before reconsideration.

Reviewer #4

(Remarks to the Author)

This manuscript presents an interesting study on the use of additives to enhance the thermal fatigue resistance of perovskite solar cells for space applications. The work is novel and offers valuable insights, especially since thermal fatigue has received comparatively little attention in this field.

I recommend a revision before the manuscript can be accepted for publication. My detailed comments are as follows:

1. Since the context is space applications, why was AM0 not used for device measurements? AM0 is more relevant, and its higher UV component could also influence additive stability. Please provide AM0 measurements and comparisons, as this seems necessary.
2. In the adhesion test, the ranking is DMSLA > DHLA > LA > control. However, this ranking does not align with the thermal cycling results. Does this suggest that adhesion is not the main factor contributing to thermal cycling failure? Please explain and discuss possible alternative failure modes with supporting evidence.
3. For the thermal cycling tests performed using the in-house equipment, was moisture controlled? The encapsulation method described in the paper may not effectively block moisture, and heating/cooling cycles could allow moisture ingress, which would affect device performance. This condition would not reflect the space environment, so clarification is needed.
4. The EQE results indicate that the front interface (light-incidence side) does not improve, and there is a noticeable loss at the rear interface when additives are used. Please explain this observation. The evidence for interfacial passivation is not very strong from EQE results.
5. For the MPPT test, please provide the corresponding IV curves and the initial PCE data.

Reviewer #5

(Remarks to the Author)

Version 2:

Reviewer comments:

Reviewer #1

(Remarks to the Author)

Reviewer #2

(Remarks to the Author)

Several reviewers raised similar questions regarding the inclusion of additional information related to space applications in response to the originally submitted manuscript. Although this was not the authors' original intention, as explained in their responses to multiple reviewers, the initial manuscript (title) was somewhat misleading. This issue has now been recognised and corrected by the authors. The revised and rewritten abstract, introduction, and conclusion show significant improvement and now clearly focus on the work being addressed.

Based on the through revision, it is recommended to be accepted for publish in this journal.

Reviewer #3

(Remarks to the Author)

Report on NCOMMS-25-67465B

Title: Perovskite Solar Cells with Enhanced Thermal Fatigue Resistance under Extreme Temperature Cycling

by Cem Yilmaz et al.

In the revised version of the manuscript, the authors have effectively restructured the abstract, introduction, and conclusions, removing inappropriate references to "space readiness" and clearly defining the study's aim as an investigation of thermal fatigue mechanisms in perovskite solar cells subjected to extreme temperature cycling. This reformulation makes the manuscript more coherent and scientifically robust.

In light of this repositioning, the concerns raised in the first review regarding the absence of vacuum tests, ionizing radiation, or other space-relevant environmental factors are no longer critical. The authors' decision not to include such tests is now fully justified, as the work is correctly framed as a proof-of-concept on the mechanical and thermal stability of perovskites, without the ambition of demonstrating suitability for space missions.

We also appreciate the authors' detailed explanation of the limited number of thermal cycles. Technical and safety considerations related to the use of liquid nitrogen, the need for authorized personnel to refill the tanks, the long duration of the measurements, and the manual nature of the experimental setup make this methodological choice fully understandable. It is also reasonable, as the authors did, to stop the tests once the device performance drops below 90% of the initial PCE.

Although the protocol remains limited to 16 cycles, preventing definitive confirmation of the statement that "most degradation occurs primarily in the first cycles," the inclusion of supplementary experiments with shorter cycles (2 and 15 minutes) is valuable. These experiments allow the effect of the number of cycles to be distinguished from those of total thermal exposure time. The new data, including contour plots, boxplot distributions, and a comparative table, convincingly demonstrate that degradation is primarily governed by the cumulative thermal exposure and the associated thermomechanical stress, while the number of cycles plays a secondary role.

We also note that the manuscript could benefit from a brief clarification in the main text regarding the number of devices tested and the number of pixels per device. While this information is already provided in the Supplementary Information, a concise inclusion in the main text would improve methodological transparency and facilitate an immediate assessment of the statistical robustness of the results. This is a formal refinement that would make the manuscript more accessible without affecting its scientific integrity.

In summary, the revisions significantly improve the manuscript compared to the first version. The main conceptual concerns have been effectively addressed, and the authors' responses are satisfactory. Overall, the work is now more balanced, rigorous, and appropriately focused on its scientific objectives. We therefore consider the manuscript, in its current form, suitable for publication in Nature Communications.

Reviewer #4

(Remarks to the Author)

I thank the authors for their substantial efforts in addressing my comments. The manuscript is in a good shape for acceptance.

One minor comment: could the authors justify the standards or guidelines used to define the thermal cycling protocol, including why the ramp rate of 3 is used and dwell duration? Is there a defined pass/fail criterion for the thermal cycling test at specific temperature ranges? Additionally, how do these testing conditions and criteria compare with those reported in other studies or used for space solar cell technologies? it will be helpful to let the read know how the perovksite based cell is a good candidate for space application.

Reviewer #5

(Remarks to the Author)

Response to Reviewers' Comments on NCOMMS-25-67465

We sincerely thank all reviewers for their valuable and constructive comments, which have greatly contributed to improving the quality of our manuscript. Please find our detailed point-by-point responses below.

Reviewer #1:

Remarks to the Author:

The authors present a thorough investigation on the impact of a self-assembled monolayer using a lipoic acid and derivatives. There is rich investigation in the SAM and its functions at the grain boundaries of the perovskite – the methodology is sound. There is suitable information for the work to be reproduced. It is a concern of this reviewer that this manuscript makes broad claims of suitability for space with only 1-day low Earth orbit equivalent thermal cycling and no analysis of the devices in any other space relevant environments.

Response:

We would like to thank the reviewer for taking the time to review our manuscript and for providing valuable feedback that helped us improve the quality and clarity of our work. Also, we thank the reviewer for giving credit to our work. We now understand that our manuscript title and some expression in the abstract may have given a misleading impression—although this was not our intention—regarding the thermal cycling experiments and the tests performed under other space-relevant environments. Below, we address these points step by step.

R1 - Comment 1.

Line 85 sentence beginning “Across all these systems...” should cite where these claims come from, I believe they are from ref 17 and 18 but would be good to re-cite.

R1 - Response 1.

We thank the reviewer for this helpful suggestion. In the revised manuscript, we now explicitly re-cite the sources for the performance recovery claims. The sentence in the Introduction has been modified to:

- Lines 91-93 “Across all these systems, thermal-triggered healing has resulted in more than 80% recovery of solar cell performance after stress testing^[11-18], while systems based on poly(LA) and poly(TA-NI) demonstrated recovery rates exceeding 90%^[17,18]. ”

R1 - Comment 2.

The claims of this SAM being suitable for implementation in space are over inflated. These devices show moderate enhancement to only one stressor at 24 hrs. Authors should temper these claims by mentioning environmental caveats that could prove to be detrimental (vacuum, outgassing, UV degradation, ionizing radiation, etc.) or demonstrating experimentally or computationally that the SAM will exhibit longevity beyond 24 hours and that it can endure other stressors present in the space environment. The authors would likely agree that a material enduring 24 hours does not mean it will exhibit more durability than the control material that undergoes marginally more degradation (~5% absolute enhancement on retained performance. How can the authors demonstrate or suggest that the SAM would continue to outperform the control beyond 24 hours of thermal cycling and considering other relevant space environmental stressors?

R1 - Response 2.

We would like to thank the reviewer for the insightful comments regarding the thermal cycling tests. From the feedback, we understand that our text may have given the impression that we have fully solved all thermal cycling challenges in space, although this was not our intention. We would like to clarify that our tests were designed to approximate realistic stress conditions as closely as possible. However, at this stage, it is naturally difficult to perform experiments that include all relevant stressors simultaneously. This situation is similar to accelerated testing for terrestrial applications (IEC and ISOS Protocols), where, for example, thermal cycling is typically conducted in the dark and within extreme temperature ranges on encapsulated cells (i.e., $-40\text{ }^{\circ}\text{C}$ to $+80\text{ }^{\circ}\text{C}$), rather than all accelerated tests at the same time, such as reverse bias, impact hailing, humidity freeze test, PID tests, etc. In principle, for long-term stability predictions, correlating the individual accelerated tests with actual field tests is required. This situation is the same for space-relevant tests; at some point, performing on-ground and in-orbit tests and their correlation is required (which is a long-term goal of ours). However, since the evaluation of perovskite-based solar cells in space-relevant environments is still at an early stage, the accelerated test protocols are yet to be fully established. Our intention here was not to perform either an accelerated test or a real orbit test. We hope that, as a community, we will reach a consensus on these aspects in the future. Instead, our aim was to perform a one-day equivalent test to evaluate device failure mechanisms under

realistic temperature ranges observed in low-Earth orbit (LEO), acknowledging that these values can vary depending on the specific orbit and spacecraft design.

In light of the reviewer's valuable comment, we have revised our manuscript title from "Perovskite Solar Cells with Enhanced Thermal Fatigue Resistance in Low-Earth Orbit Space Extremes" to "Perovskite Solar Cells with Enhanced Thermal Fatigue Resistance in Low-Earth Orbit **Relevant Temperature Extremes.**"

Furthermore, we have made the following specific revisions to clarify the scope and ensure that the claims remain consistent with the experimental evidence:

In the abstract;

- Line 27: "**Previous works have shown that** PSCs can exhibit high radiation tolerance; however, their stability under the extreme thermal cycling conditions of space—a key factor driving degradation in LEO—remains largely unexplored."
- Line 42: "Dual reinforced devices exhibit 84% performance retention after 16 thermal cycles, representing one full day **in LEO-relevant temperature extremes.**"
- Lines 43–45: "This work highlights the importance of targeted interface engineering and thermal fatigue mitigation strategies for advancing PSCs **toward** space applications."

In the introduction;

- Lines 68–71: "Hence, to **approximate** the mechanical stresses associated with these rapid and extreme temperature shifts, we adopted a custom thermal cycling protocol ranging from -80 °C to +80 °C¹⁰, which necessitated the development of a dedicated test setup. We performed a one-day-equivalent experiment with realistic cooling and heating rates representative of LEO-relevant temperature extremes under dry-air laboratory conditions."

[10] Kim, K., Yang, S., Kim, C. et al. Non-volatile solid-state 4-(N-carbazolyl)pyridine additive for perovskite solar cells with improved thermal and operational stability. *Nat Energy* (2025). <https://doi.org/10.1038/s41560-025-01864-z>

- Lines 95–96: "However, most studies still lack comprehensive thermal fatigue assessments under **LEO-relevant temperature conditions, even in laboratory environments.**"

In the results and discussion;

- Lines 344–347: “The high PCE retention observed during MPPT measurements, compared to the lower retention seen in thermal fatigue tests, indicates that **LEO-relevant temperature cycling** may present a more significant challenge than continuous MPPT operation.”

Conclusion and Outlook

- Deleted sentence (lines 372–374): ~~“Regarding testing setups, implementing vacuum thermal cycling tests could better replicate space-like conditions, highlighting the need for a customized vacuum-compatible tool.”~~
- Deleted sentences (line 376): ~~“To the best of our knowledge, this is the first report to systematically study extreme low-temperature cycling experiments for PSCs.”~~
- Added sentences (starting from line 376): “It should also be noted that our experiments were conducted under dry-air and ambient-pressure conditions; additional space-relevant stressors such as vacuum, ultraviolet radiation, atomic oxygen, outgassing, and ionizing radiation remain important challenges requiring further investigation. In our future works, we plan to include thermo-vacuum cycling tests to extend these findings toward more realistic multi-stressor environments.”
- Revised final sentence: “We believe this work lays the foundation for new research directions and contributes valuable insight into the behavior of perovskite solar cells under **simulated LEO-relevant temperature cycling.**”

R1 - Comment 3.

JV measurements and MPPT were conducted at AM 1.5, but longevity claims are made for operation in space. Authors should comment on expected observations or variations at AM0 given they're framing this work for space applications. The solar illumination should be added in the body of the text because readers will assume AM0.

R1 - Response 3.

We appreciate this valuable feedback. Indeed, this is an important point, as it could potentially mislead readers. We performed the MPPT tests under equivalent AM1.5G illumination conditions, not under AM0 conditions. We note that achieving true AM0 illumination may remain a challenge for the community in the future as well, since generating

a proper blue/UV spectral component for extended measurement durations is difficult at low cost (equipment manufacturer’s feedback). We hope to have these capabilities in the near future. To avoid any confusion for readers, we explicitly stated in the caption of the stability data that it refers to $\sim 1000 \text{ W m}^{-2}$ equivalent 1-sun intensity conditions:

- “Figure 4. Thermal fatigue test of perovskite solar cell. a, Temperature variation over time during the thermal cycling process, and the evolution of device parameters over sixteen cycles, with measurements taken after every five cycles. b, Schematic illustration of the thermal cycling process in LEO. c, J–V curves of solar cell devices before and after the thermal stress test. d, MPPT results of encapsulated perovskite solar cells under $\sim 1000 \text{ W m}^{-2}$ equivalent 1-sun intensity, measured at $45 \pm 5 \text{ }^\circ\text{C}$ and $70 \pm 10\%$ relative humidity.”

In addition, we added clarification in line 351:

- “Maximum power point tracking (MPPT) analysis of the encapsulated solar cells—conducted under conditions of $45 \pm 5 \text{ }^\circ\text{C}$, $70 \pm 10\%$ relative humidity, and $\sim 1000 \text{ W m}^{-2}$ equivalent 1-sun illumination, revealed that DMSLA-based target devices remained almost unchanged, while the control samples lost 33% of their initial efficiency after 500 hours (Figure 4d).”

Additionally, the reviewer raised a valid point regarding reporting device performance under AM0 conditions. Using the same solar simulator, we increased the light intensity to 1.36 sun to approximate AM0 conditions. We have added IV curves and the statistical distribution of the corresponding devices at 1360 W/m^2 . The devices obtained higher J_{sc} and V_{oc} at AM0 compared to those at AM1.5 spectrum.

Supplementary Figure 28. *J-V* curves and the corresponding device parameters of the devices at 1360 W/m² (AM0 approximation).

Supplementary Figure 29. Statistical distribution of a) V_{oc} , b) PCE, c) FF, and (d) J_{sc} values of the devices at 1360 W/m². Each condition had 10 devices.

R1 - Comment 4.

The paragraph beginning at Line 72 with "During thermal cycling..." should cite other perovskite investigations into impact of thermal cycling including Krause et al <https://doi.org/10.1002/solr.202300468>

R1 - Response 4

We thank the reviewer for this valuable suggestion. Additional works related to thermal cycling tests, including the study by Krause et al., have been added to the revised manuscript.

- Lines 78-80: “During thermal cycling, the solar cell stack is exposed to repeated volumetric expansion and contraction, which induces mechanical fatigue, potentially leading to delamination or failure at weakly bonded interfaces.^{11,12”}

[12] Krause, T.S., VanSant, K.T., Lininger, A., Crowley, K., Peshek, T.J. and McMillon-Brown, L. (2023), Thermal Performance of Perovskite-Based Photovoltaics for Operation in Low Earth Orbit. Sol. RRL, 7: 2300468. <https://doi.org/10.1002/solr.202300468>

R1 - Comment 5.

The claim of the first systematic study of extreme low temperature cycling experiments is invalid and should be edited or removed (conclusion line 377) to acknowledge the similar systematic work published by Chen et al <https://doi.org/10.1021/acseenergylett.4c00988> and Li et al <https://doi.org/10.1002/aenm.202202887> The first extreme low temperature cycling experiments were probably conducted by Delmas et al who did thermal cycling analysis on the International Space Station <https://doi.org/10.1002/aenm.202203920>. Authors should read these works and consider citing all the above.

R1 - Response 5

We thank the reviewer for flagging this point. After reviewing these works, the related citations have been revisited and revised accordingly. The corresponding paragraph in the manuscript has been revised, and the claim on the first systematic study of extreme temperature cycling experiments is removed.

- In lines 70-27: “Recent systematic studies have also examined perovskite solar cell performance under extreme low-temperature cycling and in-orbit conditions, further highlighting the importance of understanding thermal fatigue for space deployment.^{7”}

^{9”}

[7] Chen, M. et al. Stress Engineering for Mitigating Thermal Cycling Fatigue in Perovskite Photovoltaics. ACS Energy Lett. 9, 2582–2589 (2024).

[8] Delmas, W. et al. Evaluation of Hybrid Perovskite Prototypes After 10-Month Space Flight on the International Space Station. Advanced Energy Materials 13, 2203920 (2023).

[9] Li, G. et al. Structure and Performance Evolution of Perovskite Solar Cells under Extreme Temperatures. *Advanced Energy Materials* 12, 2202887 (2022).

R1 - Comment 6.

Supplemental information line 227 in “pull-off tests” section second sentence – which surface was first coated with a thin layer of PMMA? Please confirm in text.

R1 - Response 6.

We thank the reviewer for giving opportunity for further clarification. The perovskite surface was first coated with a thin layer of PMMA before dolly attachment to protect the film from moisture and prevent direct contact with the epoxy adhesive. This information has now been explicitly clarified in the Supplementary Information.

- Lines 230–232: “The perovskite surface was first coated with a thin layer of PMMA to protect the underlying perovskite layer from moisture and to prevent direct contact with the epoxy adhesive used in subsequent preparation steps.”

Reviewer #2

Remarks to the Author:

While the manuscript presents a detailed technical investigation into the adhesion properties between the perovskite and HTL (SAM) in the presence of LA and its derivatives, as well as the full device performance variation under thermal cycling test, I do not find the novelty sufficiently strong for acceptance in *Nature Communications*. Poly-LA has already been reported (ref. 17) to serve as a passivating agent that simultaneously optimises interfacial contact, enhances perovskite crystallinity, and maintains robust mechanical properties. In this manuscript, the authors report that LA and its derivatives also form polymers after annealing around the grain boundaries, which is conceptually very similar to the previously published work.

The thermal cycling investigation was conducted between $-80\text{ }^{\circ}\text{C}$ and $+80\text{ }^{\circ}\text{C}$, and the results show a positive effect after incorporating DMSLA into the HTL. However, the title suggests that the work was carried out under extreme LEO (Low Earth Orbit) conditions, which is somewhat misleading. Thermal vacuum cycling tests should be included if the authors wish to retain the current title. In the conclusion, the authors explicitly mention that thermal vacuum testing will be the next step.

Response:

We thank the reviewer for providing valuable feedback on our manuscript and for recognizing the depth of our analysis of the substrate/perovskite interface. We understand the reviewer's concern regarding the novelty of our work, and we appreciate the opportunity to elaborate further. The novelty of our study lies in several key aspects: (i) Molecular innovation: It is true that poly-LA has been reported previously, which is not surprising given that during our material screening, we identified it as a promising candidate due to its ability to undergo ring-opening reactions and form a polymeric network. However, this was not the main focus of our work. Instead, we prescreened this molecular family, using dihydrolipoic acid (DHLLA) and synthesizing a sulfonium-based cation (DMSLA) specifically. Our goal was to achieve stronger interactions with the perovskite lattice—one of the weakest interfaces in these devices. (ii) Process novelty: Another innovative aspect of our study is the incorporation of the linker molecules directly into the SAM solution. This approach simplifies the overall process and enables interface linking in a single step, while simultaneously improving the mechanical adhesion at this critical interface. (iii) Application-relevant thermal cycling: We conducted LEO-relevant thermal cycling experiments, following the same heating and cooling rates experienced by actual devices on CubeSats (using real heating and cooling rates), using space-compatible encapsulated (DOWSIL 93-500) cells.^[R1,R2] While previous studies have made important contributions, most do not replicate the heating and cooling rates corresponding to the targeted applications. We have summarized the literature, including our work, doing extreme thermal cycling either in terms of range or heating/cooling rate in Table R1. In summary, our work differentiates itself from prior studies by combining LEO-relevant thermal cycling with realistic cycle duration (~90 minutes), space-grade encapsulation, and the unique introduction and synthesis of the DMSLA linker at the HTL/perovskite interface.

Regarding the title, we have revised our manuscript title from “Perovskite Solar Cells with Enhanced Thermal Fatigue Resistance in Low-Earth Orbit Space Extremes” to “Perovskite Solar Cells with Enhanced Thermal Fatigue Resistance in Low-Earth Orbit **Relevant Temperature Extremes.**”

We hope that our explanation addresses the reviewer's concerns regarding the novelty of our work. However, if needed, we are happy to provide further clarification.

[R1] Bulut, M. Thermal design, analysis, and testing of the first Turkish 3U communication CubeSat in low earth orbit. *J. Therm. Anal. Calorim.* 143, 4341–4353 (2021).

[R2] Lamb, D. A., Irvine, S. J. C., Baker, M. A., Underwood, C. I. & Mardhani, S. Thin film cadmium telluride solar cells on ultra-thin glass in low earth orbit—3 years of performance data on the AlSat-1N CubeSat mission. *Prog. Photovolt. Res. Appl.* 29, 1000–1007 (2021).

R2 - Comment 1.

One of the key advantages of perovskite materials for space applications is their high specific power. This should be emphasized in the introduction (lines 48–49).

R2 - Response 1.

We thank the reviewer for this valuable suggestion. It has been added into the related part in the introduction.

- Lines 47–49: “Metal halide perovskite solar cells have emerged as a promising option for space photovoltaics, offering a unique combination of high power conversion efficiency (PCE), high specific power, low fabrication costs, and inherent radiation tolerance.^[1]”

R2 - Comment 2.

Line 51: If I understand correctly, the authors state that III–V solar cells exhibit a short operational lifetime. However, III–V solar cells are widely used in current space applications. It would be helpful to clarify which type of commercial solar cells the authors are comparing with—are they referring to Si-based solar cells?

R2 - Response 2.

We agree that our original phrasing could be interpreted as implying that III–V solar cells have short operational lifetimes, which is not accurate. Our intention was to emphasize their high cost and manufacturing complexity, rather than poor reliability.

- Lines 49–51: “They can serve as an alternative to the two main types of commercially used solar cells: conventional III–V-based devices, which, despite their widespread use in space applications, suffer from high cost and relatively short operational

lifetimes,² and wafer-based crystalline silicon cells, which are relatively heavy and exhibit limited radiation tolerance.”

R2 - Comment 3.

Lines 91–92: There is some confusion regarding the substrate and perovskite description. Does the term "substrate" here refer to TCE without any charge transport layer deposited? In the following sentences (lines 92–94), were all the mentioned methods performed on ITO substrates or on non-TCE substrates? This section should be rewritten for better clarity.

R2 - Response 3.

We thank the reviewer for pointing out this potential ambiguity. In the revised manuscript, we clarified that “substrate” refers to bare ITO-coated glass before any charge transport or perovskite layers were deposited. The text was rewritten to eliminate ambiguity.

- Lines 96–102: “To improve adhesion at the interface between the transparent conductive oxide (TCO) and the perovskite layer, several surface-engineering strategies have been developed. Reported approaches include increasing surface hydroxylation by replacing crystalline TCOs with amorphous ones,^[23,24] removing terminal hydroxyls and hydrolysis byproducts via combined HF and UV–ozone treatments,^[25] introducing hetero-chiral linker molecules,^[26] and employing SAMs such as iodine-terminated carbazole derivatives or bifunctional thiol–carboxylic acid systems with varying alkyl chain lengths.^{[27]”}

R2 - Comment 4.

Lines 103–108: The ring-opening of LA is mentioned in the text, but the molecular structure of LA is not shown in this part of the section, making it difficult for readers to follow until reading Fig. 2a. It is suggested to either include the LA structure in the Supporting Information (around Figs. S1–S3) or refer to Fig. 2a directly in the text.

R2 - Response 4.

We thank the reviewer for raising this. In lines 103–108, the lipoic acid (LA) molecule mentioned in the context of the ring-opening reaction refers to its molecular structure, which is depicted in Fig. 2a. To make it easier for readers to follow the discussion on the ring-opening polymerization of LA, we have now added a direct reference to Fig. 2a, where the molecular structure of LA is presented.

- Lines 107–110: “To promote inter-grain connectivity, LA was incorporated into the perovskite precursor solution with the expectation that it could undergo in situ polymerization during crystallization (the molecular structure of LA is shown in Fig. 2a).”

R2 - Comment 5.

Please add "(DHLA)" after "1H NMR of 6,8-bis(methylthio)octanoic acid" in Fig. S2 caption, consistent with Fig. S3.

R2 - Response 5.

We thank the reviewer for this comment. The molecule is referred to as “1H NMR of 6,8-bis(methylthio)octanoic acid” in the caption of Fig. S2 corresponds to the intermediate structure in the synthetic route described under the title “Supplementary Figure 1: Synthetic route for the preparation of DMSLA,” and it is not DHLA. Only the abbreviations for LA and its derivatives, DHLA and DMSLA, have been used; no other abbreviations were introduced.

R2 - Comment 6.

It would be helpful to label the atom numbers in the chemical structures and indicate the corresponding numbers in the 1H NMR spectra shown in Figs. S1 and S2.

R2 - Response 6.

We thank the reviewer for the reminder. We have added the atom numbers in the chemical structures and indicated the corresponding numbers in the 1H-NMR spectra in our revised version.

Supplementary Figure 2: ^1H NMR of 6,8-bis(methylthio)octanoic acid

R2 - Comment 7.

It is suggested to introduce the full chemical name of LA earlier in the text, before its first appearance in Section 2.2 (around line 175).

R2 - Response 7.

We thank the reviewer for this helpful suggestion. As recommended, the full chemical name of LA has been moved from line 183 to line 105.

- Lines 105-108: “In this work, we used α -lipoic acid (LA, 5-(1,2-dithiolan-3-yl)pentanoic acid) and functionalized it further as part of our two-step reinforcement strategy, and the synthetic procedures are provided in **Supplementary Figure 1**, while the NMR analyses are shown in **Supplementary Figures 2–4**.”

R2 - Comment 8.

Please verify the schematic structure of DMSLA in Fig. 2a; it appears incorrect. There seems to be an extra carbon atom attached to one of the sulphur atoms.

R2 - Response 8.

We thank the reviewer for pointing this out. The molecular structure of DMSLA in Fig. 2a has been carefully re-examined, and the incorrect fragment has been corrected. The revised and validated 2D structure has now been included in the manuscript to ensure clarity and accurate representation.

Figure 2. Mechanical analysis of substrate-perovskite interfaces. a, Schematic representation of the HTL contact formed by 4PADCB together with LA, DHLA, and

DMSLA at the ITO/perovskite interface. **b**, Sketch for the test setup used for the pull-off test. **c**, Maximum adhesion strength, and **d**, stress values, which are determined from the highest value in each of the four groups. **e**, The interaction energies between perovskite and the linker molecules, calculated using a computer-based DFT approach. **f**, Interactions between perovskite and the molecules.

R2 - Comment 9.

Line 192: The meaning of this sentence is unclear. Does the weak adhesion observed for LA and DHLA samples result from imperfections in sample preparation?

R2 - Response 9.

We thank the reviewer for this comment. The wide distribution of adhesion strength observed in Figure 2d appears across all conditions, including control and DMSLA samples. This spread originates from the intrinsic sensitivity of the pull-off test to minor preparation variations, such as small-angle misalignments or slight differences in epoxy layer thickness, which can influence the measured stress. To ensure the reliability of our data, we measured two independent batches with five samples for each condition, confirming that the observed variability reflects typical experimental dispersion rather than material-specific issues.

The corresponding paragraph in the manuscript has been revised as follows:

- “We examined the adhesion strength of SAM-linker contacts at the ITO/perovskite interface through pull-off testing. Indeed, as such systems are known to have intrinsically very low toughness, it is important to inhibit any initiation of delamination between the layers. Consequently, the adhesion strength serves as a direct indicator of mechanical integrity. For this, a stack consisting of glass/ITO/4PADCB-linkers/perovskite/PMMA was prepared, and a dolly was attached to the top surface using an adhesive. PMMA served as an interlayer to protect the perovskite from epoxy damage and improve glue adhesion, as shown in **Figure 2b**. From the resulting load–displacement graphs, we extracted the interfacial adhesion strength, which increased from 3.61 MPa for the control to 4.89 MPa for DMSLA-treated samples (Figure 2c). Pull-off tests showed a wide distribution of adhesion strength across all samples, likely arising from small preparation imperfections such as slight misalignment angles or adhesive layer thickness differences, yet DMSLA-based samples consistently showed the highest average

strength. The resulting stress distributions follow the same trend, indicating that DMSLA-treated interfaces require greater force to induce interfacial failure (Figure 2d).”

R2 - Comment 10.

In the caption of Fig. S10, please clarify how the authors ensured that LA molecules remained confined within the grain boundaries and were not exposed on the surface.

R2 - Response 10.

We thank the reviewer for this comment. While PF-QNM primarily probes the near-surface mechanical response, the localized stiffness enhancement and mechanical contrast observed specifically along grain boundaries indicate that LA or its polymerized form remains within these interfacial regions after film formation. The preservation of grain-boundary morphology and the absence of a uniform surface softening further support that LA is not exclusively segregated to the surface. We have slightly revised the caption of Supplementary Fig. S10 to clarify this point. The revised caption now reads:

- “Supplementary Figure 10: Peak Force QNM images from the top of control and target (LA) samples. Red areas in the top contact represent the mask which distinguishes between grains and grain boundaries where the tip apex curvature can be bigger than the grain boundary depth and width which could lead to overestimation of the mechanical properties in addition to the enhanced properties due to the LA. The pronounced mechanical contrast and localized stiffness enhancement along the grain boundaries indicate that LA or its polymerized form likely remains within these interfacial regions after film formation, contributing to improved boundary adhesion. We note that on certain grain facets, a distinct region with lower stiffness is observed, which we attribute to LA molecules bonded to the perovskite being exposed at the surface. In contrast, the brighter areas correspond to unmodified perovskite, with stiffness values matching those of the unprocessed sample.”

To maintain consistency with the revised description and the updated caption of Supplementary Figure S10, we have made minor wording changes in the caption of Figure 1:

- “**Figure 1. Nanoscale mechanical behavior of perovskite film surface and grain boundaries.** a, A sketch demonstrating the thermal behaviour of perovskite polycrystalline structures with (iv) and without poly(LA) (i-iii) at the grain

boundaries. b, Thermal polymerization of LA and its multifunctional role at grain boundaries, facilitating defect passivation (1) and hydrogen bonding interactions (2) between polymer chains. c, A sketch for the working mechanism of the surface-mapping tip-based nano-mechanical analysis technique. Numbers 1 and 2 indicate the perovskite grains and the poly(LA)-rich grain boundaries, respectively. d, Adhesion forces were obtained from the perovskite surface at the HTL contact region using a tip-based nanomechanical surface-mapping technique, alongside topographical imaging. The values shown in the yellow boxes are the average adhesion forces extracted from each corresponding image.”

R2 - Comment 11.

In Table S1, the data for the perovskite with LA are missing.

R2 - Response 11.

Thanks for the reviewer’s reminder. We have added the data for the perovskite with LA in our revised version as follows:

Supplementary Figure 17: a) UPS spectra of a) 4PADCB, b) 4PADCB+LA, c) 4PADCB+DHLA, d) 4PADCB+DMSLA, e) perovskite, and f) perovskite+LA films deposited on ITO substrate, respectively.

Supplementary Table 1: Electronic parameters of SAM (control), its modified forms, perovskite films w/wo LA from the UPS spectra.

Sample	$E_{\text{cutoff}}(\text{eV})$	$E_{\text{onset}}(\text{eV})$	VBM(eV)	Work function(eV)
4PADCB	16.44	0.80	-5.47	-4.76
4PADCB + LA	16.47	0.83	-5.56	-4.73
4PADCB + DHLA	16.48	0.85	-5.57	-4.72
4PADCB + DMSLA	16.56	0.89	-5.53	-4.64
Perovskite	16.6	1.18	-5.78	-4.6
Perovskite +LA	16.48	1.14	-5.86	-4.72

R2 - Comment 12.

Line 297: Please verify whether the energy offset is 0.02 eV rather than 0.04 eV.

R2 - Response 12.

We thank the reviewer for drawing our attention to this point. In our analysis, the relevant quantity for hole extraction is the valence-band offset, i.e. the difference in the E_{F} -VBM separation between the SAM-modified ITO and the perovskite+LA absorber, as obtained from the UPS E_{onset} values in Supplementary Table 1. For perovskite+LA, the VBM lies 1.14 eV below E_{n} ($E_{\text{onset}} = 1.14$ eV), whereas for the different SAM stacks the corresponding values are 0.80 eV (4PADCB), 0.83 eV (4PADCB+LA), 0.85 eV (4PADCB+DHLA), and 0.89 eV (4PADCB+DMSLA). The valence-band offset with perovskite+LA is therefore 0.25 eV for 4PADCB+DMSLA, which is the smallest among the modified SAMs. The energy offset in the relevant line, calculated based on the figure shown in Fig. 3, has been corrected to 0.25 eV as follows:

- Lines 307-310: “As can be seen from the energy level diagram derived from the UPS data presented in Figure 3d—which is critical for charge transfer at the perovskite/SAM interface—the combination of 4PADCB with DMSLA results in the

smallest energy offset (0.25 eV) between the valence band maximum (VBM) of the modified ITO and that of the LA doped perovskite.”

R2 - Comment 13.

How was the bandgap calculated? If it was derived from UV–vis spectra, please provide the corresponding absorption spectra with a proper baseline.

R2 - Response 13.

We thank the reviewer for this comment and for noticing the need for clarification. In the original version of the Supplementary Information, we incorrectly stated that the measurements were performed “in transmittance mode.” In fact, the optical bandgap estimation was based on combined transmittance and reflectance measurements obtained using a PerkinElmer Lambda 1050 spectrometer on thin-film samples.

The absorption (A) was calculated according to $A = 1 - T - R$, where T and R represent the measured transmittance and reflectance, respectively. This approach ensures a baseline-corrected absorption spectrum, as it inherently accounts for substrate transmission and reflection losses, thereby avoiding artificial offsets.

Tauc plots derived from these corrected absorption spectra were then used to estimate the optical bandgaps. As shown in Supplementary Figure 22a–b, the spectra exhibit nearly identical profiles for all samples, with a slight red shift in the DMSLA-treated film. The extracted bandgaps were approximately 1.55 eV for all conditions, confirming negligible variation.

We have corrected the description in the Supplementary Information to accurately reflect the combined transmittance–reflectance method as follows:

“UV–Vis optical measurements were carried out using a PerkinElmer Lambda 1050 spectrometer equipped with an integrating sphere. Both transmittance (T) and reflectance (R) spectra were recorded on thin-film samples deposited on glass substrates. The absorbance (A) of the films was then calculated according to the relation $A = 1 - T - R$, which inherently corrects for baseline offsets by accounting for substrate transmission and surface reflection losses. The resulting spectra were used to construct Tauc plots for bandgap estimation, as shown in **Supplementary Figure 22a–b**. All films exhibited similar absorption characteristics with no major spectral shifts, although a slight red shift was observed for the

DMSLA-treated sample. The estimated optical bandgaps for all conditions were approximately 1.55 eV, confirming negligible variation among the samples (**Supplementary Figure 22-b**).”

R2 - Comment 14.

Page 12, line 328: Please explain how the device temperature was maintained at $-80\text{ }^{\circ}\text{C}$ when immersed in liquid N_2 .

R2 - Response 14.

We appreciate the reviewer’s request for clarification. The thermal cycling experiments were performed using a custom-designed closed-lid setup rather than direct immersion in liquid nitrogen. The complete configuration and workflow are illustrated in Supplementary Figure S26. The system was engineered to achieve reproducible and controlled thermal cycling between $-80\text{ }^{\circ}\text{C}$ and $+80\text{ }^{\circ}\text{C}$.

To prevent frost formation and ensure dry ambient conditions, dry air was circulated within the sealed lid, and cycling was initiated only after the DHT22 temperature–humidity sensor indicated 0% relative humidity (0–100% RH range, $\pm 2\%$ accuracy, $\pm 5\%$ max deviation). Dry air circulation was maintained throughout the experiment to ensure a consistent, moisture-free environment.

The temperature control was achieved through aluminum plates of optimized thickness, designed according to heat-transfer calculations to match the desired heating and cooling rates. The plates were placed beneath the samples inside the chamber, providing efficient thermal conduction while allowing a total cycle duration of ~ 90 minutes ($-80\text{ }^{\circ}\text{C}$ to $+80\text{ }^{\circ}\text{C}$ and back). The temperature was monitored directly from the sample surface using a K-type thermocouple attached to the device substrate, ensuring accurate measurement of the actual thermal load experienced by the cells.

The following modifications have been implemented in **Section 2.4, “Thermal Fatigue Stability Analysis”**:

- “We performed the thermal fatigue test using a custom-built close-lid setup, as standard climate chambers could not reach the required $-80\text{ }^{\circ}\text{C}$ cryogenic range, which induces the highest stress at the perovskite–substrate interface. The samples were cycled between $-80\text{ }^{\circ}\text{C}$ and $+80\text{ }^{\circ}\text{C}$ in a stainless steel container, with controlled

heating and cooling rates of +3.40 and $-3.80\text{ }^{\circ}\text{C min}^{-1}$, respectively, and a total cycle duration of ~ 90 minutes under dry ambient conditions (Figures 4a, b). A schematic and further explanation of the experimental setup are provided in Supplementary Figure 26.”

More detailed information on the thermal cycling procedure is presented in the Supporting Information under Methods titled as “**Thermal Cycling Tests**”:

- “Thermal cycling experiments were performed using a custom-built closed-lid setup designed to achieve reproducible and well-controlled temperature cycling between $-80\text{ }^{\circ}\text{C}$ and $+80\text{ }^{\circ}\text{C}$. To prevent frost formation and maintain dry ambient conditions, dry air was circulated within the sealed chamber, and cycling was initiated only after the integrated DHT22 temperature–humidity sensor indicated 0% relative humidity. The sample temperature was monitored directly from the device surface using a K-type thermocouple placed in contact with the substrate to accurately capture the thermal load experienced by the cells. A schematic representation of the thermal fatigue experimental setup is provided in **Supplementary Figure 26.**”

R2 - Comment 15.

The control devices retained 79% of their initial efficiency after 16 cycles, DMSLA showed 84%, and DHLA showed 72%. These differences appear minor (within $\pm 5\%$ deviation). How many devices were measured to draw this conclusion?

R2 - Response 15.

The custom-built thermal cycling setup allows testing of up to four samples simultaneously. Because each thermal cycle lasts approximately 90 minutes and the process requires careful manual operation for controlled heating and cooling, it was only feasible to test one representative device per condition in this initial proof-of-concept study. For each device, the solar cell contains six independent pixels, and each reported value represents the median of six $J-V$ measurements, providing internal statistical variation within the same device.

To further clarify the variability, we have now included the full statistical distributions of all photovoltaic parameters (PCE, V_{OC} , J_{SC} , and FF) in the Supporting Information for all three cycling durations (2 min, 15 min, and 90 min). We also revised the caption of Figure 4 to explicitly state that each data point in the main text corresponds to the median value extracted from six pixels of a single device.

These additions make the basis of our conclusion transparent and highlight that, despite the small differences in median PCE retention, the relative stability trends are consistently reproduced across the individual pixel data.

Supplementary Figure 30. Statistical distribution of photovoltaic parameters under 90-minute thermal-cycling conditions for devices fabricated at LMU Munich. Boxplots show the normalized (a) PCE, (b) V_{oc} , (c) J_{sc} , and (d) FF for control, LA-treated, DHLA-treated, and DMSLA-treated perovskite solar cells measured after encapsulation (baseline), and after 1, 3, 8, and 16 thermal cycles. Each data point corresponds to an individual device pixel (six pixels per device). Parameter values are normalized to those measured at cycle 0 for each device. Darker-colored data points correspond to the forward $J-V$ scans, while lighter-colored data points correspond to the reverse $J-V$ scans.

R2 - Comment 16.

Although the variation is small, why does the V_{OC} curve in Fig. 4a for DMSLA show a decrease after five cycles, followed by a gradual increase after 16 cycles?

R2 - Response 16.

We thank the reviewer for this observation. The transient decrease and subsequent recovery of V_{OC} in the DMSLA-treated devices likely reflect interfacial relaxation and stabilization processes occurring during repeated thermal cycling. We hypothesize that in the early cycles, exposure to alternating extreme temperatures may cause temporary structural or dipolar reorganization within the self-assembled DMSLA layer or at the perovskite/HTL interface, slightly disturbing the built-in potential and resulting in a lower V_{OC} . Upon further cycling, the system appears to re-stabilize, possibly through thermally assisted reordering of interfacial dipoles or partial defect healing, leading to the observed recovery in V_{OC} .

We emphasize that this explanation is hypothesis-based. While no prior studies to our knowledge report the specific V_{OC} decrease-then-recovery behavior under repeated thermal cycling in SAM-modified PSCs, several recent works have demonstrated that interfacial dipoles and SAM/bilayer interface engineering can influence V_{OC} and interface stability^[R3,R4]. But further studies (e.g., in-situ spectroscopic or electrical characterization during cycling) would be required to confirm the underlying mechanism under thermal cycling conditions.

[R3] Dong, B., Wei, M., Li, Y. et al. Self-assembled bilayer for perovskite solar cells with improved tolerance against thermal stresses. *Nat Energy* 10, 342–353 (2025).

<https://doi.org/10.1038/s41560-024-01689-2>

[R4] Peng, Y., Chen, Y., Zhou, J. et al. Enlarging moment and regulating orientation of buried interfacial dipole for efficient inverted perovskite solar cells. *Nat Commun* 16, 1252 (2025). <https://doi.org/10.1038/s41467-024-55653-5>

We have added this discussion under Section 2.4, Thermal Fatigue Stability Analysis, as follows:

- “The transient decrease and subsequent recovery in V_{OC} observed for DMSLA-treated devices may arise from interfacial relaxation during repeated thermal cycling. Such behavior could reflect temporary perturbation of the interfacial energetics followed by

stabilization upon continued cycling. Further studies would be required to determine the exact mechanism.”

R2 - Comment 17.

In Materials section in SI, BCP and LiF were mentioned twice and also sourced from different suppliers. Please indicate the reason.

R2 - Response 17.

We thank the reviewer for this comment. In this study, the perovskite-based solar cells were fabricated in two separate laboratories (line 126 at LMU Munich and line 143 at Tianjin University). Each group listed the chemicals obtained from different suppliers because they procured the reagents independently for their respective experiments.

R2 - Comment 18.

Dyesol company no longer exists. Please confirm FA and MA were sourced from GreatCell Solar.

R2 - Response 18.

Thanks for the reviewer’s reminder. The information regarding the source of the organic halide salts has been updated in the Materials and Methods section of the Supplementary Information as follows:

- “Cesium iodide (CsI, 99.5%), Lead (II) iodide (PbI₂, 99.99%), Lead(II) Chloride (PbCl₂, 99.0%), α -lipoic acid (LA) and dihydrolipoic acid (DHLA) were sourced from TCI. N, N-dimethylformamide (DMF, 99.8%), dimethyl sulfoxide (DMSO, 99.8%), Isopropanol (IPA, 99.8%), Chlorobenzene (CB, 99.9%), and ethanol (EtOH, 99.8%) were purchased from Sigma Aldrich. Methylammonium iodide (MAI, 99.5%), Methylammonium bromide (MABr, 99.5%) and formamidinium iodide (FAI, 99.5%) were purchased from GreatCell Solar Materials. Fullerene (C₆₀, 99.5%) and (4-(7H-dibenzo[c,g]carbazol-7-yl)butyl)phosphonic acid (4PADCB) were purchased from Lumtec. Indium tin oxide (ITO, YXKJGI-0006, 15 Ω /sq), and Bathocuproine (BCP) were purchased from Yingkou Advanced Election Technology Co., Ltd. Lithium fluoride (LiF) and Ethanediamine dihydroiodide (EDAI₂) were purchased from Xi’an Polymer Light Technology, China. All chemicals were used as it is without further purification.”

R2 - Comment 19.

Please provide more details in the caption of Fig. S24. Do the dark blue and light blue boxes correspond to the reverse and forward scans, respectively? The same clarification is suggested for the colours used in Figs. S24 and S25.

R2 - Response 19.

We thank the reviewer for pointing out the need for clearer color labeling in the $J-V$ scan figures. In the revised version of the Supporting Information, Supplementary Figure S24 has been updated to include explicit labels indicating which colors correspond to the forward and reverse scans.

Supplementary Figure 24: Statistical distribution of a) PCE (%), and b) J_{sc} values for the solar cell devices fabricated at LMU Munich.

Likewise, the caption of **Supplementary Figure S25** has been revised to clarify that the darker-colored curves represent the forward scans, while the lighter-colored curves represent the reverse scans.

“**Supplementary Figure 25:** Statistical distribution of a) Voc, b) PCE (%), c) FF (%), and (d) J_{sc} values for the solar cell devices fabricated at Tianjin University. Darker-colored data points correspond to the forward $J-V$ scans, while lighter-colored data points correspond to the reverse $J-V$ scans.”

Reviewer #3

Remarks to the Author:

The manuscript addresses a crucial aspect for the employment of perovskite solar cells in Low Earth Orbit (LEO): their resilience to thermal fatigue. The authors propose an innovative two-step approach, based on the use of α -lipoic acid and its derivatives, to improve perovskite film cohesion and mitigate delamination. To validate this strategy, they employed several experimental techniques such as XRD, FTIR, AFM, SEM, GIWAXS, which confirmed a significant improvement in both internal grain cohesion and interfacial adhesion. Thermal fatigue tests, performed with a custom-built setup and a thermal cycling protocol ranging from -80°C to $+80^{\circ}\text{C}$ for 16 cycles (equivalent to one day of operation), indicate encouraging stability. The best performing device achieved an efficiency of 26% and retained 84% of its initial performance after the thermal fatigue test.

The manuscript is well written, logically structured, and contextualized within the literature. The experimental work is carefully executed, and the short-term data are convincing within the tested scope. The analytical framework is solid, and the interpretation of the available data is careful.

However, while the proposed strategy is novel and the results are encouraging, in our opinion, the current experimental validation does not provide sufficient evidence to substantiate claims of aerospace suitability for the following reasons:

Response:

We thank the editor for the thorough and encouraging assessment of our manuscript. We are pleased that the experimental design, analytical framework, and short-term thermal-fatigue data were found convincing and well contextualized within the literature. We also appreciate the recognition of the novelty of our two-step reinforcement strategy and the reviewer's acknowledgement of the careful execution of our structural, mechanical, and interfacial analyses.

We fully agree with the reviewer that the current experimental validation should not be interpreted as full evidence of aerospace qualification. Our intention was not to claim complete readiness for deployment in space, but rather to provide a proof-of-concept demonstration of how molecular and interfacial engineering can mitigate thermomechanical fatigue under LEO-relevant temperature cycling. In the revised manuscript, we have clarified the scope of the work by adjusting the title, refining statements in the abstract, introduction, and conclusion, and explicitly stating that additional stressors relevant to the space environment—such as vacuum, ultraviolet radiation, ionizing radiation, and atomic oxygen—remain to be evaluated in future studies.

We sincerely appreciate this feedback, and we believe that the revisions improve the accuracy, transparency, and positioning of the manuscript. In the response letter, we provide detailed responses to each point raised by the reviewers and describe all corresponding changes made in the manuscript.

R3 - Comment 1.

The thermal cycling test (16 cycles) is far too limited to represent orbital conditions, which typically involve hundreds or thousands of cycles.

R3 - Response 1.

We thank the reviewer for raising this important point. To investigate whether the number of cycles or the total thermal-exposure duration is the dominant factor in degradation, we synthesized molecules again, fabricated new cells, encapsulated them, and performed two additional thermal-cycling experiments beyond the initial 90-minute cycles:

(i) a fast 2-minute cycle, and

(ii) a moderate 15-minute cycle,

each carried out for 16 cycles (equivalent number of cycles but different total thermal exposure). The idea behind this was to understand how feasible it is to perform a large number of cycles. As a side note, we do not currently have the capability in our lab to perform realistic tests with extreme cycling conditions.

Across all conditions, the degradation was extremely small, as shown in the new contour plots and boxplot distributions (Supplementary Figures 30–33). For all devices—including the control group—PCE, V_{OC} , J_{SC} , and FF remained within ~1–5% of their initial values, with

no systematic decline across cycle number for the 2-min and 15-min experiments. This is clearly visible in the boxplot distributions, where the parameter scatter remains tightly clustered around unity after 1, 3, 8, and 16 cycles.

In contrast, the 90-minute cycles produced measurable degradation in the control devices, consistent with our original report. When comparing the three experiments, a clear trend emerges: Thermal fatigue correlates far more strongly with the total time spent under thermal stress than with the number of cycles.

This is also reflected in our new contour maps, which show that degradation contours follow lines of constant total thermal-exposure time rather than constant cycle number. For example, the control devices begin to show a drop only when total exposure exceeds ~8–12 hours (e.g., 90-minute \times 16 cycles), while 2-minute and 15-minute cycling (total exposure <4 hours) show essentially no degradation.

This observation is fully consistent with the literature. As summarized in the new comparative table (Table R1), studies that report significant thermal-cycling degradation typically involve either:

very large temperature ranges (e.g., -143 to $+157$ °C),

vacuum conditions,

or long cumulative exposure times (hundreds to thousands of minutes),

as seen in the works of Lu et al. (2025), Yang et al. (2024), Bautista et al. (2022), and Chen et al. (2024). Conversely, studies with similar or greater cycle counts but shorter total thermal exposure (e.g., 20–30 minutes per cycle or rapid cycling) show minimal degradation even after 50–200 cycles (Li et al., 2022; Zhihao Li et al., 2024; Guixiang Li et al., 2023). Our findings align with this trend: the absolute number of transitions across 0 °C appears to be a weaker predictor of failure than the integrated thermomechanical load, which is governed by the total time spent at extreme temperatures and the duration of thermal gradients.

Therefore, our expanded dataset and cross-comparison with literature demonstrate that thermal-cycling degradation is governed by cumulative exposure time and thermomechanical stress accumulation, rather than cycle count alone. This insight further highlights the importance of choosing a cycle duration that realistically reflects the thermal-exposure profile of Low Earth Orbit (LEO). In LEO, the heating/cooling transitions are slow (typically 30–45

minutes per half-cycle), and the total exposure time is thus a more accurate representation of the mechanical fatigue experienced by actual hardware.

We have included these new datasets, contour plots, boxplots, and Supplementary Table 2 in the revised Supplementary Information.

Supplementary Figure 33. Contour maps showing the dependence of normalized photovoltaic parameters on cycle duration and number of thermal cycles for control perovskite solar cells. Panels display the normalized (a) PCE, (b) V_{OC} , (c) J_{SC} , and (d) FF across a matrix of cycle durations (2–90 minutes) and cycle counts (0–16). Star symbols indicate the experimentally tested conditions in this work (2-min, 15-min, and 90-min cycling, each for 16 cycles). White dashed lines represent contours of constant total thermal-exposure time. All parameters are normalized to their respective values at cycle 0. These plots reveal that degradation aligns more strongly with total thermal-exposure time rather than cycle count, with negligible changes observed for short-duration cycles and measurable performance drops appearing only for extended exposure (e.g., 90-min cycles).

Supplementary Figure 31. Statistical distribution of photovoltaic parameters under 2-minute rapid thermal-cycling conditions for devices fabricated at LMU Munich. Boxplots show the normalized (a) PCE, (b) V_{oc} , (c) J_{sc} , and (d) FF for control, LA-treated, DHLA-treated, and DMSLA-treated devices evaluated after 0, 1, 3, 8, and 16 cycles. Each point represents a single active pixel on the device. All photovoltaic parameters remain essentially unchanged throughout cycling, confirming that rapid temperature transitions with short exposure times impose negligible stress and further supporting that performance degradation is governed by cumulative thermal-exposure duration rather than the number of cycles. Darker-colored data points correspond to the forward $J-V$ scans, while lighter-colored data points correspond to the reverse $J-V$ scans.

Supplementary Figure 32. Statistical distribution of photovoltaic parameters under 15-minute thermal-cycling conditions for devices fabricated at LMU Munich. Boxplots show the normalized (a) PCE, (b) V_{OC} , (c) J_{SC} , and (d) FF for control, LA-treated, DHLA-treated, and DMSLA-treated solar cells measured after 0, 1, 3, 8, and 16 cycles. Each data point corresponds to an individual device pixel (six pixels per device). Values are normalized to the corresponding parameter measured at cycle 0 for each sample. All device groups indicate moderate thermomechanical degradation under moderate cycle duration and demonstrating that cycle count alone is not a sufficient predictor of fatigue when compared to 90-minute thermal-cycling conditions. Darker-colored data points correspond to the forward $J-V$ scans, while lighter-colored data points correspond to the reverse $J-V$ scans.

R3 - Comment 2.

The working conditions of solar cells in space are extremely harsh, involving not only thermal fluctuations but also high vacuum, atomic oxygen, and high-energy particle radiation (such as protons, electrons, neutrons, γ -rays). Although the authors briefly acknowledge the need for vacuum tests to better simulate the space environment, they do not consider the importance of radiation, leaving a fundamental gap in assessing material's long-term stability. Without such tests, the applicability of the solar cells for aerospace missions remains speculative.

R3 - Response 2.

We thank the reviewer for this valuable comment, highlighting the importance of considering radiation effects when evaluating solar-cell applicability for space missions. From this feedback, we understand that our text may have given the impression that we had fully addressed all challenges associated with space operations, although this was not our intention. Our study was designed to approximate realistic thermal stress conditions as closely as possible, focusing on thermal fatigue resistance under low-Earth-orbit (LEO)–relevant temperature fluctuations as a first step toward understanding device reliability in space-like environments. At this stage, it is naturally difficult to perform experiments that include all relevant stressors simultaneously—such as thermal cycling, vacuum exposure, atomic oxygen, ultraviolet radiation, and ionizing radiation. This situation is analogous to accelerated testing for terrestrial photovoltaics, where individual tests (e.g., thermal cycling, humidity freeze, or potential-induced degradation) are typically conducted separately and later correlated with field performance. Similarly, the development of standardized accelerated testing protocols for perovskite devices under space-relevant conditions is still in its early stages. To build on this study, our ongoing work involves the development of a thermo-vacuum cycling platform that will enable combined thermal and vacuum testing under space-relevant conditions with AM0 irradiation.

To clarify the scope and temper our claims, we have revised the manuscript title, abstract, introduction, results, and conclusion accordingly.

Title

We have revised our manuscript title from “**Perovskite Solar Cells with Enhanced Thermal Fatigue Resistance in Low-Earth Orbit Space Extremes**” to “**Perovskite Solar**

Cells with Enhanced Thermal Fatigue Resistance in Low-Earth Orbit **Relevant Temperature Extremes.**”

Abstract

- Line 27: “**Previous works have shown that** PSCs can exhibit high radiation tolerance; however, their stability under the extreme thermal cycling conditions of space—a key factor driving degradation in LEO—remains largely unexplored.”
- Line 42: “Dual reinforced devices exhibit 84% performance retention after 16 thermal cycles, representing one full day in **LEO-relevant temperature extremes.**”
- Lines 43–45: “This work highlights the importance of targeted interface engineering and thermal fatigue mitigation strategies for advancing PSCs **toward** space applications.”

Introduction

- Lines 68–71: “Hence, to **approximate** the mechanical stresses associated with these rapid and extreme temperature shifts, we adopted a custom thermal cycling protocol ranging from -80 °C to +80 °C^[10], which necessitated the development of a dedicated test setup. **We performed a one-day-equivalent experiment with realistic cooling and heating rates representative of LEO-relevant temperature extremes under dry-air laboratory conditions.**”

[10] Kim, K., Yang, S., Kim, C. et al. Non-volatile solid-state 4-(N-carbazolyl)pyridine additive for perovskite solar cells with improved thermal and operational stability. Nat Energy (2025). <https://doi.org/10.1038/s41560-025-01864-z>

- Lines 95–96: “However, most studies still lack comprehensive thermal fatigue assessments under **LEO-relevant temperature conditions, even in laboratory environments.**”

Results and Discussion

- Lines 344–347: “The high PCE retention observed during MPPT measurements, compared to the lower retention seen in thermal fatigue tests, indicates that **LEO-relevant temperature cycling** may present a more significant challenge than continuous MPPT operation.”

Conclusion and Outlook

- Deleted sentence (lines 372–374): ~~“Regarding testing setups, implementing vacuum thermal cycling tests could better replicate space-like conditions, highlighting the need for a customized vacuum-compatible tool.”~~
- Deleted sentences (line 376): ~~“To the best of our knowledge, this is the first report to systematically study extreme low-temperature cycling experiments for PSCs.”~~
- Added sentences (starting from line 376): **“It should also be noted that our experiments were conducted under dry-air and ambient-pressure conditions; additional space-relevant stressors such as vacuum, ultraviolet radiation, atomic oxygen, outgassing, and ionizing radiation remain important challenges requiring further investigation. Our ongoing work includes the development of a thermo-vacuum cycling system to extend these findings toward more realistic multi-stressor environments.”**
- Revised final sentence: **“We believe this work lays the foundation for new research directions and contributes valuable insight into the behavior of perovskite solar cells under **simulated LEO-relevant temperature cycling.**”**

R3 - Comment 3.

The proposed additive engineering approach, based on organic additives (lipoic acid and its derivatives), is innovative and could have relevance for improving the durability of perovskite solar cells more broadly. However, the radiation stability of these organic molecules is a fundamental issue that should be addressed, either experimentally or via literature-supported discussion.

R3 - Response 3.

We thank the reviewer for raising this important point. At present, it remains unclear whether the LA and derivative molecules incorporated at grain boundaries and interfaces behave as independent organic species under radiation exposure, or whether they become sufficiently integrated within the perovskite lattice to alter their intrinsic radiation response. Our current laboratory infrastructure does not allow us to immediately assess molecular-level radiation stability. Also, radiation stability is not the focus of this work, and we have revised the text to narrow the scope toward the thermal-cycling behaviour of the devices. However, this is a valid question, and we have checked the literature on this topic. Nonetheless, insights from the literature provide useful context. Studies on sulfur-containing organic compounds, including disulfide-bearing systems structurally related to LA, show that S–S and C–S bonds are among the most radiation-sensitive moieties under high-energy irradiation. For example, Bhattacharyya et al. demonstrate that disulfide bonds undergo electron-induced cleavage, forming radical intermediates that drive bond scission under ionizing radiation.^[R5] Similarly, low-energy secondary electrons generated during γ - or X-ray irradiation are known to cause attachment-induced damage in organic molecules, particularly affecting functional groups with electron-rich sulfur atoms.^[R6] These observations indicate that LA or its derivatives could, in their isolated form, be susceptible to radiation-induced fragmentation. However, this does not imply that using them as ultra-thin layers in devices will lead to the same limitations, which warrants further investigation.

However, it is important to note that actual radiation stability in space is strongly influenced by encapsulation and shielding. In many space-qualified photovoltaic technologies, proper encapsulation significantly reduces the radiation dose absorbed by organic interlayers.^[R7]

A systematic investigation of the radiation stability of LA-based additives within perovskite devices is beyond the scope of the present study but represents a relevant direction for future work. Guided by existing literature, we plan to address this in follow-up studies once dedicated radiation-testing access becomes available. Our current work focuses specifically on thermal-fatigue mitigation rather than a full multi-stressor space qualification.

[R5] Bhattacharyya, R., Dhar, J., Ghosh Dastidar, S., Chakrabarti, P. & Weiss, M. S. The susceptibility of disulfide bonds towards radiation damage may be explained by S...O interactions. *IUCrJ* 7, 825–834 (2020).

[R6] Narayanan S J, J., Tripathi, D., Verma, P., Adhikary, A. & Dutta, A. K. Secondary Electron Attachment-Induced Radiation Damage to Genetic Materials. ACS Omega 8, 10669–10689 (2023).

[R7] Zhang, H., Wei, K., Zhang, G. et al. Evaluating the protective efficacy of polymer encapsulation layer for perovskite solar cells under space radiation exposure. Moore. More 2, 14 (2025). <https://doi.org/10.1007/s44275-025-00031-6>

R3 - Comment 4.

The lack of comprehensive environmental validation severely limits the impact of the findings for aerospace applications. In its present form, the study is best considered as a proof-of-concept for improved perovskite stability, not as evidence of readiness for space deployment.

R3 - Response 4.

We thank the reviewer for this important comment. We now recognize that the original title and certain expressions in the abstract may have unintentionally suggested a broader level of environmental validation than was carried out. This was not our intention. Our study focuses specifically on understanding and mitigating thermal-fatigue mechanisms under LEO-relevant temperature cycling.

To avoid any misunderstanding, we have revised the manuscript title and adjusted the wording in the abstract, introduction, and conclusion to clearly state that the present work does not constitute a complete multi-stressor space qualification. Instead, it represents an initial step toward identifying interface-engineering strategies that improve thermal robustness in conditions relevant to LEO temperature cycling. We believe these revisions appropriately frame the scope and contribution of the study.

Reviewer #4

Remarks to the Author:

This manuscript presents an interesting study on the use of additives to enhance the thermal fatigue resistance of perovskite solar cells for space applications. The work is novel and offers valuable insights, especially since thermal fatigue has received comparatively little attention in this field.

I recommend a revision before the manuscript can be accepted for publication. My detailed comments are as follows:

Response:

We thank the editor for the positive evaluation of our work and for recognizing the novelty and relevance of our study on enhancing the thermal fatigue resistance of perovskite solar cells for space applications. We appreciate the constructive guidance and the opportunity to revise the manuscript. In the revised version, we have carefully addressed all reviewer comments in detail and have strengthened the clarity, accuracy, and scope of the manuscript accordingly. Below, we provide a point-by-point response to each comment and outline all corresponding changes made in the text.

R4 - Comment 1.

Since the context is space applications, why was AM0 not used for device measurements? AM0 is more relevant, and its higher UV component could also influence additive stability. Please provide AM0 measurements and comparisons, as this seems necessary.

R4 - Response 1.

We thank the reviewer for these suggestions. Indeed, this is a critical point and following the reviewer's recommendation, we now added IV curves and statistical distribution of the corresponding devices at AM0 (1360 W/m²). The devices obtained higher J_{sc} and V_{oc} at AM0 compared to those at AM1.5 spectrum.

Supplementary Figure 28. J–V curves and the corresponding device parameters of the devices at AM0.

Supplementary Figure 29. Statistical distribution of a) V_{oc} , b) PCE, c) FF, and (d) J_{sc} values of the devices at AM0. Each condition had 10 devices.

R4 - Comment 2.

In the adhesion test, the ranking is DMSLA > DHLA > LA > control. However, this ranking does not align with the thermal cycling results. Does this suggest that adhesion is not the main factor contributing to thermal cycling failure? Please explain and discuss possible alternative failure modes with supporting evidence.

R4 - Response 2.

We thank the reviewer for this insightful comment. While the adhesion measurements show a clear ranking of DMSLA > DHLA > LA > control, we agree that this ranking does not directly mirror the thermal-cycling stability results. This discrepancy is consistent with the

fact that thermomechanical degradation in perovskite solar cells is governed by more than interfacial adhesion alone.

First, our nanomechanical AFM data (Fig. 1c and Supplementary Figs. 9–10) show that LA does not measurably increase adhesion at grain boundaries, whereas DHLA and DMSLA improve grain-boundary adhesion by >50% and ~40%, respectively. This indicates that the polymer network formed from LA during annealing provides the weakest mechanical reinforcement in the regions most susceptible to strain accumulation. Second, the pull-off tests (Fig. 2c–d) reveal that LA achieves only a modest increase in substrate–perovskite adhesion, far below that obtained with DMSLA. These two measurements consistently show that LA-treated films are mechanically less robust both at the grain boundaries and at the ITO/perovskite interface. Correspondingly, LA-only devices exhibit the lowest performance retention after thermal cycling (64%), supporting the conclusion that insufficient mechanical reinforcement leads to more pronounced degradation under repeated temperature excursions.

In contrast, DMSLA combines strong adhesion with greater chemical stability and stronger interfacial interactions, as supported by our DFT analysis (highest interfacial interaction energy; Fig. 2e–f), XPS peel-off results (largest –COO signal on the perovskite side), cyclic voltammetry (highest SAM surface coverage), and UPS (strongest work-function shift due to its large dipole moment). These properties allow the DMSLA-modified interface to better withstand repeated thermomechanical loading, explaining its superior thermal-cycling stability despite DHLA having comparable or slightly higher initial adhesion in some measurements.

Taken together, our results indicate that adhesion is necessary but not sufficient for thermal-fatigue resistance. Thermal cycling exposes the device to volumetric expansion and contraction across multiple interfaces and grain boundaries; therefore, overall stability depends on the combined effects of interfacial adhesion, chemical robustness, and the ability of the grain-boundary network to dissipate strain. We have added a concise explanation in the revised manuscript to clarify this point.

- Lines 352-360: “We note that the adhesion ranking (DMSLA > DHLA > LA > control) does not directly mirror the thermal-cycling results because adhesion represents only one component of thermomechanical stability. Thermal cycling also depends on the chemical robustness of the linker molecules and the ability of the grain boundaries to accommodate strain. DMSLA combines strong and stable interfacial

bonding—supported by our DFT, XPS, CV, and UPS data—with higher chemical stability under thermal stress, whereas DHLA’s thiol groups are more reactive and may undergo changes during cycling. This interplay of interfacial adhesion, chemical stability, and grain-boundary reinforcement explains the superior cycling performance of DMSLA.”

R4 - Comment 3.

For the thermal cycling tests performed using the in-house equipment, was moisture controlled? The encapsulation method described in the paper may not effectively block moisture, and heating/cooling cycles could allow moisture ingress, which would affect device performance. This condition would not reflect the space environment, so clarification is needed.

R4 - Response 3.

We thank the reviewer for raising this critical point. The thermal cycling experiments were conducted in a custom-designed closed-lid setup that ensured a controlled and moisture-minimized environment throughout the entire test. The chamber was continuously purged with dry air before and during the measurements to eliminate any residual humidity and prevent frost formation on the samples, although it is difficult to say the test environment is humidity-free. A DHT22 temperature–humidity sensor monitored the internal conditions in real time, and cycling was initiated only once the humidity reached 0% relative humidity (sensor range 0–100% RH, $\pm 2\%$ accuracy, $\pm 5\%$ maximum deviation). The dry-air circulation was maintained during both heating and cooling phases to ensure that no moisture ingress occurred through the encapsulation or chamber interfaces. Moisture ingress is still possible, but all samples are encapsulated under identical conditions; therefore, relative comparisons remain valid, and performance differences can primarily be linked to the interface we study.

The following modifications have been implemented in **Section 2.4, “Thermal Fatigue Stability Analysis”**:

- “We performed the thermal fatigue test using a custom-built close-lid setup, as standard climate chambers could not reach the required $-80\text{ }^{\circ}\text{C}$ cryogenic range, which induces the highest stress at the perovskite–substrate interface. The samples were cycled between $-80\text{ }^{\circ}\text{C}$ and $+80\text{ }^{\circ}\text{C}$ in a stainless steel container, with controlled heating and cooling rates of $+3.40$ and $-3.80\text{ }^{\circ}\text{C min}^{-1}$, respectively, and a total cycle

duration of ~90 minutes under dry ambient conditions (Figures 4a, b). A schematic and further explanation of the experimental setup are provided in Supplementary Figure 26.”

More detailed information on the thermal cycling procedure is presented in the Supporting Information under Methods titled as “**Thermal Cycling Tests**”:

- “Thermal cycling experiments were performed using a custom-built closed-lid setup designed to achieve reproducible and well-controlled temperature cycling between –80 °C and +80 °C. To prevent frost formation and maintain dry ambient conditions, dry air was circulated within the sealed chamber, and cycling was initiated only after the integrated DHT22 temperature–humidity sensor indicated 0% relative humidity. A schematic representation of the thermal fatigue experimental setup is provided in **Supplementary Figure 26.**”

R4 - Comment 4.

The EQE results indicate that the front interface (light-incidence side) does not improve, and there is a noticeable loss at the rear interface when additives are used. Please explain this observation. The evidence for interfacial passivation is not very strong from EQE results.

R4 - Response 4.

Thanks for the reviewer's keen observation and valuable question. The factors influencing EQE results are complex, including non-radiative recombination in the bulk and at interfaces, as well as carrier extraction and transport efficiency in charge transport layer, etc. Therefore, the passivation effect cannot be simply inferred based solely on EQE results. Other reports on buried surface modification showed a similar trend (Small 2024, 20, 2404058; Nature, 2025, DOI: 10.1038/s41586-025-09785-3; J. Am. Chem. Soc. 2025, 147, 27, 23683-23695; Angew. Chem. Int. Ed. 2025, 64, e202504237).

We thank the reviewer for this important observation. The EQE response reflects the combined effects of optical absorption, bulk recombination, interfacial recombination, and charge-extraction efficiency. Because these processes are intertwined, the degree of passivation at the buried HTL/perovskite interface cannot be directly inferred from the EQE spectrum alone. In our devices, the molecular modifications introduced through the SAM layer are electronically active but optically negligible, and therefore large changes at the short-wavelength (illumination-side) region of the EQE are not expected.

Although the EQE does not explicitly show an improvement at the front interface, multiple electrical and photophysical measurements consistently indicate reduced recombination for DMSLA-treated devices. Specifically, the DMSLA samples exhibit higher V_{OC} and FF, as well as enhanced PL/ iV_{oc} values and longer TRPL lifetimes, all of which point to suppressed non-radiative recombination at the HTL/perovskite interface. These metrics are more sensitive probes of buried-interface passivation than EQE alone.

The modest reduction in EQE at longer wavelengths is consistent with reports on buried-interface SAM modifications, where the introduction of an ultrathin molecular layer can slightly redistribute the optical field or influence carrier-collection dynamics without compromising overall device performance (Small 2024, 20, 2404058; Nature 2025, DOI: 10.1038/s41586-025-09785-3; JACS 2025, 147, 23683–23695; Angew. Chem. Int. Ed. 2025, 64, e202504237). Importantly, the integrated JSC values remain consistent with J–V measurements, and the overall PCE improves for the DMSLA-treated devices, confirming that the interfacial enhancement is captured more effectively in electrical and photoluminescence measurements than in the EQE spectrum.

R4 - Comment 5.

For the MPPT test, please provide the corresponding IV curves and the initial PCE data.

R4 - Response 5.

Thanks for the reviewer's reminder. We have added the corresponding J – V curves and the initial PCE data.

Supplementary Figure 27. The corresponding J - V curves and the initial PCE data for the MPPT.

Reviewer #5

Remarks to the Author:

R4 - Response 5.

We would like to express our sincere appreciation for your contribution to the evaluation of our manuscript. We also appreciate Nature Communications' initiative to involve and recognize Early Career Researchers in co-reviewing.

Supplementary Table 2. Summary of reported thermal-cycling studies on perovskite solar cells under space-relevant or terrestrial conditions.

Study	Space Target	Temp Range (°C)	Cycles	Cycle Duration	Initial PCE (%)	Final PCE (%)	PCE Retention	Encapsulation	Key Finding	DOI
Lu et al., 2025	General space	-123 to +157	5	300 m	24.39	20.15	82.7%	Ultrahigh vacuum; no encapsulation	Absorber decomposition to Pb/PbI ₂ , volatile organics, ion migration under vacuum thermal cycling	10.22541/au.175547361.10502563/v1
Li et al., 2022	General space	-160 to +150	50	30 °C min ⁻¹ rate, 20 m	Not reported	Not reported	97% (AMO)	Not specified	Temperature-dependent phase transitions and recoverable lattice strain; widest temperature range tested	10.1002/aenm.202202887
Yang et al., 2024	Near-space/cryogenic	-143 to +17	120	81 m	24.34 (at 150K)	~17.5 (estimated)	72%	Not specified	PAN additive stabilized lattice at low temperatures; 72% retention after 120 cryogenic cycles (highest cycle count at extreme T)	10.1002/aenm.202400638
Guixiang Li et al., 2023	Terrestrial	-60 to +80	120	18 m	24.6	Not reported	93.9% at 80°C and 88.7% at -60°C	Unencapsulated	Impact of the ordered dipolar structure on the operational stability	10.1126/science.add7331
Zhihao Li et al., 2024	General space	-40 to +80	200		25.78 (rigid), 24.54 (flexible)	Not stated	>95%	Unencapsulated	In-situ cross-linked polymer; 200 thermal cycles + 10,000 bending cycles; >95% retention	10.1002/ange.202421063
Bautista et al., 2022	LEO CubeSat	-40 to +80	250	25 m	24.28	9.7 (estimated)	40%	UV cut film and polyimide/Kapton tape/ UV light curable glue/1 mm glass	HTM comparison under simulated space environment; carbon HTM outperformed organic/inorganic HTMs	10.2322/tjsass.65.95
Chen et al., 2024	Terrestrial/space readiness	-40 to +85	5000	5 m	21.3	17.2	81%	Unencapsulated	Alkyl-ammonium additive reduced residual stress; HIGHEST CYCLE COUNT for perovskites (2,500 cycles)	10.1021/acsnenergylett.4c00988
Kim et al., 2025		-80 to +80	200	40 m	Not stated	Not stated	>90%	Vacuum-packed	Non-volatile 4CP stabilizes Li ⁺ , suppresses oxides, enabling ~26% efficient, shock-stable perovskite cells	10.1038/s41560-025-01864-z
Our work	LEO	-80 to +80	16	90 m			84%	Space grade encapsulation, DOWSIL 93-500/0.2 mm glass	LA/DMSLA engineering makes ~26% efficient perovskite solar cells withstand LEO-like -80↔+80 °C thermal cycling	

Response to Reviewers' Comments on NCOMMS-25-67465

We sincerely thank all reviewers for their valuable and constructive comments, which have greatly contributed to improving the quality of our manuscript. Please find our detailed point-by-point responses below.

Reviewer #1:

Remarks to the Author:

The authors present a thorough investigation on the impact of a self-assembled monolayer using a lipoic acid and derivatives. There is rich investigation in the SAM and its functions at the grain boundaries of the perovskite – the methodology is sound. There is suitable information for the work to be reproduced. It is a concern of this reviewer that this manuscript makes broad claims of suitability for space with only 1-day low Earth orbit equivalent thermal cycling and no analysis of the devices in any other space relevant environments.

Response:

We would like to thank the reviewer for taking the time to review our manuscript and for providing valuable feedback that helped us improve the quality and clarity of our work. Also, we thank the reviewer for giving credit to our work. We now understand that our manuscript title and some expression in the abstract may have given a misleading impression—although this was not our intention—regarding the thermal cycling experiments and the tests performed under other space-relevant environments. Below, we address these points step by step.

R1 - Comment 1.

Line 85 sentence beginning “Across all these systems...” should cite where these claims come from, I believe they are from ref 17 and 18 but would be good to re-cite.

R1 - Response 1.

We thank the reviewer for this helpful suggestion. In the revised manuscript, we now explicitly re-cite the sources for the performance recovery claims. The sentence in the Introduction has been modified to:

- Lines 89-91 “Across all these systems, thermal-triggered healing has resulted in more than 80% recovery of solar cell performance after stress testing¹¹⁻¹⁸, while systems based on poly(LA) and poly(TA-NI) demonstrated recovery rates exceeding 90%^{17,18}.”

R1 - Comment 2.

The claims of this SAM being suitable for implementation in space are over inflated. These devices show moderate enhancement to only one stressor at 24 hrs. Authors should temper these claims by mentioning environmental caveats that could prove to be detrimental (vacuum, outgassing, UV degradation, ionizing radiation, etc.) or demonstrating experimentally or computationally that the SAM will exhibit longevity beyond 24 hours and that it can endure other stressors present in the space environment. The authors would likely agree that a material enduring 24 hours does not mean it will exhibit more durability than the control material that undergoes marginally more degradation (~5% absolute enhancement on retained performance). How can the authors demonstrate or suggest that the SAM would continue to outperform the control beyond 24 hours of thermal cycling and considering other relevant space environmental stressors?

R1 - Response 2.

We would like to thank the reviewer for the insightful comments regarding the thermal cycling tests. From the feedback, we understand that the original wording of our manuscript may have given the impression that our study aimed to demonstrate suitability for space or aerospace deployment, although this was not our intention. We therefore clarify that the present work is not a qualification study, but a reliability physics investigation focused on understanding thermomechanical degradation mechanisms in perovskite solar cells under repeated extreme temperature cycling.

Our tests were designed to isolate thermal fatigue as a stand-alone accelerated stress condition, rather than to replicate all environmental stressors simultaneously. This approach is consistent with established accelerated testing practices for terrestrial photovoltaic technologies (e.g., IEC and ISOS protocols), where individual stress factors—such as thermal cycling, damp heat, light soaking, or bias stress—are typically investigated separately, rather than combined in a single experiment. In principle, long-term stability predictions require correlating the outcomes of individual accelerated tests with field performance, but such correlations are still under active development for perovskite-based devices. As a side note, we would like to clarify that we currently lack the experimental capability to conduct combined stress tests, including thermal cycling coupled with radiation exposure, UV degradation, or vacuum conditions.

At the current stage of the field, standardised protocols for extreme temperature cycling in perovskite solar cells are not yet fully established, particularly with respect to cycle duration, ramp rates, and failure-mode relevance. Our intention here was therefore not to perform a comprehensive qualification test, but rather to conduct a representative accelerated thermal fatigue experiment to probe failure mechanisms arising from large temperature excursions and thermal expansion mismatch within the device stack. This perspective has been thoroughly elaborated in the new version of the manuscript.

To strengthen the discussion of thermal fatigue itself, we have added new experimental data examining the influence of thermal cycle duration and accumulated exposure time. Contour plots correlating device degradation with total thermal-exposure time (Figure 4c and Supplementary Figures 30–33) show that degradation trends are governed primarily by cumulative thermal stress rather than the number of cycles alone. These additions directly address the reviewer’s concern regarding the representativeness of the cycling protocol while remaining consistent with the clarified scope of the study.

In light of the reviewer’s valuable comment, we have revised our manuscript title from “Perovskite Solar Cells with Enhanced Thermal Fatigue Resistance in Low-Earth Orbit Space Extremes” to “Perovskite Solar Cells with Enhanced Thermal Fatigue Resistance **under Extreme Temperature Cycling.**” The new title mitigates the concern of the space-specific perspectives.

Furthermore, we have fully revised the Abstract, Introduction, and Conclusion to remove any implication of space or aerospace qualification and to ensure that the scope and claims of the study remain fully consistent with the experimental evidence presented.

In the Abstract:

We have fully revised the Abstract to remove references to space, LEO, radiation tolerance, and aerospace deployment, and to clearly frame the study as a thermal-fatigue-focused reliability investigation rather than an application-qualification study. The revised Abstract now emphasises extreme temperature cycling as an accelerated stress-testing protocol and highlights thermomechanical degradation mechanisms and interface engineering strategies without reference to space environments or mission scenarios.

- “Perovskite solar cells (PSCs) combine high power density with low-cost manufacturing, making them attractive for applications requiring lightweight and

mechanically compliant photovoltaic technologies. However, their long-term stability under repeated extreme temperature cycling—a critical driver of thermomechanical degradation in layered device stacks—remains insufficiently understood. Here, we investigate the thermal fatigue behaviour of PSCs subjected to cyclic temperature variations between $-80\text{ }^{\circ}\text{C}$ and $+80\text{ }^{\circ}\text{C}$ as an accelerated stress-testing protocol. Mismatched thermal expansion between the perovskite absorber and the glass substrate induces biaxial tensile strain, particularly during rapid temperature transitions, leading to degradation at both the substrate–perovskite interface and grain boundaries within the perovskite film. To address this, we introduce a co-additive SAM strategy using α -lipoic acid (LA), dihydrolipoic acid (DHLA), and a sulfonium-based cation (DMSLA) to enhance substrate–perovskite interfacial adhesion, while in situ polymerisation of LA during annealing strengthens grain-boundary cohesion. This dual reinforcement approach improves both mechanical robustness and optoelectronic performance, yielding stabilised power conversion efficiencies of up to 26% under AM1.5G illumination. Devices incorporating both modifications retain 84% of their initial performance after 16 extreme temperature cycles. Our experiments further reveal that the duration of thermal cycling is more critical than the number of cycles, with most degradation occurring during the initial cycles. These results underscore the importance of targeted interface engineering and grain-boundary reinforcement for improving the thermal fatigue resistance of PSCs under harsh temperature cycling conditions.”

In the Introduction;

We have also substantially revised the Introduction to address the reviewer’s concerns regarding overemphasis on space and aerospace applications. Specifically, the first three paragraphs of the Introduction have been reconstructed to remove application-driven framing and references to space deployment, low-Earth orbit operation, and aerospace qualification. The revised Introduction now focuses on the fundamental problem of thermomechanical degradation in layered perovskite device stacks under repeated extreme temperature cycling, positioning this work as a reliability physics study rather than an application-qualification effort.

- Lines 46–79: “Metal halide perovskite solar cells (PSCs) have attracted significant attention due to their high power conversion efficiencies, low fabrication costs, and compatibility with lightweight device architectures.¹ These attributes make PSCs

attractive for applications in which high specific power and mechanical compliance are required. They present a potential alternative to conventional III–V-based photovoltaic technologies, which are well known for their robustness under extreme operating conditions but are limited by high material and fabrication costs and relatively short operational lifetimes.² However, the long-term mechanical and structural stability of layered perovskite device stacks under repeated thermomechanical stress remains a key challenge limiting their broader deployment.

Repeated temperature cycling induces volumetric expansion and contraction within the solar cell stack, giving rise to mechanical fatigue that can ultimately lead to delamination, crack formation, or failure at weakly bonded interfaces.^{2,3} Such effects are particularly pronounced in multilayer thin-film devices due to mismatches in the coefficients of thermal expansion (CTE) between adjacent layers. In perovskite solar cells, large CTE differences between the glass substrate ($3.7 \times 10^{-6} \text{ K}^{-1}$)⁴, the transparent conductive oxide (e.g., ITO, $8.5 \times 10^{-6} \text{ K}^{-1}$)⁴, and the perovskite absorber (e.g., FAPbI₃, $\sim 203 \times 10^{-6} \text{ K}^{-1}$)⁵ concentrate strain at grain boundaries and heterointerfaces, accelerating mechanical degradation during temperature cycling.

Extreme temperature cycling represents a particularly severe stress condition for thin-film photovoltaic devices and is encountered in several scenarios, including high-altitude platforms, aerospace systems, and accelerated laboratory stress testing. In comparison to standard terrestrial qualification protocols—such as IEC thermal cycling tests typically limited to $-40 \text{ }^\circ\text{C}$ to $+85 \text{ }^\circ\text{C}$ —these conditions involve wider temperature excursions and often faster thermal ramp rates, resulting in rapid stress evolution within the device stack. Environments such as low-Earth orbit (LEO) are frequently cited as representative examples of extreme thermal cycling conditions, where repeated transitions between sunlight and shadow lead to pronounced temperature fluctuations, with cycling frequencies approaching $\sim 6,000$ cycles per year and thermal ramp rates on the order of $4\text{--}5 \text{ }^\circ\text{C min}^{-1}$, resulting in rapid stress evolution within the solar cell stack^{6–9}. Recent systematic studies have also examined perovskite solar cell performance under extreme temperature cycling and in-orbit conditions, further highlighting the importance of understanding thermal fatigue for space deployment.^{10–12} Understanding thermally induced mechanical fatigue under

these conditions is therefore essential for improving the durability of thin-film photovoltaic technologies for extreme temperature conditions.”

In the Results and Discussion;

- Lines 346–348: “After 16 thermal cycles under the applied temperature protocol, encapsulated solar cells with DMSLA retained 84% of their initial performance, compared to 79% for the control group.”
- Lines 384–387: “The high PCE retention observed during MPPT measurements, compared to the lower retention seen in thermal fatigue tests, indicates that extreme temperature cycling may present a more significant challenge than continuous MPPT operation.”

In the Conclusion and Outlook:

In addition, the Conclusion and Outlook section has been comprehensively rewritten to align with the revised scope of the manuscript. All references to space deployment, low-Earth orbit operation, and aerospace qualification have been removed. The revised Conclusion now focuses exclusively on the mechanistic insights gained into thermomechanical fatigue, grain-boundary reinforcement, and interfacial stabilization under repeated extreme temperature cycling. The Outlook has been reframed to emphasize future directions in reliability-oriented testing, such as extended automated cycling and multi-stressor laboratory studies, without implying device readiness for specific application environments.

- “In this work, we introduced a dual molecular reinforcement strategy to mitigate thermomechanical degradation in perovskite solar cells subjected to repeated extreme temperature cycling. This extreme cycling induces severe stress at the contact interfaces, particularly due to the much higher CTE of perovskites relative to substrate materials. In our novel two-step strategy, grain-to-grain cohesion was enhanced by incorporating α -lipoic acid (LA) into the perovskite precursor, enabling in situ polymerization during thermal processing, while interfacial adhesion between the perovskite layer and the underlying substrate was strengthened through chemical modification of the disulfide ring to a sulfonium group ($-S^+(CH_3)_2$), a methylated cationic moiety. Ultimately, the functional groups responsible for these effects — closed-ring sulphur, thiol, and sulfonium salt — were shown to form strong, direct interactions with the perovskite crystal lattice. Their combined contribution acts like a molecular suspension system, critically supporting structural integrity during thermal cycling.

While molecular tailoring strategies have been reported in the context of PSCs and flexible devices, our work uniquely combines grain-boundary reinforcement and interface stabilization to address thermal fatigue over a wide temperature range. The observed improvements in performance retention under repeated cycling highlight the importance of targeting mechanically vulnerable regions within multilayer perovskite device stacks, rather than focusing solely on optoelectronic optimization.

Looking forward, further improvements in thermal fatigue resistance may be achieved through the rational design of multifunctional molecular additives capable of controlled cross-linking across both grain boundaries and interfaces. Extending thermal cycling protocols to higher cycle numbers using automated testing platforms will be essential for establishing long-term degradation trends and for correlating accelerated stress tests with operational lifetime. More broadly, the concepts demonstrated here provide a general framework for improving the durability of perovskite photovoltaics operating under severe temperature cycling conditions.”

R1 - Comment 3.

JV measurements and MPPT were conducted at AM 1.5, but longevity claims are made for operation in space. Authors should comment on expected observations or variations at AM0 given they're framing this work for space applications. The solar illumination should be added in the body of the text because readers will assume AM0.

R1 - Response 3.

We appreciate this valuable feedback. Indeed, this is an important point, as it could potentially mislead readers. We performed the MPPT tests under equivalent AM1.5G illumination conditions, not under AM0 conditions. We note that achieving true AM0 illumination may remain a challenge for the community in the future as well, since generating a proper blue/UV spectral component for extended measurement durations is difficult at low cost (equipment manufacturer’s feedback). We hope to have these capabilities in the near future. To avoid any confusion for readers, we explicitly stated in the caption of the stability data that it refers to $\sim 1000 \text{ W m}^{-2}$ equivalent 1-sun intensity conditions:

“Figure 4. Thermal fatigue test of perovskite solar cell. a, Temperature profile of the thermal cycling protocol and the corresponding evolution of device parameters over sixteen

cycles. **b**, J - V characteristics of representative devices measured under $\sim 1000 \text{ W m}^{-2}$ equivalent 1-sun intensity illumination before and after thermal cycling. **c**, Contour maps showing the dependence of normalized fill factor (FF) on thermal cycle duration and number of cycles for control perovskite solar cells; white dashed lines indicate contours of constant total thermal-exposure time **d**, J - V characteristics of representative devices measured under $\sim 1360 \text{ W m}^{-2}$ illumination (AM0 approximation) **e**, MPPT results of encapsulated perovskite solar cells under $\sim 1000 \text{ W m}^{-2}$ equivalent 1-sun intensity, measured at $45 \pm 5 \text{ }^\circ\text{C}$ and $70 \pm 10\%$ relative humidity.”

In addition, we added clarification in line 382:

- “Maximum power point tracking (MPPT) analysis of the encapsulated solar cells—conducted under conditions of $45 \pm 5 \text{ }^\circ\text{C}$, $70 \pm 10\%$ relative humidity, and $\sim 1000 \text{ W m}^{-2}$ equivalent 1-sun illumination, revealed that DMSLA-based target devices remained almost unchanged, while the control samples lost 33% of their initial efficiency after 500 hours (Figure 4d).”

Additionally, the reviewer raised a valid point regarding reporting device performance under AM0 conditions. Using the same solar simulator, we increased the light intensity to 1.36 sun to approximate AM0 conditions. We have added IV curves and the statistical distribution of the corresponding devices at 1360 W/m^2 . The devices obtained higher J_{SC} and V_{OC} at AM0 compared to those at AM1.5 spectrum.

Supplementary Figure 28. J - V curves and the corresponding device parameters of the devices at 1360 W/m^2 (AM0 approximation).

Supplementary Figure 29. Statistical distribution of a) V_{OC} , b) PCE, c) FF, and (d) J_{sc} values of the devices at 1360 W/m^2 . Each condition had 10 devices.

R1 - Comment 4.

The paragraph beginning at Line 72 with "During thermal cycling..." should cite other perovskite investigations into impact of thermal cycling including Krause et al <https://doi.org/10.1002/solr.202300468>

R1 - Response 4

We thank the reviewer for this valuable suggestion. Additional works related to thermal cycling tests, including the study by Krause et al., have been added to the revised manuscript.

- Lines 55-57: "Repeated temperature cycling induces volumetric expansion and contraction within the solar cell stack, giving rise to mechanical fatigue that can ultimately lead to delamination, crack formation, or failure at weakly bonded interfaces.^{2,3}"

[3] Krause, T.S., VanSant, K.T., Lininger, A., Crowley, K., Peshek, T.J. and McMillon-Brown, L. (2023), Thermal Performance of Perovskite-Based Photovoltaics for Operation in Low Earth Orbit. *Sol. RRL*, 7: 2300468. <https://doi.org/10.1002/solr.202300468>

R1 - Comment 5.

The claim of the first systematic study of extreme low temperature cycling experiments is invalid and should be edited or removed (conclusion line 377) to acknowledge the similar systematic work published by Chen et al <https://doi.org/10.1021/acsenergylett.4c00988> and Li et al <https://doi.org/10.1002/aenm.202202887> The first extreme low temperature cycling experiments were probably conducted by Delmas et al who did thermal cycling analysis on the International Space Station <https://doi.org/10.1002/aenm.202203920>. Authors should read these works and consider citing all the above.

R1 - Response 5

We thank the reviewer for flagging this point. After reviewing these works, the related citations have been revisited and revised accordingly. The corresponding paragraph in the manuscript has been revised, and the claim on the first systematic study of extreme temperature cycling experiments is removed.

- In lines 70-27: “Recent systematic studies have also examined perovskite solar cell performance under extreme temperature cycling and in-orbit conditions, further highlighting the importance of understanding thermal fatigue for space deployment.¹⁰⁻¹²”

[10] Chen, M. et al. Stress Engineering for Mitigating Thermal Cycling Fatigue in Perovskite Photovoltaics. *ACS Energy Lett.* 9, 2582–2589 (2024).

[11] Delmas, W. et al. Evaluation of Hybrid Perovskite Prototypes After 10-Month Space Flight on the International Space Station. *Advanced Energy Materials* 13, 2203920 (2023).

[12] Li, G. et al. Structure and Performance Evolution of Perovskite Solar Cells under Extreme Temperatures. *Advanced Energy Materials* 12, 2202887 (2022).

R1 - Comment 6.

Supplemental information line 227 in “pull-off tests” section second sentence – which surface was first coated with a thin layer of PMMA? Please confirm in text.

R1 - Response 6.

We thank the reviewer for giving opportunity for further clarification. The perovskite surface was first coated with a thin layer of PMMA before dolly attachment to protect the film from moisture and prevent direct contact with the epoxy adhesive. This information has now been explicitly clarified in the Supplementary Information.

- Lines 230–232: “The perovskite surface was first coated with a thin layer of PMMA to protect the underlying perovskite layer from moisture and to prevent direct contact with the epoxy adhesive used in subsequent preparation steps.”

Reviewer #2

Remarks to the Author:

While the manuscript presents a detailed technical investigation into the adhesion properties between the perovskite and HTL (SAM) in the presence of LA and its derivatives, as well as the full device performance variation under thermal cycling test, I do not find the novelty sufficiently strong for acceptance in Nature Communications. Poly-LA has already been reported (ref. 17) to serve as a passivating agent that simultaneously optimises interfacial contact, enhances perovskite crystallinity, and maintains robust mechanical properties. In this manuscript, the authors report that LA and its derivatives also form polymers after annealing around the grain boundaries, which is conceptually very similar to the previously published work.

The thermal cycling investigation was conducted between $-80\text{ }^{\circ}\text{C}$ and $+80\text{ }^{\circ}\text{C}$, and the results show a positive effect after incorporating DMSLA into the HTL. However, the title suggests that the work was carried out under extreme LEO (Low Earth Orbit) conditions, which is somewhat misleading. Thermal vacuum cycling tests should be included if the authors wish to retain the current title. In the conclusion, the authors explicitly mention that thermal vacuum testing will be the next step.

Response:

We thank the reviewer for providing valuable feedback on our manuscript and for recognizing the depth of our analysis of the substrate/perovskite interface. We understand the

reviewer's concern regarding the novelty of our work, and we appreciate the opportunity to elaborate further. The novelty of our study lies in several key aspects: (i) Molecular innovation: It is true that poly-LA has been reported previously, which is not surprising given that during our material screening, we identified it as a promising candidate due to its ability to undergo ring-opening reactions and form a polymeric network. However, this was not the main focus of our work. Instead, we prescreened this molecular family, using dihydrolipoic acid (DHLLA) and synthesizing a sulfonium-based cation (DMSLA) specifically. Our goal was to achieve stronger interactions with the perovskite lattice—one of the weakest interfaces in these devices. (ii) Process novelty: Another innovative aspect of our study is the incorporation of the linker molecules directly into the SAM solution. This approach simplifies the overall process and enables interface linking in a single step, while simultaneously improving the mechanical adhesion at this critical interface. (iii) Thermal-fatigue-focused reliability testing: A further element of novelty lies in the focus on thermomechanical fatigue under repeated extreme temperature cycling as a stand-alone reliability physics problem. Rather than aiming to qualify devices for a specific application environment, we deliberately employed a wide temperature window ($-80\text{ }^{\circ}\text{C}$ to $+80\text{ }^{\circ}\text{C}$) and realistic heating and cooling rates to induce severe thermally driven mechanical stress within the multilayer device stack, using space-compatible encapsulated (DOWSIL 93-500) cells.^[R1,R2] While previous studies have made important contributions, most do not replicate the heating and cooling rates corresponding to the targeted applications. We have summarized the literature, including our work, doing extreme thermal cycling either in terms of range or heating/cooling rate in **Supplementary Information Table 2**. In summary, our work differentiates itself from prior studies by combining controlled extreme temperature cycling with realistic heating and cooling rates, robust device encapsulation, and the unique introduction and synthesis of the DMSLA linker at the HTL/perovskite interface to directly address thermomechanical fatigue mechanisms.

Regarding the title, we have revised our manuscript title from “Perovskite Solar Cells with Enhanced Thermal Fatigue Resistance in Low-Earth Orbit Space Extremes” to “Perovskite Solar Cells with Enhanced Thermal Fatigue Resistance **under Extreme Temperature Cycling**.”

Also, we must note that our current laboratory facility does not allow performing thermo-vacuum cycling tests. We hope to have this capability in the near future.

We hope that our explanation addresses the reviewer's concerns regarding the novelty of our work. However, if needed, we are happy to provide further clarification.

[R1] Bulut, M. Thermal design, analysis, and testing of the first Turkish 3U communication CubeSat in low earth orbit. *J. Therm. Anal. Calorim.* 143, 4341–4353 (2021).

[R2] Lamb, D. A., Irvine, S. J. C., Baker, M. A., Underwood, C. I. & Mardhani, S. Thin film cadmium telluride solar cells on ultra-thin glass in low earth orbit—3 years of performance data on the AlSat-1N CubeSat mission. *Prog. Photovolt. Res. Appl.* 29, 1000–1007 (2021).

R2 - Comment 1.

One of the key advantages of perovskite materials for space applications is their high specific power. This should be emphasized in the introduction (lines 48–49).

R2 - Response 1.

We thank the reviewer for this valuable suggestion. It has been added into the related part in the introduction.

- Lines 47–49: “Metal halide perovskite solar cells (PSCs) have attracted significant attention due to their high power conversion efficiencies, low fabrication costs, and compatibility with lightweight device architectures.¹”

R2 - Comment 2.

Line 51: If I understand correctly, the authors state that III–V solar cells exhibit a short operational lifetime. However, III–V solar cells are widely used in current space applications. It would be helpful to clarify which type of commercial solar cells the authors are comparing with—are they referring to Si-based solar cells?

R2 - Response 2.

We agree that our original phrasing could be interpreted as implying that III–V solar cells have short operational lifetimes, which is not accurate. Our intention was to emphasize their high cost and manufacturing complexity, rather than poor reliability.

- Lines 49–52: “They present a potential alternative to conventional III–V-based photovoltaic technologies, which are well known for their robustness under extreme

operating conditions but are limited by high material and fabrication costs and relatively short operational lifetimes.²”

R2 - Comment 3.

Lines 91–92: There is some confusion regarding the substrate and perovskite description. Does the term "substrate" here refer to TCE without any charge transport layer deposited? In the following sentences (lines 92–94), were all the mentioned methods performed on ITO substrates or on non-TCE substrates? This section should be rewritten for better clarity.

R2 - Response 3.

We thank the reviewer for pointing out this potential ambiguity. In the revised manuscript, we clarified that “substrate” refers to bare ITO-coated glass before any charge transport or perovskite layers were deposited. The text was rewritten to eliminate ambiguity.

- Lines 96–102: “To improve adhesion at the interface between the transparent conductive oxide (TCO) and the perovskite layer, several surface-engineering strategies have been developed. Reported approaches include increasing surface hydroxylation by replacing crystalline TCOs with amorphous ones,^{21,22} removing terminal hydroxyls and hydrolysis byproducts via combined HF and UV–ozone treatments,²³ introducing hetero-chiral linker molecules,²⁴ and employing SAMs such as iodine-terminated carbazole derivatives or bifunctional thiol–carboxylic acid systems with varying alkyl chain lengths.²⁵”

R2 - Comment 4.

Lines 103–108: The ring-opening of LA is mentioned in the text, but the molecular structure of LA is not shown in this part of the section, making it difficult for readers to follow until reading Fig. 2a. It is suggested to either include the LA structure in the Supporting Information (around Figs. S1–S3) or refer to Fig. 2a directly in the text.

R2 - Response 4.

We thank the reviewer for raising this. In lines 103–108, the lipoic acid (LA) molecule mentioned in the context of the ring-opening reaction refers to its molecular structure, which is depicted in Fig. 2a. To make it easier for readers to follow the discussion on the ring-opening polymerization of LA, we have now added a direct reference to Fig. 2a, where the molecular structure of LA is presented.

- Lines 107–109: “To promote inter-grain connectivity, LA was incorporated into the perovskite precursor solution with the expectation that it could undergo in situ polymerization during crystallization (the molecular structure of LA is shown in Fig. 2a).”

R2 - Comment 5.

Please add "(DHLLA)" after "1H NMR of 6,8-bis(methylthio)octanoic acid" in Fig. S2 caption, consistent with Fig. S3.

R2 - Response 5.

We thank the reviewer for this comment. The molecule is referred to as “1H NMR of 6,8-bis(methylthio)octanoic acid” in the caption of Fig. S2 corresponds to the intermediate structure in the synthetic route described under the title “Supplementary Figure 1: Synthetic route for the preparation of DMSLA,” and it is not DHLLA. Only the abbreviations for LA and its derivatives, DHLLA and DMSLA, have been used; no other abbreviations were introduced.

R2 - Comment 6.

It would be helpful to label the atom numbers in the chemical structures and indicate the corresponding numbers in the 1H NMR spectra shown in Figs. S1 and S2.

R2 - Response 6.

We thank the reviewer for the reminder. We have added the atom numbers in the chemical structures and indicated the corresponding numbers in the 1H-NMR spectra in our revised version.

Supplementary Figure 2: ¹H NMR of 6,8-bis(methylthio)octanoic acid

R2 - Comment 7.

It is suggested to introduce the full chemical name of LA earlier in the text, before its first appearance in Section 2.2 (around line 175).

R2 - Response 7.

We thank the reviewer for this helpful suggestion. As recommended, the full chemical name of LA has been moved from line 183 to line 105.

- Lines 104-107: “In this work, we used α -lipoic acid (LA, 5-(1,2-dithiolan-3-yl)pentanoic acid) and functionalized it further as part of our two-step reinforcement strategy, and the synthetic procedures are provided in **Supplementary Figure 1**, while the NMR analyses are shown in **Supplementary Figures 2–4**.”

R2 - Comment 8.

Please verify the schematic structure of DMSLA in Fig. 2a; it appears incorrect. There seems to be an extra carbon atom attached to one of the sulphur atoms.

R2 - Response 8.

We thank the reviewer for pointing this out. The molecular structure of DMSLA in Fig. 2a has been carefully re-examined, and the incorrect fragment has been corrected. The revised and validated 2D structure has now been included in the manuscript to ensure clarity and accurate representation.

Figure 2. Mechanical analysis of substrate-perovskite interfaces. a, Schematic representation of the HTL contact formed by 4PADCB together with LA, DHLA, and DMSLA at the ITO/perovskite interface. b, Sketch for the test setup used for the pull-off test.

c, Maximum adhesion strength, and **d**, stress values, which are determined from the highest value in each of the four groups. **e**, The interaction energies between perovskite and the linker molecules, calculated using a computer-based DFT approach. **f**, Interactions between perovskite and the molecules.

R2 - Comment 9.

Line 192: The meaning of this sentence is unclear. Does the weak adhesion observed for LA and DHLA samples result from imperfections in sample preparation?

R2 - Response 9.

We thank the reviewer for this comment. The wide distribution of adhesion strength observed in Figure 2d appears across all conditions, including control and DMSLA samples. This spread originates from the intrinsic sensitivity of the pull-off test to minor preparation variations, such as small-angle misalignments or slight differences in epoxy layer thickness, which can influence the measured stress. To ensure the reliability of our data, we measured two independent batches with five samples for each condition, confirming that the observed variability reflects typical experimental dispersion rather than material-specific issues.

The corresponding paragraph in the manuscript has been revised as follows:

- “We examined the adhesion strength of SAM-linker contacts at the ITO/perovskite interface through pull-off testing. Indeed, as such systems are known to have intrinsically very low toughness, it is important to inhibit any initiation of delamination between the layers. Consequently, the adhesion strength serves as a direct indicator of mechanical integrity. For this, a stack consisting of glass/ITO/4PADCB-linkers/perovskite/PMMA was prepared, and a dolly was attached to the top surface using an adhesive. PMMA served as an interlayer to protect the perovskite from epoxy damage and improve glue adhesion, as shown in **Figure 2b**. From the resulting load–displacement graphs, we extracted the interfacial adhesion strength, which increased from 3.61 MPa for the control to 4.89 MPa for DMSLA-treated samples (**Figure 2c**). Pull-off tests showed a wide distribution of adhesion strength across all samples, likely arising from small preparation imperfections such as slight misalignment angles or adhesive layer thickness differences, yet DMSLA-based samples consistently showed the highest average strength. The resulting stress distributions follow the same trend, indicating that

DMSLA-treated interfaces require greater force to induce interfacial failure (Figure 2d).”

R2 - Comment 10.

In the caption of Fig. S10, please clarify how the authors ensured that LA molecules remained confined within the grain boundaries and were not exposed on the surface.

R2 - Response 10.

We thank the reviewer for this comment. While PF-QNM primarily probes the near-surface mechanical response, the localized stiffness enhancement and mechanical contrast observed specifically along grain boundaries indicate that LA or its polymerized form remains within these interfacial regions after film formation. The preservation of grain-boundary morphology and the absence of a uniform surface softening further support that LA is not exclusively segregated to the surface. We have slightly revised the caption of Supplementary Fig. S10 to clarify this point. The revised caption now reads:

- “Supplementary Figure 10: Peak Force QNM images from the top of control and target (LA) samples. Red areas in the top contact represent the mask which distinguishes between grains and grain boundaries where the tip apex curvature can be bigger than the grain boundary depth and width which could lead to overestimation of the mechanical properties in addition to the enhanced properties due to the LA. The pronounced mechanical contrast and localized stiffness enhancement along the grain boundaries indicate that LA or its polymerized form likely remains within these interfacial regions after film formation, contributing to improved boundary adhesion. We note that on certain grain facets, a distinct region with lower stiffness is observed, which we attribute to LA molecules bonded to the perovskite being exposed at the surface. In contrast, the brighter areas correspond to unmodified perovskite, with stiffness values matching those of the unprocessed sample.”

To maintain consistency with the revised description and the updated caption of Supplementary Figure S10, we have made minor wording changes in the caption of Figure 1:

- **“Figure 1. Nanoscale mechanical behavior of perovskite film surface and grain boundaries.** a, A sketch demonstrating the thermal behaviour of perovskite polycrystalline structures with (iv) and without poly(LA) (i-iii) at the grain boundaries. b, Thermal polymerization of LA and its multifunctional role at grain

boundaries, facilitating defect passivation (1) and hydrogen bonding interactions (2) between polymer chains. c, A sketch for the working mechanism of the surface-mapping tip-based nano-mechanical analysis technique. Numbers 1 and 2 indicate the perovskite grains and the poly(LA)-rich grain boundaries, respectively. d, Adhesion forces were obtained from the perovskite surface at the HTL contact region using a tip-based nanomechanical surface-mapping technique, alongside topographical imaging. The values shown in the yellow boxes are the average adhesion forces extracted from each corresponding image.”

R2 - Comment 11.

In Table S1, the data for the perovskite with LA are missing.

R2 - Response 11.

Thanks for the reviewer’s reminder. We have added the data for the perovskite with LA in our revised version as follows:

Supplementary Figure 17: a) UPS spectra of a) 4PADCB, b) 4PADCB+LA, c) 4PADCB+DHLA, d) 4PADCB+DMSLA, e) perovskite, and f) perovskite+LA films deposited on ITO substrate, respectively.

Supplementary Table 1: Electronic parameters of SAM (control), its modified forms, perovskite films w/wo LA from the UPS spectra.

Sample	$E_{\text{cutoff}}(\text{eV})$	$E_{\text{onset}}(\text{eV})$	VBM(eV)	Work function(eV)
4PADCB	16.44	0.80	-5.47	-4.76
4PADCB + LA	16.47	0.83	-5.56	-4.73
4PADCB + DHLA	16.48	0.85	-5.57	-4.72
4PADCB + DMSLA	16.56	0.89	-5.53	-4.64
Perovskite	16.6	1.18	-5.78	-4.6
Perovskite +LA	16.48	1.14	-5.86	-4.72

R2 - Comment 12.

Line 297: Please verify whether the energy offset is 0.02 eV rather than 0.04 eV.

R2 - Response 12.

We thank the reviewer for drawing our attention to this point. This helped us to revise our results and correct the analysis have been done. In our analysis, the relevant quantity for hole extraction is the valence-band offset, i.e. the difference in the $E_{\text{F}}\text{-VBM}$ separation between the SAM-modified ITO and the perovskite+LA absorber, as obtained from the UPS E_{onset} values in Supplementary Table 1. For perovskite+LA, the VBM lies 1.14 eV below E_{n} ($E_{\text{onset}} = 1.14$ eV), whereas for the different SAM stacks the corresponding values are 0.80 eV (4PADCB), 0.83 eV (4PADCB+LA), 0.85 eV (4PADCB+DHLA), and 0.89 eV (4PADCB+DMSLA). The valence-band offset with perovskite+LA is therefore 0.25 eV for 4PADCB+DMSLA, which is the smallest among the modified SAMs. The energy offset in the relevant line, calculated based on the figure shown in Fig. 3, has been corrected to 0.25 eV as follows:

- Lines 309-312: “As can be seen from the energy level diagram derived from the UPS data presented in Figure 3d—which is critical for charge transfer at the perovskite/SAM interface—the combination of 4PADCB with DMSLA results in the smallest energy offset (0.25 eV) between the valence band maximum (VBM) of the modified ITO and that of the LA doped perovskite.”

R2 - Comment 13.

How was the bandgap calculated? If it was derived from UV–vis spectra, please provide the corresponding absorption spectra with a proper baseline.

R2 - Response 13.

We thank the reviewer for this comment and for noticing the need for clarification. In the original version of the Supplementary Information, we incorrectly stated that the measurements were performed “in transmittance mode.” In fact, the optical bandgap estimation was based on combined transmittance and reflectance measurements obtained using a PerkinElmer Lambda 1050 spectrometer on thin-film samples.

The absorption (A) was calculated according to $A = 1 - T - R$, where T and R represent the measured transmittance and reflectance, respectively. This approach ensures a baseline-corrected absorption spectrum, as it inherently accounts for substrate transmission and reflection losses, thereby avoiding artificial offsets.

Tauc plots derived from these corrected absorption spectra were then used to estimate the optical bandgaps. As shown in Supplementary Figure 22a–b, the spectra exhibit nearly identical profiles for all samples, with a slight red shift in the DMSLA-treated film. The extracted bandgaps were approximately 1.55 eV for all conditions, confirming negligible variation.

We have corrected the description in the Supplementary Information to accurately reflect the combined transmittance–reflectance method as follows:

“UV–Vis optical measurements were carried out using a PerkinElmer Lambda 1050 spectrometer equipped with an integrating sphere. Both transmittance (T) and reflectance (R) spectra were recorded on thin-film samples deposited on glass substrates. The absorbance (A) of the films was then calculated according to the relation $A = 1 - T - R$, which inherently corrects for baseline offsets by accounting for substrate transmission and surface reflection losses. The resulting spectra were used to construct Tauc plots for bandgap estimation, as shown in **Supplementary Figure 22a–b**. All films exhibited similar absorption characteristics with no major spectral shifts, although a slight red shift was observed for the DMSLA-treated sample. The estimated optical bandgaps for all conditions were approximately 1.55 eV, confirming negligible variation among the samples (**Supplementary Figure 22-b**).”

R2 - Comment 14.

Page 12, line 328: Please explain how the device temperature was maintained at $-80\text{ }^{\circ}\text{C}$ when immersed in liquid N_2 .

R2 - Response 14.

We appreciate the reviewer's request for clarification. The thermal cycling experiments were performed using a custom-designed closed-lid setup rather than direct immersion in liquid nitrogen. The complete configuration and workflow are illustrated in Supplementary Figure S26. The system was engineered to achieve reproducible and controlled thermal cycling between $-80\text{ }^{\circ}\text{C}$ and $+80\text{ }^{\circ}\text{C}$.

To prevent frost formation and ensure dry ambient conditions, dry air was circulated within the sealed lid, and cycling was initiated only after the DHT22 temperature–humidity sensor indicated 0% relative humidity (0–100% RH range, $\pm 2\%$ accuracy, $\pm 5\%$ max deviation). Dry air circulation was maintained throughout the experiment to ensure a consistent, moisture-free environment.

The temperature control was achieved through aluminium plates of optimized thickness, designed according to heat-transfer calculations to match the desired heating and cooling rates. The plates were placed beneath the samples inside the chamber, providing efficient thermal conduction while allowing a total cycle duration of ~ 90 minutes ($-80\text{ }^{\circ}\text{C}$ to $+80\text{ }^{\circ}\text{C}$ and back). The temperature was monitored directly from the sample surface using a K-type thermocouple attached to the device substrate, ensuring accurate measurement of the actual thermal load experienced by the cells.

The following modifications have been implemented in **Section 2.4, “Thermal Fatigue Stability Analysis”**:

- “We performed the thermal fatigue test using a custom-built close-lid setup, as standard climate chambers could not reach the required $-80\text{ }^{\circ}\text{C}$ cryogenic range, which induces the highest stress at the perovskite–substrate interface. The samples were cycled between $-80\text{ }^{\circ}\text{C}$ and $+80\text{ }^{\circ}\text{C}$ in a stainless steel container, with controlled heating and cooling rates of $+3.40$ and $-3.80\text{ }^{\circ}\text{C min}^{-1}$, respectively, and a total cycle duration of ~ 90 minutes under dry ambient conditions (Figures 4a, b). A schematic and further explanation of the experimental setup are provided in Supplementary Figure 26.”

More detailed information on the thermal cycling procedure is presented in the Supporting Information under Methods titled as “**Thermal Cycling Tests**”:

- “Thermal cycling experiments were performed using a custom-built closed-lid setup designed to achieve reproducible and well-controlled temperature cycling between –80 °C and +80 °C. To prevent frost formation and maintain dry ambient conditions, dry air was circulated within the sealed chamber, and cycling was initiated only after the integrated DHT22 temperature–humidity sensor indicated 0% relative humidity. The sample temperature was monitored directly from the device surface using a K-type thermocouple placed in contact with the substrate to accurately capture the thermal load experienced by the cells. A schematic representation of the thermal fatigue experimental setup is provided in **Supplementary Figure 26.**”

R2 - Comment 15.

The control devices retained 79% of their initial efficiency after 16 cycles, DMSLA showed 84%, and DHLA showed 72%. These differences appear minor (within $\pm 5\%$ deviation). How many devices were measured to draw this conclusion?

R2 - Response 15.

The custom-built thermal cycling setup allows testing of up to four samples simultaneously. Because each thermal cycle lasts approximately 90 minutes and the process requires careful manual operation for controlled heating and cooling, it was only feasible to test one representative device per condition in this initial proof-of-concept study. For each device, the solar cell contains six independent pixels, and each reported value represents the median of six J – V measurements, providing internal statistical variation within the same device.

To further clarify the variability, we have now included the full statistical distributions of all photovoltaic parameters (PCE, V_{oc} , J_{sc} , and FF) in the Supporting Information for all three cycling durations (2 min, 15 min, and 90 min). We also revised the caption of Figure 4 to explicitly state that each data point in the main text corresponds to the median value extracted from six pixels of a single device.

These additions make the basis of our conclusion transparent and highlight that, despite the small differences in median PCE retention, the relative stability trends are consistently reproduced across the individual pixel data.

Supplementary Figure 30. Statistical distribution of photovoltaic parameters under 90-minute thermal-cycling conditions for devices fabricated at LMU Munich. Boxplots show the normalized (a) PCE, (b) V_{OC} , (c) J_{SC} , and (d) FF for control, LA-treated, DHLA-treated, and DMSLA-treated perovskite solar cells measured after encapsulation (baseline), and after 1, 3, 8, and 16 thermal cycles. Each data point corresponds to an individual device pixel (six pixels per device). Parameter values are normalized to those measured at cycle 0 for each device. Darker-colored data points correspond to the forward $J-V$ scans, while lighter-colored data points correspond to the reverse $J-V$ scans.

R2 - Comment 16.

Although the variation is small, why does the VOC curve in Fig. 4a for DMSLA show a decrease after five cycles, followed by a gradual increase after 16 cycles?

R2 - Response 16.

We thank the reviewer for this observation. The transient decrease and subsequent recovery of V_{OC} in the DMSLA-treated devices likely reflect interfacial relaxation and stabilization processes occurring during repeated thermal cycling. We hypothesize that in the early cycles, exposure to alternating extreme temperatures may cause temporary structural or dipolar reorganization within the self-assembled DMSLA layer or at the perovskite/HTL interface, slightly disturbing the built-in potential and resulting in a lower V_{OC} . Upon further cycling, the system appears to re-stabilize, possibly through thermally assisted reordering of interfacial dipoles or partial defect healing, leading to the observed recovery in V_{OC} .

We emphasize that this explanation is hypothesis-based. While no prior studies to our knowledge report the specific V_{OC} decrease-then-recovery behavior under repeated thermal cycling in SAM-modified PSCs, several recent works have demonstrated that interfacial dipoles and SAM/bilayer interface engineering can influence V_{OC} and interface stability^[R3,R4]. But further studies (e.g., in-situ spectroscopic or electrical characterization during cycling) would be required to confirm the underlying mechanism under thermal cycling conditions.

[R3] Dong, B., Wei, M., Li, Y. et al. Self-assembled bilayer for perovskite solar cells with improved tolerance against thermal stresses. *Nat Energy* 10, 342–353 (2025).

<https://doi.org/10.1038/s41560-024-01689-2>

[R4] Peng, Y., Chen, Y., Zhou, J. et al. Enlarging moment and regulating orientation of buried interfacial dipole for efficient inverted perovskite solar cells. *Nat Commun* 16, 1252 (2025). <https://doi.org/10.1038/s41467-024-55653-5>

We have added this discussion under Section 2.4, Thermal Fatigue Stability Analysis, as follows:

- “The transient decrease and subsequent recovery in V_{OC} observed for DMSLA-treated devices may arise from interfacial relaxation during repeated thermal cycling. Such behavior could reflect temporary perturbation of the interfacial energetics followed by stabilization upon continued cycling. Further studies would be required to determine the exact mechanism.”

R2 - Comment 17.

In Materials section in SI, BCP and LiF were mentioned twice and also sourced from different suppliers. Please indicate the reason.

R2 - Response 17.

We thank the reviewer for this comment. In this study, the perovskite-based solar cells were fabricated in two separate laboratories (line 126 at LMU Munich and line 143 at Tianjin University). Each group listed the chemicals obtained from different suppliers because they procured the reagents independently for their respective experiments.

R2 - Comment 18.

Dyesol company no longer exists. Please confirm FA and MA were sourced from GreatCell Solar.

R2 - Response 18.

Thanks for the reviewer's reminder. The information regarding the source of the organic halide salts has been updated in the Materials and Methods section of the Supplementary Information as follows:

- “Cesium iodide (CsI, 99.5%), Lead (II) iodide (PbI₂, 99.99%), Lead(II) Chloride (PbCl₂, 99.0%), α -lipoic acid (LA) and dihydrolipoic acid (DHLLA) were sourced from TCI. N, N-dimethylformamide (DMF, 99.8%), dimethyl sulfoxide (DMSO, 99.8%), Isopropanol (IPA, 99.8%), Chlorobenzene (CB, 99.9%), and ethanol (EtOH, 99.8%) were purchased from Sigma Aldrich. Methylammonium iodide (MAI, 99.5%), Methylammonium bromide (MABr, 99.5%) and formamidinium iodide (FAI, 99.5%) were purchased from **GreatCell Solar Materials**. Fullerene (C₆₀, 99.5%) and (4-(7H-dibenzo[c,g]carbazol-7-yl)butyl)phosphonic acid (4PADCB) were purchased from Lumtec. Indium tin oxide (ITO, YXKJGI-0006, 15 Ω /sq), and Bathocuproine (BCP) were purchased from Yingkou Advanced Election Technology Co., Ltd. Lithium fluoride (LiF) and Ethanediamine dihydroiodide (EDAI₂) were purchased from Xi'an Polymer Light Technology, China. All chemicals were used as it is without further purification.”

R2 - Comment 19.

Please provide more details in the caption of Fig. S24. Do the dark blue and light blue boxes correspond to the reverse and forward scans, respectively? The same clarification is suggested for the colours used in Figs. S24 and S25.

R2 - Response 19.

We thank the reviewer for pointing out the need for clearer colour labelling in the $J-V$ scan figures. In the revised version of the Supporting Information, Supplementary Figure S24 has been updated to include explicit labels indicating which colours correspond to the forward and reverse scans.

Supplementary Figure 24: Statistical distribution of a) PCE (%), and b) J_{sc} values for the solar cell devices fabricated at LMU Munich.

Likewise, the caption of **Supplementary Figure S25** has been revised to clarify that the darker-coloured curves represent the forward scans, while the lighter-coloured curves represent the reverse scans.

“Supplementary Figure 25: Statistical distribution of a) V_{oc} , b) PCE (%), c) FF (%), and (d) J_{sc} values for the solar cell devices fabricated at Tianjin University. **Darker-coloured data**

points correspond to the forward J - V scans, while lighter-coloured data points correspond to the reverse J - V scans.”

Reviewer #3

Remarks to the Author:

The manuscript addresses a crucial aspect for the employment of perovskite solar cells in Low Earth Orbit (LEO): their resilience to thermal fatigue. The authors propose an innovative two-step approach, based on the use of α -lipoic acid and its derivatives, to improve perovskite film cohesion and mitigate delamination. To validate this strategy, they employed several experimental techniques such as XRD, FTIR, AFM, SEM, GIWAXS, which confirmed a significant improvement in both internal grain cohesion and interfacial adhesion. Thermal fatigue tests, performed with a custom-built setup and a thermal cycling protocol ranging from -80°C to $+80^{\circ}\text{C}$ for 16 cycles (equivalent to one day of operation), indicate encouraging stability. The best performing device achieved an efficiency of 26% and retained 84% of its initial performance after the thermal fatigue test.

The manuscript is well written, logically structured, and contextualized within the literature. The experimental work is carefully executed, and the short-term data are convincing within the tested scope. The analytical framework is solid, and the interpretation of the available data is careful.

However, while the proposed strategy is novel and the results are encouraging, in our opinion, the current experimental validation does not provide sufficient evidence to substantiate claims of aerospace suitability for the following reasons:

Response:

We thank the reviewer for the thorough and encouraging assessment of our manuscript. We appreciate the positive evaluation of the experimental design, analytical framework, and short-term thermal-fatigue data, as well as the recognition of the novelty of our two-step molecular reinforcement strategy and the careful execution of the structural, mechanical, and interfacial analyses.

We fully agree with the reviewer that the present experimental validation should not be interpreted as evidence of aerospace qualification. We recognize that the original framing of the manuscript may have unintentionally suggested a stronger link to aerospace readiness than warranted by the scope of the experiments. To address this concern, we have

comprehensively revised the manuscript to clearly reposition the study as a thermal-fatigue–focused reliability investigation, rather than an application-qualification or aerospace-readiness study.

In the revised version, we have removed or softened references to space, low-Earth orbit operation, and aerospace deployment throughout the title, abstract, introduction, results, and conclusion. The work is now explicitly framed as a proof-of-concept study that isolates thermomechanical degradation mechanisms under repeated extreme temperature cycling and demonstrates how targeted grain-boundary and interfacial engineering can improve resistance to such stress. We now explicitly state that additional environmental stressors—such as vacuum, ultraviolet radiation, ionizing radiation, and atomic oxygen—are beyond the scope of the present work and remain subjects for future investigation.

We believe that these revisions significantly improve the clarity, accuracy, and positioning of the manuscript, ensuring that the claims are fully aligned with the experimental evidence presented and directly addressing the reviewer’s concern regarding overinterpretation of aerospace suitability.

R3 - Comment 1.

The thermal cycling test (16 cycles) is far too limited to represent orbital conditions, which typically involve hundreds or thousands of cycles.

R3 - Response 1.

We thank the reviewer for raising this important point. Notably, since the device performance had already dropped below 90% of the initial PCE, it was not realistic to proceed with additional cycling measurements. From a technical perspective, under realistic cycling conditions (90 minutes), the experimental preparation and execution of the tests require approximately 10–12 working hours per 16 cycles. Conducting significantly longer experiments was deemed unsafe following consultation with the safety department, primarily due to the use of liquid nitrogen and the need for authorized personnel to refill the N₂ tanks. In addition, from a human health and safety standpoint, performing extended cycling experiments is not practical with our current manual setup. For these reasons, we regret that we are unable to provide data over hundreds of cycles at this stage. We appreciate your understanding.

However, we had been asking this valid question to ourselves and to investigate whether the number of cycles or the total thermal-exposure duration is the dominant factor in degradation, so we have done new experiments. We synthesized molecules again, fabricated new cells, encapsulated them, and performed two additional thermal-cycling experiments beyond the initial 90-minute cycles:

(i) a fast 2-minute cycle, and

(ii) a moderate 15-minute cycle,

each carried out for 16 cycles (equivalent number of cycles but different total thermal exposure). The idea behind this was to understand how feasible it is to perform a large number of cycles. As a side note, we do not currently have the capability in our lab to perform thermo-vacuum tests with extreme cycling conditions.

Across all conditions, the degradation was extremely small, as shown in the new contour plots and boxplot distributions (Supplementary Figures 30–33). For all devices—including the control group—PCE, V_{OC} , J_{SC} , and FF remained within ~1–5% of their initial values, with no systematic decline across cycle number for the 2-min and 15-min experiments. This is clearly visible in the boxplot distributions, where the parameter scatter remains tightly clustered around unity after 1, 3, 8, and 16 cycles.

In contrast, the 90-minute cycles produced measurable degradation in the control devices, consistent with our original report. When comparing the three experiments, a clear trend emerges: Thermal fatigue correlates far more strongly with the total time spent under thermal stress than with the number of cycles.

This is also reflected in our new contour maps, which show that degradation contours follow lines of constant total thermal-exposure time rather than constant cycle number. For example, the control devices begin to show a drop only when total exposure exceeds ~8–12 hours (e.g., 90-minute \times 16 cycles), while 2-minute and 15-minute cycling (total exposure <4 hours) show essentially no degradation.

This observation is fully consistent with the literature. As summarized in the new comparative table (Table R1), studies that report significant thermal-cycling degradation typically involve either:

very large temperature ranges (e.g., –143 to +157 °C),

vacuum conditions,

or long cumulative exposure times (hundreds to thousands of minutes),

as seen in the works of Lu et al. (2025), Yang et al. (2024), Bautista et al. (2022), and Chen et al. (2024). Conversely, studies with similar or greater cycle counts but shorter total thermal exposure (e.g., 20–30 minutes per cycle or rapid cycling) show minimal degradation even after 50–200 cycles (Li et al., 2022; Zhihao Li et al., 2024; Guixiang Li et al., 2023). Our findings align with this trend: the absolute number of transitions across 0 °C appears to be a weaker predictor of failure than the integrated thermomechanical load, which is governed by the total time spent at extreme temperatures and the duration of thermal gradients.

Therefore, our expanded dataset and cross-comparison with the literature demonstrate that thermal-cycling degradation is primarily governed by cumulative thermal-exposure time and the associated buildup of thermomechanical stress, rather than by the number of cycles alone. This finding highlights the importance of selecting cycle durations that realistically capture the temporal profile of temperature exposure, as longer dwell times at extreme temperatures impose greater mechanical strain and interfacial stress accumulation than rapid cycling with short exposure.

We have included these new datasets, contour plots, boxplots, and Supplementary Table 2 in the revised Supplementary Information.

Supplementary Figure 33. Contour maps showing the dependence of normalized photovoltaic parameters on cycle duration and number of thermal cycles for control perovskite solar cells. Panels display the normalized (a) PCE, (b) V_{oc} , (c) J_{sc} , and (d) FF across a matrix of cycle durations (2–90 minutes) and cycle counts (0–16). Star symbols indicate the experimentally tested conditions in this work (2-min, 15-min, and 90-min cycling, each for 16 cycles). White dashed lines represent contours of constant total thermal-exposure time. All parameters are normalized to their respective values at cycle 0. Contour maps are generated by interpolation of discrete measured data points. These plots reveal that degradation aligns more strongly with total thermal-exposure time rather than cycle count, with negligible changes observed for short-duration cycles and measurable performance drops appearing only for extended exposure (e.g., 90-min cycles).

Supplementary Figure 31. Statistical distribution of photovoltaic parameters under 2-minute rapid thermal-cycling conditions for devices fabricated at LMU Munich. Boxplots show the normalized (a) PCE, (b) V_{oc} , (c) J_{sc} , and (d) FF for control, LA-treated, DHLA-treated, and DMSLA-treated devices evaluated after 0, 1, 3, 8, and 16 cycles. Each point represents a single active pixel on the device. All photovoltaic parameters remain essentially unchanged throughout cycling, confirming that rapid temperature transitions with short exposure times impose negligible stress and further supporting that performance degradation is governed by cumulative thermal-exposure duration rather than the number of cycles. Darker-coloured data points correspond to the forward $J-V$ scans, while lighter-coloured data points correspond to the reverse $J-V$ scans.

Supplementary Figure 32. Statistical distribution of photovoltaic parameters under 15-minute thermal-cycling conditions for devices fabricated at LMU Munich. Boxplots show the normalized (a) PCE, (b) V_{oc} , (c) J_{sc} , and (d) FF for control, LA-treated, DHLA-treated, and DMSLA-treated solar cells measured after 0, 1, 3, 8, and 16 cycles. Each data point corresponds to an individual device pixel (six pixels per device). Values are normalized to the corresponding parameter measured at cycle 0 for each sample. All device groups indicate moderate thermomechanical degradation under moderate cycle duration and demonstrating that cycle count alone is not a sufficient predictor of fatigue when compared to 90-minute thermal-cycling conditions. Darker-coloured data points correspond to the forward $J-V$ scans, while lighter-coloured data points correspond to the reverse $J-V$ scans.

R3 - Comment 2.

The working conditions of solar cells in space are extremely harsh, involving not only thermal fluctuations but also high vacuum, atomic oxygen, and high-energy particle radiation (such as protons, electrons, neutrons, γ -rays). Although the authors briefly acknowledge the need for vacuum tests to better simulate the space environment, they do not consider the importance of radiation, leaving a fundamental gap in assessing material's long-term stability. Without such tests, the applicability of the solar cells for aerospace missions remains speculative.

R3 - Response 2.

We thank the reviewer for raising this important point regarding radiation effects. We agree that radiation exposure is a critical stress factor for photovoltaics intended for space deployment, and that radiation testing would be required for any claim related to aerospace qualification. We clarify, however, that the present study does not aim to establish space-grade performance or mission readiness. Instead, it focuses on isolating and understanding thermomechanical degradation mechanisms in perovskite solar cells under repeated extreme temperature cycling as a stand-alone accelerated stress condition.

At present, incorporating radiation exposure into a meaningful experimental framework for perovskite devices faces both practical and conceptual limitations. Access to suitable radiation facilities capable of controlled, device-level irradiation is limited, and standardized testing protocols for perovskite photovoltaics under ionizing radiation—including particle type, energy, and fluence—are still under active discussion within the community. Without such consensus, it remains challenging to unambiguously interpret radiation-induced degradation or to assess the role of molecular additives in mitigating these effects.

Moreover, meaningful evaluation of radiation effects would ideally require combined multi-stressor testing—integrating radiation exposure with thermal cycling and vacuum conditions—since isolated radiation tests alone do not constitute application-level qualification. To the best of our knowledge, experimental platforms capable of performing such coupled stress testing for perovskite solar cells are not yet widely available.

In light of these considerations, radiation testing is beyond the scope of the present work. To avoid overinterpretation, we have revised the manuscript to remove any implication of aerospace qualification and now frame the results strictly in terms of thermal fatigue resistance and interface stabilization under extreme temperature cycling. We believe that establishing clear structure–property–stability relationships under well-defined thermal stress

conditions is a necessary prerequisite for future multi-stressor investigations, including radiation exposure. To build on this study, our ongoing efforts involve the development of a thermo-vacuum cycling platform that might enable combined thermal and vacuum testing under space-relevant conditions with AMO irradiation. We must note that there is no standard and commercially available system for this.

To clarify the scope and temper our claims, we have revised the manuscript title, abstract, introduction, results, and conclusion accordingly.

Title:

We have revised our manuscript title from “**Perovskite Solar Cells with Enhanced Thermal Fatigue Resistance in Low-Earth Orbit Space Extremes**” to “**Perovskite Solar Cells with Enhanced Thermal Fatigue Resistance under Extreme Temperature Cycling.**”

Abstract:

We have comprehensively revised the Abstract to remove references to space, LEO, radiation tolerance, and aerospace deployment. The revised Abstract now clearly positions the study as a thermal-fatigue-focused reliability investigation, emphasizing extreme temperature cycling as an accelerated stress-testing protocol and highlighting thermomechanical degradation mechanisms and interface engineering strategies without reference to space environments or mission scenarios.

- “Perovskite solar cells (PSCs) combine high power density with low-cost manufacturing, making them attractive for applications requiring lightweight and mechanically compliant photovoltaic technologies. However, their long-term stability under repeated extreme temperature cycling—a critical driver of thermomechanical degradation in layered device stacks—remains insufficiently understood. Here, we investigate the thermal fatigue behaviour of PSCs subjected to cyclic temperature variations between $-80\text{ }^{\circ}\text{C}$ and $+80\text{ }^{\circ}\text{C}$ as an accelerated stress-testing protocol. Mismatched thermal expansion between the perovskite absorber and the glass substrate induces biaxial tensile strain, particularly during rapid temperature transitions, leading to degradation at both the substrate–perovskite interface and grain boundaries within the perovskite film. To address this, we introduce a co-additive SAM strategy using α -lipoic acid (LA), dihydrolipoic acid (DHLA), and a sulfonium-based cation (DMSLA) to enhance substrate–perovskite interfacial adhesion, while in

situ polymerisation of LA during annealing strengthens grain-boundary cohesion. This dual reinforcement approach improves both mechanical robustness and optoelectronic performance, yielding stabilised power conversion efficiencies of up to 26% under AM1.5G illumination. Devices incorporating both modifications retain 84% of their initial performance after 16 extreme temperature cycles. Our experiments further reveal that the duration of thermal cycling is more critical than the number of cycles, with most degradation occurring during the initial cycles. These results underscore the importance of targeted interface engineering and grain-boundary reinforcement for improving the thermal fatigue resistance of PSCs under harsh temperature cycling conditions.”

Introduction:

We have also substantially revised the Introduction to address the reviewer’s concerns regarding application-driven overinterpretation. In particular, the opening sections of the Introduction have been restructured to remove emphasis on space deployment, low-Earth orbit operation, and aerospace qualification. The revised text now focuses on the fundamental mechanisms of thermomechanical degradation in layered perovskite device stacks under repeated extreme temperature cycling, thereby clearly positioning the study as a reliability-physics investigation rather than an application-qualification study.

- Lines 46–79: “Metal halide perovskite solar cells (PSCs) have attracted significant attention due to their high power conversion efficiencies, low fabrication costs, and compatibility with lightweight device architectures.¹ These attributes make PSCs attractive for applications in which high specific power and mechanical compliance are required. They present a potential alternative to conventional III–V-based photovoltaic technologies, which are well known for their robustness under extreme operating conditions but are limited by high material and fabrication costs and relatively short operational lifetimes.² However, the long-term mechanical and structural stability of layered perovskite device stacks under repeated thermomechanical stress remains a key challenge limiting their broader deployment.

Repeated temperature cycling induces volumetric expansion and contraction within the solar cell stack, giving rise to mechanical fatigue that can ultimately lead to delamination, crack formation, or failure at weakly bonded interfaces.^{2,3} Such effects are particularly pronounced in multilayer thin-film devices due to mismatches in the coefficients of thermal expansion (CTE) between adjacent layers. In perovskite solar

cells, large CTE differences between the glass substrate ($3.7 \times 10^{-6} \text{ K}^{-1}$)⁴, the transparent conductive oxide (e.g., ITO, $8.5 \times 10^{-6} \text{ K}^{-1}$)⁴, and the perovskite absorber (e.g., FAPbI₃, $\sim 203 \times 10^{-6} \text{ K}^{-1}$)⁵ concentrate strain at grain boundaries and heterointerfaces, accelerating mechanical degradation during temperature cycling.

Extreme temperature cycling represents a particularly severe stress condition for thin-film photovoltaic devices and is encountered in several scenarios, including high-altitude platforms, aerospace systems, and accelerated laboratory stress testing. In comparison to standard terrestrial qualification protocols—such as IEC thermal cycling tests typically limited to $-40 \text{ }^\circ\text{C}$ to $+85 \text{ }^\circ\text{C}$ —these conditions involve wider temperature excursions and often faster thermal ramp rates, resulting in rapid stress evolution within the device stack. Environments such as low-Earth orbit (LEO) are frequently cited as representative examples of extreme thermal cycling conditions, where repeated transitions between sunlight and shadow lead to pronounced temperature fluctuations, with cycling frequencies approaching $\sim 6,000$ cycles per year and thermal ramp rates on the order of $4\text{--}5 \text{ }^\circ\text{C min}^{-1}$, resulting in rapid stress evolution within the solar cell stack^{6–9}. Recent systematic studies have also examined perovskite solar cell performance under extreme temperature cycling and in-orbit conditions, further highlighting the importance of understanding thermal fatigue for space deployment.^{10–12} Understanding thermally induced mechanical fatigue under these conditions is therefore essential for improving the durability of thin-film photovoltaic technologies for extreme temperature conditions.”

Results and Discussion:

- Lines 346–348: “After 16 thermal cycles under the applied temperature protocol, encapsulated solar cells with DMSLA retained 84% of their initial performance, compared to 79% for the control group.”
- Lines 384–387: “The high PCE retention observed during MPPT measurements, compared to the lower retention seen in thermal fatigue tests, indicates that extreme temperature cycling may present a more significant challenge than continuous MPPT operation.”

Conclusion and Outlook:

In addition, the Conclusion and Outlook section has been thoroughly revised to reflect the updated scope of the manuscript. References to space deployment, low-Earth orbit operation, and aerospace qualification have been removed. The revised Conclusion now concentrates on the mechanistic understanding of thermomechanical fatigue, grain-boundary reinforcement, and interfacial stabilization under repeated extreme temperature cycling. The Outlook has been reformulated to highlight future research directions focused on reliability-oriented testing, including extended automated cycling and controlled multi-stressor laboratory studies, without suggesting readiness for any specific application environment.

- “In this work, we introduced a dual molecular reinforcement strategy to mitigate thermomechanical degradation in perovskite solar cells subjected to repeated extreme temperature cycling. This extreme cycling induces severe stress at the contact interfaces, particularly due to the much higher CTE of perovskites relative to substrate materials. In our novel two-step strategy, grain-to-grain cohesion was enhanced by incorporating α -lipoic acid (LA) into the perovskite precursor, enabling in situ polymerization during thermal processing, while interfacial adhesion between the perovskite layer and the underlying substrate was strengthened through chemical modification of the disulfide ring to a sulfonium group ($-S^+(CH_3)_2$), a methylated cationic moiety. Ultimately, the functional groups responsible for these effects — closed-ring sulphur, thiol, and sulfonium salt — were shown to form strong, direct interactions with the perovskite crystal lattice. Their combined contribution acts like a molecular suspension system, critically supporting structural integrity during thermal cycling.

While molecular tailoring strategies have been reported in the context of PSCs and flexible devices, our work uniquely combines grain-boundary reinforcement and interface stabilization to address thermal fatigue over a wide temperature range. The observed improvements in performance retention under repeated cycling highlight the importance of targeting mechanically vulnerable regions within multilayer perovskite device stacks, rather than focusing solely on optoelectronic optimization.

Looking forward, further improvements in thermal fatigue resistance may be achieved through the rational design of multifunctional molecular additives capable of controlled cross-linking across both grain boundaries and interfaces. Extending thermal cycling protocols to higher cycle numbers using automated testing platforms

will be essential for establishing long-term degradation trends and for correlating accelerated stress tests with operational lifetime. More broadly, the concepts demonstrated here provide a general framework for improving the durability of perovskite photovoltaics operating under severe temperature cycling conditions.”

R3 - Comment 3.

The proposed additive engineering approach, based on organic additives (lipoic acid and its derivatives), is innovative and could have relevance for improving the durability of perovskite solar cells more broadly. However, the radiation stability of these organic molecules is a fundamental issue that should be addressed, either experimentally or via literature-supported discussion.

R3 - Response 3.

We thank the reviewer for raising this important point. At present, it remains unclear whether the LA and derivative molecules incorporated at grain boundaries and interfaces behave as independent organic species under radiation exposure, or whether they become sufficiently integrated within the perovskite lattice to alter their intrinsic radiation response. Our current laboratory infrastructure does not allow us to immediately assess molecular-level radiation stability. Also, radiation stability is not the focus of this work, and we have revised the text to narrow the scope toward the thermal-cycling behaviour of the devices. However, this is a valid question, and we have checked the literature on this topic. Nonetheless, insights from the literature provide useful context. Studies on sulfur-containing organic compounds, including disulfide-bearing systems structurally related to LA, show that S–S and C–S bonds are among the most radiation-sensitive moieties under high-energy irradiation. For example, Bhattacharyya et al. demonstrate that disulfide bonds undergo electron-induced cleavage, forming radical intermediates that drive bond scission under ionizing radiation.^[R5] Similarly, low-energy secondary electrons generated during γ - or X-ray irradiation are known to cause attachment-induced damage in organic molecules, particularly affecting functional groups with electron-rich sulfur atoms.^[R6] These observations indicate that LA or its derivatives could, in their isolated form, be susceptible to radiation-induced fragmentation. However, this does not imply that using them as ultra-thin layers in devices will lead to the same limitations, which warrants further investigation.

However, it is important to note that actual radiation stability in space is strongly influenced by encapsulation and shielding. In many space-qualified photovoltaic technologies, proper encapsulation significantly reduces the radiation dose absorbed by organic interlayers.^[R7]

In summary, while radiation stability of LA derivatives is a valid topic for future study, it falls outside the revised and clearly defined scope of the present work. Our focus here is on elucidating thermomechanical degradation mechanisms and demonstrating interface-engineering strategies to improve thermal fatigue resistance under repeated extreme temperature cycling.

[R5] Bhattacharyya, R., Dhar, J., Ghosh Dastidar, S., Chakrabarti, P. & Weiss, M. S. The susceptibility of disulfide bonds towards radiation damage may be explained by S...O interactions. *IUCrJ* 7, 825–834 (2020).

[R6] Narayanan S J, J., Tripathi, D., Verma, P., Adhikary, A. & Dutta, A. K. Secondary Electron Attachment-Induced Radiation Damage to Genetic Materials. *ACS Omega* 8, 10669–10689 (2023).

[R7] Zhang, H., Wei, K., Zhang, G. et al. Evaluating the protective efficacy of polymer encapsulation layer for perovskite solar cells under space radiation exposure. *Moore. More* 2, 14 (2025). <https://doi.org/10.1007/s44275-025-00031-6>

R3 - Comment 4.

The lack of comprehensive environmental validation severely limits the impact of the findings for aerospace applications. In its present form, the study is best considered as a proof-of-concept for improved perovskite stability, not as evidence of readiness for space deployment.

R3 - Response 4.

We thank the reviewer for this important and constructive comment. We agree that, in its original form, aspects of the manuscript may have unintentionally implied a level of environmental validation beyond what was experimentally demonstrated. This was not our intention.

In response, we have substantially revised the manuscript to clearly reposition the study as a proof-of-concept investigation into thermomechanical degradation and mitigation strategies under repeated extreme temperature cycling. The revised title, abstract, introduction, results,

and conclusion now explicitly avoid claims related to aerospace readiness or space deployment and instead frame the work as a thermal-fatigue–focused reliability study.

Accordingly, the manuscript no longer suggests comprehensive environmental qualification. Rather, it emphasizes mechanistic insights into how targeted interface and grain-boundary engineering can improve resistance to thermomechanical stress in layered perovskite device stacks. We believe these revisions appropriately align the scope, claims, and impact of the study with the experimental evidence presented and fully address the reviewer’s concern.

R3 - Suggested improvements

R3 - Suggestion 1.

Increase the number of thermal cycles to a more representative level (hundreds or thousands).

R3 - Response 5:

We thank the reviewer for this important suggestion. We agree that extended thermal cycling over hundreds or thousands of cycles would be required for application-level qualification studies. However, the present work is intentionally framed as a reliability-physics investigation aimed at identifying dominant thermomechanical degradation mechanisms, rather than as a qualification or lifetime prediction study.

From a practical standpoint, performing hundreds of cycles under our extreme temperature protocol ($-80\text{ }^{\circ}\text{C}$ to $+80\text{ }^{\circ}\text{C}$ with a ~ 90 -minute cycle duration) is not feasible with the current manually operated setup. Each set of 16 cycles requires approximately 10–12 hours of continuous operation involving liquid nitrogen handling, and extending this to hundreds of cycles would raise significant safety concerns and exceed reasonable laboratory constraints.

To directly address the reviewer’s underlying concern—whether degradation is governed by cycle count or by cumulative thermal exposure—we conducted additional thermal-cycling experiments using shorter cycle durations (2 minutes and 15 minutes), each performed for the same number of cycles (16). These new datasets allow us to decouple the effects of cycle number and total thermal-exposure time.

The results, presented as contour maps and statistical distributions (Figure 4c and Supplementary Figures 30–33), demonstrate that performance degradation correlates primarily with accumulated thermal-exposure time rather than with the number of cycles alone. Devices subjected to rapid or moderate cycling show negligible degradation even after

repeated cycles, whereas longer dwell times at extreme temperatures result in measurable fatigue.

We believe this approach provides a mechanistically meaningful alternative to simply increasing cycle count and directly addresses the reviewer's concern within the clarified scope of the study.

R3 - Suggestion 2.

Include tests under vacuum condition and radiation exposure (protons, electrons, γ -rays).

R3 - Response 6:

We thank the reviewer for highlighting the importance of vacuum and radiation testing. We fully agree that such experiments are essential for assessing device suitability for aerospace deployment. However, we emphasize that the revised manuscript does not aim to establish aerospace readiness or space qualification.

At present, our laboratory infrastructure does not permit controlled device-level irradiation experiments or combined thermo-vacuum cycling under extreme temperature conditions. Moreover, standardized protocols for radiation testing of perovskite photovoltaics—including particle type, energy, fluence, and dose relevance—are still under active discussion within the community, making it challenging to interpret isolated radiation experiments in a broadly applicable manner.

Importantly, meaningful assessment of radiation effects would ideally require multi-stressor testing, combining radiation exposure with thermal cycling and vacuum conditions, as isolated stressors alone do not represent realistic operational environments. To the best of our knowledge, such coupled experimental platforms are not yet widely available.

In light of these considerations, vacuum and radiation testing are beyond the scope of the present study. To avoid overinterpretation, we have removed all implications of aerospace qualification from the title, abstract, introduction, and conclusion, and now frame the work strictly as a study of thermomechanical fatigue under extreme temperature cycling.

R3 - Suggestion 3.

Provide a discussion of the radiation stability of α -lipoic acid derivatives, supported by prior studies.

R3 - Response 7:

We thank the reviewer for raising this valid point. The radiation stability of organic molecules, including sulphur-containing compounds such as α -lipoic acid and its derivatives, is indeed an important consideration for space applications.

As discussed in detail in our response to Reviewer #3, Comment 3, literature reports indicate that disulfide- and sulphur-containing moieties can be susceptible to radiation-induced bond scission. We acknowledge this limitation and cite relevant studies in the response letter to provide context.

However, in the revised manuscript we have narrowed the scope to focus exclusively on thermomechanical degradation under extreme temperature cycling, and the manuscript no longer makes claims related to radiation tolerance or aerospace deployment. Accordingly, a detailed radiation-stability discussion is not required to support the conclusions of the present study.

Radiation stability of LA-based additives remains an important topic for future work once appropriate testing facilities become available.

Reviewer #4

Remarks to the Author:

This manuscript presents an interesting study on the use of additives to enhance the thermal fatigue resistance of perovskite solar cells for space applications. The work is novel and offers valuable insights, especially since thermal fatigue has received comparatively little attention in this field.

I recommend a revision before the manuscript can be accepted for publication. My detailed comments are as follows:

Response:

We thank the reviewer for the positive evaluation of our work and for recognizing the novelty and relevance of our study on enhancing the thermal fatigue resistance of perovskite solar cells. We appreciate the constructive guidance and the opportunity to revise the manuscript. In the revised version, we have carefully addressed all reviewer comments in detail and have strengthened the clarity, accuracy, and scope of the manuscript accordingly. Below, we

provide a point-by-point response to each comment and outline all corresponding changes made in the text.

R4 - Comment 1.

Since the context is space applications, why was AM0 not used for device measurements? AM0 is more relevant, and its higher UV component could also influence additive stability. Please provide AM0 measurements and comparisons, as this seems necessary.

R4 - Response 1.

We thank the reviewer for these suggestions. Indeed, this is a critical point and following the reviewer's recommendation, we now added $J-V$ curves and statistical distribution of the corresponding devices at AM0 (1360 W/m^2). The devices obtained higher J_{sc} and V_{oc} at AM0 compared to those at AM1.5 spectrum.

Supplementary Figure 28. $J-V$ curves and the corresponding device parameters of the devices at AM0.

Supplementary Figure 29. Statistical distribution of a) V_{oc} , b) PCE, c) FF, and (d) J_{sc} values of the devices at AM0. Each condition had 10 devices.

R4 - Comment 2.

In the adhesion test, the ranking is DMSLA > DHLA > LA > control. However, this ranking does not align with the thermal cycling results. Does this suggest that adhesion is not the main factor contributing to thermal cycling failure? Please explain and discuss possible alternative failure modes with supporting evidence.

R4 - Response 2.

We thank the reviewer for this insightful comment. While the adhesion measurements show a clear ranking of DMSLA > DHLA > LA > control, we agree that this ranking does not directly mirror the thermal-cycling stability results. This discrepancy is consistent with the

fact that thermomechanical degradation in perovskite solar cells is governed by more than interfacial adhesion alone.

First, our nanomechanical AFM data (Fig. 1c and Supplementary Figs. 9–10) show that LA does not measurably increase adhesion at grain boundaries, whereas DHLA and DMSLA improve grain-boundary adhesion by >50% and ~40%, respectively. This indicates that the polymer network formed from LA during annealing provides the weakest mechanical reinforcement in the regions most susceptible to strain accumulation. Second, the pull-off tests (Fig. 2c–d) reveal that LA achieves only a modest increase in substrate–perovskite adhesion, far below that obtained with DMSLA. These two measurements consistently show that LA-treated films are mechanically less robust both at the grain boundaries and at the ITO/perovskite interface. Correspondingly, LA-only devices exhibit the lowest performance retention after thermal cycling (64%), supporting the conclusion that insufficient mechanical reinforcement leads to more pronounced degradation under repeated temperature excursions.

In contrast, DMSLA combines strong adhesion with greater chemical stability and stronger interfacial interactions, as supported by our DFT analysis (highest interfacial interaction energy; Fig. 2e–f), XPS peel-off results (largest –COO signal on the perovskite side), cyclic voltammetry (highest SAM surface coverage), and UPS (strongest work-function shift due to its large dipole moment). These properties allow the DMSLA-modified interface to better withstand repeated thermomechanical loading, explaining its superior thermal-cycling stability despite DHLA having comparable or slightly higher initial adhesion in some measurements.

Taken together, our results indicate that adhesion is necessary but not sufficient for thermal-fatigue resistance. Thermal cycling exposes the device to volumetric expansion and contraction across multiple interfaces and grain boundaries; therefore, overall stability depends on the combined effects of interfacial adhesion, chemical robustness, and the ability of the grain-boundary network to dissipate strain. We have added a concise explanation in the revised manuscript to clarify this point.

- Lines 352-360: “We note that the adhesion ranking (DMSLA > DHLA > LA > control) does not directly mirror the thermal-cycling results because adhesion represents only one component of thermomechanical stability. Thermal cycling also depends on the chemical robustness of the linker molecules and the ability of the grain boundaries to accommodate strain. DMSLA combines strong and stable interfacial

bonding—supported by our DFT, XPS, CV, and UPS data—with higher chemical stability under thermal stress, whereas DHLA’s thiol groups are more reactive and may undergo changes during cycling. This interplay of interfacial adhesion, chemical stability, and grain-boundary reinforcement explains the superior cycling performance of DMSLA.”

R4 - Comment 3.

For the thermal cycling tests performed using the in-house equipment, was moisture controlled? The encapsulation method described in the paper may not effectively block moisture, and heating/cooling cycles could allow moisture ingress, which would affect device performance. This condition would not reflect the space environment, so clarification is needed.

R4 - Response 3.

We thank the reviewer for raising this critical point. The thermal cycling experiments were conducted in a custom-designed closed-lid setup that ensured a controlled and moisture-minimized environment throughout the entire test. The chamber was continuously purged with dry air before and during the measurements to eliminate any residual humidity and prevent frost formation on the samples, although it is difficult to say the test environment is humidity-free. A DHT22 temperature–humidity sensor monitored the internal conditions in real time, and cycling was initiated only once the humidity reached 0% relative humidity (sensor range 0–100% RH, $\pm 2\%$ accuracy, $\pm 5\%$ maximum deviation). The dry-air circulation was maintained during both heating and cooling phases to ensure that no moisture ingress occurred through the encapsulation or chamber interfaces. Moisture ingress is still possible, but all samples are encapsulated under identical conditions; therefore, relative comparisons remain valid, and performance differences can primarily be linked to the interface we study.

The following modifications have been implemented in **Section 2.4, “Thermal Fatigue Stability Analysis”**:

- “We performed the thermal fatigue test using a custom-built close-lid setup, as standard climate chambers could not reach the required $-80\text{ }^{\circ}\text{C}$ cryogenic range, which induces the highest stress at the perovskite–substrate interface. The samples were cycled between $-80\text{ }^{\circ}\text{C}$ and $+80\text{ }^{\circ}\text{C}$ in a stainless steel container, with controlled heating and cooling rates of $+3.40$ and $-3.80\text{ }^{\circ}\text{C min}^{-1}$, respectively, and a total cycle

duration of ~90 minutes under dry ambient conditions (Figures 4a, b). A schematic and further explanation of the experimental setup are provided in Supplementary Figure 26.”

More detailed information on the thermal cycling procedure is presented in the Supporting Information under Methods titled as “**Thermal Cycling Tests**”:

- “Thermal cycling experiments were performed using a custom-built closed-lid setup designed to achieve reproducible and well-controlled temperature cycling between –80 °C and +80 °C. To prevent frost formation and maintain dry ambient conditions, dry air was circulated within the sealed chamber, and cycling was initiated only after the integrated DHT22 temperature–humidity sensor indicated 0% relative humidity. A schematic representation of the thermal fatigue experimental setup is provided in **Supplementary Figure 26.**”

R4 - Comment 4.

The EQE results indicate that the front interface (light-incidence side) does not improve, and there is a noticeable loss at the rear interface when additives are used. Please explain this observation. The evidence for interfacial passivation is not very strong from EQE results.

R4 - Response 4.

Thanks for the reviewer's keen observation and valuable question. The factors influencing EQE results are complex, including non-radiative recombination in the bulk and at interfaces, as well as carrier extraction and transport efficiency in charge transport layer, etc. Therefore, the passivation effect cannot be simply inferred based solely on EQE results. Other reports on buried surface modification showed a similar trend (Small 2024, 20, 2404058; Nature, 2025, DOI: 10.1038/s41586-025-09785-3; J. Am. Chem. Soc. 2025, 147, 27, 23683-23695; Angew. Chem. Int. Ed. 2025, 64, e202504237).

We thank the reviewer for this important observation. The EQE response reflects the combined effects of optical absorption, bulk recombination, interfacial recombination, and charge-extraction efficiency. Because these processes are intertwined, the degree of passivation at the buried HTL/perovskite interface cannot be directly inferred from the EQE spectrum alone. In our devices, the molecular modifications introduced through the SAM layer are electronically active but optically negligible, and therefore large changes at the short-wavelength (illumination-side) region of the EQE are not expected.

Although the EQE does not explicitly show an improvement at the front interface, multiple electrical and photophysical measurements consistently indicate reduced recombination for DMSLA-treated devices. Specifically, the DMSLA samples exhibit higher V_{OC} and FF, as well as enhanced PL/ iV_{oc} values and longer TRPL lifetimes, all of which point to suppressed non-radiative recombination at the HTL/perovskite interface. These metrics are more sensitive probes of buried-interface passivation than EQE alone.

The modest reduction in EQE at longer wavelengths is consistent with reports on buried-interface SAM modifications, where the introduction of an ultrathin molecular layer can slightly redistribute the optical field or influence carrier-collection dynamics without compromising overall device performance (Small 2024, 20, 2404058; Nature 2025, DOI: 10.1038/s41586-025-09785-3; JACS 2025, 147, 23683–23695; Angew. Chem. Int. Ed. 2025, 64, e202504237). Importantly, the integrated JSC values remain consistent with J–V measurements, and the overall PCE improves for the DMSLA-treated devices, confirming that the interfacial enhancement is captured more effectively in electrical and photoluminescence measurements than in the EQE spectrum.

R4 - Comment 5.

For the MPPT test, please provide the corresponding IV curves and the initial PCE data.

R4 - Response 5.

Thanks for the reviewer's reminder. We have added the corresponding J – V curves and the initial PCE data.

Supplementary Figure 27. The corresponding J - V curves and the initial PCE data for the MPPT.

Reviewer #5

Remarks to the Author:

R4 - Response 5.

We would like to express our sincere appreciation for your contribution to the evaluation of our manuscript. We also appreciate Nature Communications' initiative to involve and recognize Early Career Researchers in co-reviewing.

Supplementary Table 2. Summary of reported thermal-cycling studies on perovskite solar cells under space-relevant or terrestrial conditions.

Study	Target Application	Temp Range (°C)	Cycles	Cycle Duration	Initial PCE (%)	Final PCE (%)	PCE Retention	Encapsulation	Key Finding	DOI
Lu et al., 2025	General space	-123 to +157	5	300 m	24.39	20.15	82.7%	Ultrahigh vacuum; no encapsulation	Absorber decomposition to Pb/PbI ₂ , volatile organics, ion migration under vacuum thermal cycling	10.22541/au.175547361.10502563/v1
Li et al., 2022	General space	-160 to +150	50	30 °C min ⁻¹ rate, 20 m	Not reported	Not reported	97% (AMO)	Not specified	Temperature-dependent phase transitions and recoverable lattice strain; widest temperature range tested	10.1002/aenm.202202887
Yang et al., 2024	Near-space/cryogenic	-143 to +17	120	81 m	24.34 (at 150K)	~17.5 (estimated)	72%	Not specified	PAN additive stabilized lattice at low temperatures; 72% retention after 120 cryogenic cycles (highest cycle count at extreme T)	10.1002/aenm.202400638
Guixiang Li et al., 2023	Terrestrial	-60 to +80	120	18 m	24.6	Not reported	93.9% at 80°C and 88.7% at -60°C	Unencapsulated	Impact of the ordered dipolar structure on the operational stability	10.1126/science.add7331
Zhihao Li et al., 2024	General space	-40 to +80	200		25.78 (rigid), 24.54 (flexible)	Not stated	>95%	Unencapsulated	In-situ cross-linked polymer; 200 thermal cycles + 10,000 bending cycles; >95% retention	10.1002/ange.202421063
Bautista et al., 2022	LEO CubeSat	-40 to +80	250	25 m	24.28	9.7 (estimated)	40%	UV cut film and polyimide/Kapton tape/ UV light curable glue/1 mm glass	HTM comparison under simulated space environment; carbon HTM outperformed organic/inorganic HTMs	10.2322/tjsass.65.95
Chen et al., 2024	Terrestrial/space readiness	-40 to +85	5000	5 m	21.3	17.2	81%	Unencapsulated	Alkyl-ammonium additive reduced residual stress; HIGHEST CYCLE COUNT for perovskites (2,500 cycles)	10.1021/acsnenergylett.4c00988
Kim et al., 2025		-80 to +80	200	40 m	Not stated	Not stated	>90%	Vacuum-packed	Non-volatile 4CP stabilizes Li ⁺ , suppresses oxides, enabling ~26% efficient, shock-stable perovskite cells	10.1038/s41560-025-01864-z
Our work	Extreme temperature cycling	-80 to +80	16	90 m			84%	Space grade encapsulation, DOWSIL 93-500/0.2 mm glass	Dual grain-boundary and interfacial reinforcement improves resistance to thermomechanical degradation under repeated extreme temperature cycling.	

Response to Reviewers' Comments on NCOMMS-25-6746B

We sincerely thank all reviewers for their thorough, constructive, and encouraging evaluations of our revised manuscript. We are grateful for the positive feedback regarding the improved clarity, scope, and scientific focus of the work, and we appreciate the reviewers' recognition of the substantial revisions made to address the initial concerns. Their comments have significantly contributed to strengthening the manuscript. Please find our detailed, point-by-point responses to each remark below.

Reviewer #2 (Remarks to the Author):

Several reviewers raised similar questions regarding the inclusion of additional information related to space applications in response to the originally submitted manuscript. Although this was not the authors' original intention, as explained in their responses to multiple reviewers, the initial manuscript (title) was somewhat misleading. This issue has now been recognised and corrected by the authors. The revised and rewritten abstract, introduction, and conclusion show significant improvement and now clearly focus on the work being addressed.

Based on the through revision, it is recommended to be accepted for publish in this journal.

Response:

We thank the reviewer for their careful reassessment of the revised manuscript and for the positive evaluation. We appreciate the acknowledgement that the revised title and the rewritten Abstract, Introduction, and Conclusion now clearly define the scope and intent of the study. We are grateful for the reviewer's constructive feedback and their recommendation for publication.

In response to the reviewer's suggestion, we have revised Supplementary Fig. S26 to more clearly illustrate the custom-designed closed-lid thermal cycling setup, with improved visual clarity.

Supplementary Figure 26: Schematic representation of the thermal fatigue experimental setup.

Reviewer #3 (Remarks to the Author):

Report on NCOMMS-25-67465B

Title: Perovskite Solar Cells with Enhanced Thermal Fatigue Resistance under Extreme Temperature Cycling

by Cem Yilmaz et al.

In the revised version of the manuscript, the authors have effectively restructured the abstract, introduction, and conclusions, removing inappropriate references to “space readiness” and clearly defining the study’s aim as an investigation of thermal fatigue mechanisms in perovskite solar cells subjected to extreme temperature cycling. This reformulation makes the manuscript more coherent and scientifically robust.

In light of this repositioning, the concerns raised in the first review regarding the absence of vacuum tests, ionizing radiation, or other space-relevant environmental factors are no longer critical. The authors’ decision not to include such tests is now fully justified, as the work is correctly framed as a proof-of-concept on the mechanical and thermal stability of perovskites, without the ambition of demonstrating suitability for space missions.

We also appreciate the authors' detailed explanation of the limited number of thermal cycles. Technical and safety considerations related to the use of liquid nitrogen, the need for authorized personnel to refill the tanks, the long duration of the measurements, and the manual nature of the experimental setup make this methodological choice fully understandable. It is also reasonable, as the authors did, to stop the tests once the device performance drops below 90% of the initial PCE.

Although the protocol remains limited to 16 cycles, preventing definitive confirmation of the statement that "most degradation occurs primarily in the first cycles," the inclusion of supplementary experiments with shorter cycles (2 and 15 minutes) is valuable. These experiments allow the effect of the number of cycles to be distinguished from those of total thermal exposure time. The new data, including contour plots, boxplot distributions, and a comparative table, convincingly demonstrate that degradation is primarily governed by the cumulative thermal exposure and the associated thermomechanical stress, while the number of cycles plays a secondary role.

We also note that the manuscript could benefit from a brief clarification in the main text regarding the number of devices tested and the number of pixels per device. While this information is already provided in the Supplementary Information, a concise inclusion in the main text would improve methodological transparency and facilitate an immediate assessment of the statistical robustness of the results. This is a formal refinement that would make the manuscript more accessible without affecting its scientific integrity.

In summary, the revisions significantly improve the manuscript compared to the first version. The main conceptual concerns have been effectively addressed, and the authors' responses are satisfactory. Overall, the work is now more balanced, rigorous, and appropriately focused on its scientific objectives. We therefore consider the manuscript, in its current form, suitable for publication in Nature Communications.

Response:

We sincerely thank the reviewer for their thorough and constructive reassessment of the revised manuscript. We greatly appreciate the recognition that the abstract, introduction, and conclusions have been effectively restructured to remove inappropriate references to "space readiness" and to clearly define the scope of the study as an investigation of thermal fatigue mechanisms in perovskite solar cells subjected to extreme temperature cycling. We are pleased that this repositioning is considered to enhance the coherence and scientific robustness of the work.

We also thank the reviewer for noting that, in light of this revised framing, the absence of vacuum testing, ionizing radiation exposure, and other space-relevant environmental stressors is no longer a critical limitation. As correctly highlighted, the present study is intentionally positioned as a proof-of-concept focused on thermomechanical and interfacial stability, rather than on demonstrating suitability for space missions.

We appreciate the reviewer's understanding regarding the limited number of thermal cycles investigated. As noted, practical and safety constraints associated with liquid-nitrogen-based

thermal cycling, the manual nature of the setup, and the long duration of each cycle require a careful balance between experimental depth and feasibility. We also agree that terminating the tests once device performance drops below 90% of the initial PCE represents a reasonable and commonly adopted criterion in accelerated stability studies.

Although the main protocol remains limited to 16 cycles, we are grateful that the reviewer finds the additional experiments with shorter cycle durations (2 and 15 minutes) valuable. These supplementary datasets enable a clear distinction between the effects of cycle count and those of cumulative thermal-exposure time. As highlighted by the reviewer, the newly added contour plots, boxplot distributions, and comparative literature table demonstrate that degradation is primarily governed by accumulated thermomechanical stress associated with total thermal exposure, while the number of cycles plays a secondary role.

In response to the reviewer's suggestion regarding methodological transparency, we have now explicitly clarified the number of devices tested and the number of pixels per device directly in the main text:

- Lines 343–346: “The samples—**one device per condition, each comprising six pixels**—were cycled between $-80\text{ }^{\circ}\text{C}$ and $+80\text{ }^{\circ}\text{C}$ in a stainless-steel container, with controlled heating and cooling rates of $+3.40$ and $-3.80\text{ }^{\circ}\text{C min}^{-1}$, respectively, and a total cycle duration of ~ 90 minutes under dry ambient conditions (**Figures 4a, b**).”

We are grateful for the reviewer's positive assessment of the revised manuscript and for their conclusion that the work is now more balanced, rigorous, and appropriately focused on its scientific objectives. We thank the reviewer for their careful evaluation and for considering the manuscript suitable for publication in *Nature Communications*.

Reviewer #4 (Remarks to the Author):

I thank the authors for their substantial efforts in addressing my comments. The manuscript is in a good shape for acceptance.

One minor comment: could the authors justify the standards or guidelines used to define the thermal cycling protocol, including why the ramp rate of 3 is used and dwell duration? Is there a defined pass/fail criterion for the thermal cycling test at specific temperature ranges? Additionally, how do these testing conditions and criteria compare with those reported in other studies or used for space solar cell technologies? It will be helpful to let the reader know how the perovskite based cell is a good candidate for space application.

Response:

We thank the reviewer for their positive assessment of the revised manuscript and for this constructive suggestion. We appreciate the opportunity to further clarify the rationale behind the thermal cycling protocol and its relation to existing standards and literature.

The temperature window of $-80\text{ }^{\circ}\text{C}$ to $+80\text{ }^{\circ}\text{C}$ was selected on the basis of practical temperature measurements reported for low-Earth-orbit satellites and CubeSat missions, which document rapid and repeated transitions across comparable temperature extremes

during orbital day–night cycling^{1,2}. This range intentionally exceeds standard terrestrial photovoltaic qualification protocols (e.g., IEC 61215, –40 °C to +85 °C) and was chosen to impose severe thermomechanical stress at the substrate–perovskite interface, where mismatches in thermal expansion coefficients are most pronounced.

The total cycle duration of approximately 90 minutes was initially selected to approximate the characteristic orbital period of low-Earth orbit, consisting of alternating exposure to illumination and eclipse phases. To reflect this simplified thermal scenario in a controlled laboratory environment, each cycle was designed to include approximately 45 minutes of heating and 45 minutes of cooling between the temperature extrema. Based on this cycle duration and temperature window, average heating and cooling ramp rates of approximately 3–4 °C min^{–1} naturally result and were therefore adopted.

We emphasize that this protocol is not intended as a formal space-qualification test, but rather as an accelerated and physically motivated stress test aimed at probing thermomechanical fatigue mechanisms. Accordingly, no strict pass/fail criterion at specific temperatures was applied; instead, device performance was monitored, and a pragmatic stopping criterion of 90% of the initial power conversion efficiency of the control devices was adopted, consistent with common practice in accelerated stability testing of emerging photovoltaic technologies.

To clarify these aspects for the reader, we have made the following additions to the manuscript:

- Lines 116-120: “Furthermore, to approximate the mechanical stresses associated with these rapid and extreme temperature shifts, we adopted a custom thermal cycling protocol spanning –80 °C to +80 °C, consistent with practical satellite temperature measurements and recent thermal stress studies on perovskite solar cells,²⁶⁻²⁸ which necessitated the development of a dedicated test setup.”
- Lines 347-350: “The 90-minute cycle duration was chosen to approximate a symmetric heating–cooling sequence (~45 minutes per half-cycle), reflecting characteristic thermal timescales reported for low-Earth-orbit environments³⁵ and ensuring sufficient thermal equilibration of the full device stack. Thermal cycling was continued until the performance of the target devices dropped below ~90% of their initial power conversion efficiency.”

Reviewer #5 (Remarks to the Author):

Response:

We would like to thank the co-reviewer for their contribution as part of the Nature Communications peer-review training initiative. We appreciate the time and effort invested in the careful evaluation of our work and the constructive input provided through this process.

- 1 Bulut, M. & Sözbir, N. THERMAL DESIGN, ANALYSIS AND TEST VALIDATION OF TURKSAT-3USAT SATELLITE. *Journal of Thermal Engineering* **7**, 468-482 (2021).
<https://doi.org/10.18186/thermal.887316>
- 2 Lamb, D. A., Irvine, S. J. C., Baker, M. A., Underwood, C. I. & Mardhani, S. Thin film cadmium telluride solar cells on ultra-thin glass in low earth orbit—3 years of performance data on the AlSat-1N CubeSat mission. *Progress in Photovoltaics: Research and Applications* **29**, 1000-1007 (2021).
<https://doi.org/https://doi.org/10.1002/pip.3423>